# A comparison of catchment travel times and storage deduced from deuterium and tritium tracers using StorAge Selection functions

Nicolas B. Rodriguez[1,2], Laurent Pfister[1], Erwin Zehe[2], and Julian Klaus[1]

[1]Catchment and Eco-Hydrology Research Group, Environmental Research and Innovation Department, Luxembourg Institute of Science and Technology, Belvaux, Luxembourg
[2]Institute of Water Resources and River Basin Management, Karlsruhe Institute of Technology, Karlsruhe, Germany

**Correspondence:** Nicolas Rodriguez (nicolas.bjorn.rodriguez@gmail.com)

**Abstract.** Catchment travel time distributions (TTDs) are an efficient concept to summarize the time-varying 3-dimensional transport of water and solutes towards an outlet in a single function of water age and to estimate catchment storage by leveraging information contained in tracer data (e.g. deuterium $^2$H and tritium $^3$H). It is argued that the preferential use of the stable isotopes of O and H as tracers compared to tritium has truncated our vision of streamflow TTDs, meaning that the long tails of the distribution associated with old water tend to be neglected. However, the reasons for the truncation of the TTD tails are still obscured by methodological and data limitations. In this study, we went beyond these limitations and evaluated the differences between streamflow TTDs calculated using only deuterium ($^2$H) or only tritium ($^3$H). We also compared mobile catchment storage (derived from the TTDs) associated with each tracer. For this we additionally constrained a model that successfully simulated high-frequency stream deuterium measurements with 24 stream tritium measurements over the same period (2015–2017). We used data from the forested headwater Weierbach catchment (42 ha) in Luxembourg. Time-varying streamflow TTDs were estimated by consistently using both tracers within a framework based on StorAge Selection (SAS) functions. We found similar TTDs and similar mobile storage between the $^2$H- and $^3$H-derived estimates, despite statistically significant differences for certain measures of TTDs and storage. The streamflow mean travel time was estimated at 2.90±0.54 years using $^2$H and 3.12±0.59 years using $^3$H (mean ± one standard deviation). Both tracers consistently suggested that less than 10% of stream water in the Weierbach is older than 5 years. The travel time differences between the tracers were small compared to previous studies in other catchments, and contrary to prior expectations, we found that these differences were more pronounced for young water than for old water. The found differences could be explained by the calculation uncertainties and by a limited sampling frequency for tritium. We conclude that stable isotopes do not seem to systematically underestimate travel times or storage compared to tritium. Using both stable and radioactive isotopes of H as tracers reduced the travel time and storage calculation uncertainties. Tritium and stable isotopes both had the ability to reveal short travel times in streamflow. Using both tracers together better exploited the more specific information about longer travel times that $^3$H inherently contains due to its radioactive decay. The two tracers thus had different information contents overall. Tritium was slightly more informative than stable isotopes for travel time analysis despite a lower number of tracer samples. In the future, it would be useful to similarly test the consistency of travel time estimates and the potential differences in travel time information contents between those

tracers in catchments with other characteristics or with a considerable fraction of stream water older than 5 years, since this could emphasize the role of the radioactive decay of tritium for discriminating younger from older water.

## 1   Introduction

Sustainable water resource management is based upon a sound understanding of how much water is stored in catchments, and
how it is released to the streams. Isotopic tracers such as deuterium ($^2$H), oxygen 18 ($^{18}$O), and tritium ($^3$H) have become the cornerstone of several approaches to tackle these two critical questions (Kendall and McDonnell, 1998). For instance, hydrograph separation using stable isotopes of O and H (Buttle, 1994; Klaus and McDonnell, 2013) has unfolded the difference between catchments hydraulic response (i.e. streamflow) and chemical response (e.g. solutes) (Kirchner, 2003) related to the different concepts of water celerity and water velocity (McDonnell and Beven, 2014). Isotopic tracers have also been the
backbone to unravel water flow paths in soils (Sprenger et al., 2016), and to distinguish soil water going back to the atmosphere and flowing to the streams (Brooks et al., 2010; McDonnell, 2014; McCutcheon et al., 2017; Berry et al., 2018; Dubbert et al., 2019).

The determination of travel time distributions (TTDs) is the method relying the most on isotopic tracers (McGuire and McDonnell, 2006). TTDs provide a concise summary of water flow paths to an outlet by leveraging the information on storage
and release contained in tracer input-output relationships. TTDs are essential to link water quantity to water quality (Hrachowitz et al., 2016), for example by allowing calculations of stream solute dynamics from a hydrological model (Rinaldo and Marani, 1987; Maher, 2011; Benettin et al., 2015a, 2017a). TTDs are commonly calculated from isotopic tracers in many sub-disciplines of hydrology and thus have the potential to link the individual studies focused on the various compartments of the critical zone (e.g. groundwater and surface water) (Sprenger et al., 2019). $^3$H has been used as an environmental tracer since the late 1950s
(Begemann and Libby, 1957; Eriksson, 1958; Dinçer et al., 1970; Hubert et al., 1969; Martinec, 1975) and it gained particular momentum in the eighties with its use in diverse TTD models (Małoszewski and Zuber, 1982; Stewart et al., 2010). It is argued that $^3$H contains more information on travel times than stable isotopes due to its radioactive decay (Stewart et al., 2012). For example, low tritium content generally indicates old water in which most of the $^3$H from nuclear tests has decayed. Despite its potential, $^3$H is used only rarely in travel time studies nowadays (Stewart et al., 2010), most likely because high precision
analyses are laborious (Morgenstern and Taylor, 2009) and rather expensive. In contrast, the use of stable isotopes in travel time studies has soared in the last three decades (Kendall and McDonnell, 1998; McGuire and McDonnell, 2006; Fenicia et al., 2010; Heidbuechel et al., 2012; Klaus et al., 2015a; Benettin et al., 2015a; Pfister et al., 2017; Rodriguez et al., 2018). This is notably due to the fast and low-cost analyses provided by recent advances in laser spectroscopy (e.g. Lis et al., 2008; Gupta et al., 2009; Keim et al., 2014) and the associated technological progress in sampling techniques of various water sources
(Berman et al., 2009; Koehler and Wassenaar, 2011; Herbstritt et al., 2012; Munksgaard et al., 2011; Pangle et al., 2013;

Herbstritt et al., 2019). According to Stewart et al. (2012) and Stewart and Morgenstern (2016), the limited use of $^3$H may have cause a biased or "truncated" vision of stream TTDs, in which the long TTD tails remain mostly undetected by stable isotopes. Longer mean travel times (MTT) were inferred from $^3$H than from stable isotopes in several studies employing both tracers (Stewart et al., 2010). Longer MTTs may have profound consequences for catchment storage, usually estimated from

TTDs as $S = Q \times MTT$ (with $Q$ the flux through the catchment), assuming steady-state flow conditions (i.e. $S(t) = \overline{S(t)} = S$, $Q(t) = \overline{Q(t)} = Q$, $MTT(t) = \overline{MTT(t)} = MTT$) (McGuire and McDonnell, 2006; Soulsby et al., 2009; Birkel et al., 2015; Pfister et al., 2017). Under this assumption, a truncated TTD would result in an underestimated MTT thus an underestimated catchment storage. A different perspective on catchment storage and on its relation with travel times may however be adopted by calculating storage from unsteady TTDs.

A water molecule that reached an outlet has only one travel time, defined as the duration between entry and exit. The use of different methods of travel time analysis for stable isotopes of O and H and for $^3$H (e.g. amplitudes of seasonal variations vs. radioactive decay) was first pointed out as a main reason for the discrepancies in MTT (Stewart et al., 2012). Further research is thus needed for developing mathematical frameworks that coherently incorporate stable isotopes of O and H and $^3$H in travel time calculations. Moreover, several limiting assumptions were used in previous studies employing $^3$H to derive

the MTT, which is in itself an insufficient statistic to describe various aspects (e.g. shape, modes, percentiles) of the TTDs. For example, the steady-state flow assumption has been used in almost all $^3$H travel time studies (McGuire and McDonnell, 2006; Stewart et al., 2010; Cartwright and Morgenstern, 2016; Duvert et al., 2016; Gallart et al., 2016). Yet, time variance is a fundamental characteristic of TTDs (Botter et al., 2011; Rinaldo et al., 2015), and it has been acknowledged in simulations of stream $^3$H only very recently (Visser et al., 2019). Hydrological recharge models or tracer weighting functions have also

been employed to account for the influence of the mixing of precipitation tracer values in the unsaturated zone and for the influence of the seasonal (hence time-varying) losses to atmosphere via $ET(t)$ (e.g., Małoszewski and Zuber, 1982) on the catchment inputs in $^3$H (Stewart et al., 2007). However, these methods do not explicitly represent the influence of the TTD of $ET$ on the age-labeled water balance and thus represent indirect approximations. In contrast, explicit considerations of $ET$ and of the influence of its TTD on the streamflow TTD are becoming common for stable isotopes (van der Velde et al., 2015;

Visser et al., 2019). Finally, more guidance on the calibration of the TTD models against $^3$H measurements is needed (see e.g. Gallart et al., 2016). Especially, uncertainties of $^3$H-inferred travel times may have been overlooked, while these could explain the differences to the stable isotope-inferred travel time estimates.

Besides methodological problems, the reasons for the travel time differences (hence apparent storage or mixing) are still not well understood, because little is known about the difference in information content of $^3$H compared to stable isotopes when

determining TTDs. First, $^3$H sampling in catchments typically differs from stable isotope sampling in terms of frequency and flow conditions. Stable isotope records in precipitation and in the streams have lately shown increasing resolution, covering a wide range of flow conditions (McGuire et al., 2005; Benettin et al., 2015a; Birkel et al., 2015; Pfister et al., 2017; von Freyberg et al., 2017; Visser et al., 2019; Rodriguez and Klaus, 2019). Tritium records in precipitation and streams are on the other hand usually at a monthly resolution in many places around the globe (IAEA and WMO, 2019; IAEA, 2019; Halder et al.,

2015). Only a handful of travel time studies employing $^3$H report more than a dozen stream samples for a given site and for conditions other than baseflow (e.g. Małoszewski et al., 1983; Visser et al., 2019). This general focus on baseflow $^3$H sampling introduces by design a bias towards older water. Second, the natural variability of $^3$H compared to that of stable isotopes has rarely been documented. $^3$H in precipitation has returned to the pre-bomb levels, and like stable isotopes it shows a clear yearly seasonality (e.g. Stamoulis et al., 2005; Bajjali, 2012). However, ambiguous travel time estimates may still be obtained with $^3$H in the northern hemisphere because the current precipitation has similar $^3$H concentrations than water recharged in the 1980s (Stewart et al., 2012). Higher sampling frequencies of precipitation $^3$H are almost nonexistent. Rank and Papesch (2005) revealed a short term variability of precipitation $^3$H likely due to different air masses. This variability was observed also during complex meteorological conditions such as hurricanes (Östlund, 2013). $^3$H in streams also exhibits yearly seasonality (Różański et al., 2001; Rank et al., 2018), but short term dynamics are not understood well because high frequency data sets are limited. Dinçer et al. (1970) showed that short-term stream tritium variations can be caused by the melting of the snowpack from the current and the previous winters. In addition, the seasonally-higher values of precipitation $^3$H in spring could explain some of the $^3$H peaks observed in the large rivers (Rank et al., 2018). More studies employing both $^3$H and stable isotopes and comparing their travel time information content are therefore crucial to understand travel times in catchments from a multi-tracer perspective.

In this study, we go beyond previous work and assess the differences between streamflow TTDs and the associated catchment storage (considering their uncertainties) when those are inferred from stable isotopes or from $^3$H measurements used in a coherent mathematical framework for both tracers. For this, we use high frequency isotopic tracer data from an experimental headwater catchment in Luxembourg. Here we focus on the stable isotope of H (deuterium $^2$H) for which we have more precise measurements than for oxygen 18. A transport model based on TTDs was recently developed and successfully applied to simulate a two-year high frequency (sub-daily) record of $\delta^2$H in the stream (Rodriguez and Klaus, 2019). Here, we additionally constrain the same model within the same mathematical framework against 24 stream samples of $^3$H collected during highly varying flow conditions over the same period as for $^2$H. We do not assume steady-state flow conditions and we employ StorAge Selection functions to account for the type and the variability of the TTDs of Q and ET that affect the water age balance in the catchment. The tracer input-output relationships and the $^3$H radioactive decay are accounted for in the method, which reduces $^3$H-derived travel time ambiguities usually due to similar tritium activities between recent precipitation and the water recharged since the eighties. We provide guidance on how to jointly calibrate the model to both tracers and on how to derive likely ranges of storage estimates and travel time measures other than the MTT. This work addresses the following related research questions:

  – Are travel times and storage inferred from a common transport model for $^2$H and $^3$H in disagreement?

  – Are the travel time information contents of $^2$H and $^3$H similar?

## 2 Methods

### 2.1 Study site description

This study is carried out in the Weierbach catchment, which has been the focus of an increasing number of investigations in the last few years about streamflow generation (Glaser et al., 2016, 2019; Scaini et al., 2017, 2018; Carrer et al., 2019; Rodriguez
and Klaus, 2019), biogeochemistry (Moragues-Quiroga et al., 2017; Schwab et al., 2018), and pedology and geology (Juilleret et al., 2011).

The Weierbach is a forested headwater catchment of 42 ha located in northwestern Luxembourg (Fig. 1). The vegetation consists mostly of deciduous hardwood trees (European beech and Oak), and conifers (*Picea abies* and *Pseudotsuga menziesii*). Short vegetation covers a riparian area that is up to 3 m wide and that surrounds most of the stream. The catchment morphology
is a deep V-shaped valley in a gently sloping plateau. The geology is essentially Devonian slate of the Ardennes massif, phyllite, and quartzite (Juilleret et al., 2011). Pleistocene Periglacial Slope Deposits (PPSD) cover the bedrock and are oriented parallel to the slope (Juilleret et al., 2011). The upper part of the PPSD ($\sim$ 0–50 cm) has higher drainable porosity than the lower part of the PPSD ($\sim$ 50–140 cm) (Martínez-Carreras et al., 2016). Fractured and weathered bedrock lies from $\sim$ 140 cm depth to $\sim$ 5 m depth on average. Below $\sim$ 5 m depth lies the fresh bedrock that can be considered impervious. The climate is temperate
and semi oceanic. The flow regime is governed by the interplay of seasonality between precipitation and evapotranspiration. Precipitation is fairly uniformly distributed over the year, and averages 953 mm/yr over 2006–2014 (Pfister et al., 2017). The runoff coefficient over the same period is 50 %. Streamflow ($Q$) is double-peaked during wetter periods (Martínez-Carreras et al., 2016), and single-peaked during drier periods occurring normally in summer when evapotranspiration ($ET$) is high.

Based on previous modeling (e.g. Fenicia et al., 2014; Glaser et al., 2019) and experimental studies (e.g. Martínez-Carreras
et al., 2016; Juilleret et al., 2016; Scaini et al., 2017; Glaser et al., 2018), Rodriguez and Klaus (2019) proposed a perceptual model of streamflow generation in the Weierbach. In this model, the first and flashy peaks of double-peaked hydrographs are generated by precipitation falling directly into the stream, by saturation excess flow from the near-stream soils, and by infiltration excess overland flow in the riparian area. The second peaks are generated by delayed lateral subsurface flow. The lateral fluxes are assumed higher at the PPSD/bedrock interface due to the hydraulic conductivity contrasts (Glaser et al., 2016,
2019; Loritz et al., 2017). Lateral subsurface flows are thus accelerated when groundwater rises after a rapid vertical infiltration through the soils (Rodriguez and Klaus, 2019). The model based on travel times presented in this study was developed in a step-wise manner based on this hypothesis of streamflow generation, and the consistency between simulated and observed $\delta^2$H points toward a robust representation of the key processes. Water flow paths and streamflow generation processes in this catchment are however not completely resolved. Other studies carried out in the Colpach catchment (containing the Weierbach)
suggested that first peaks are caused by lateral subsurface flow through a highly conductive soil layer and that second peaks are caused by groundwater flow in the bedrock (Angermann et al., 2017; Loritz et al., 2017). This is contrary to the conclusions from other studies in the Weierbach (Glaser et al., 2016, 2020), showing that the key processes are still under debate.

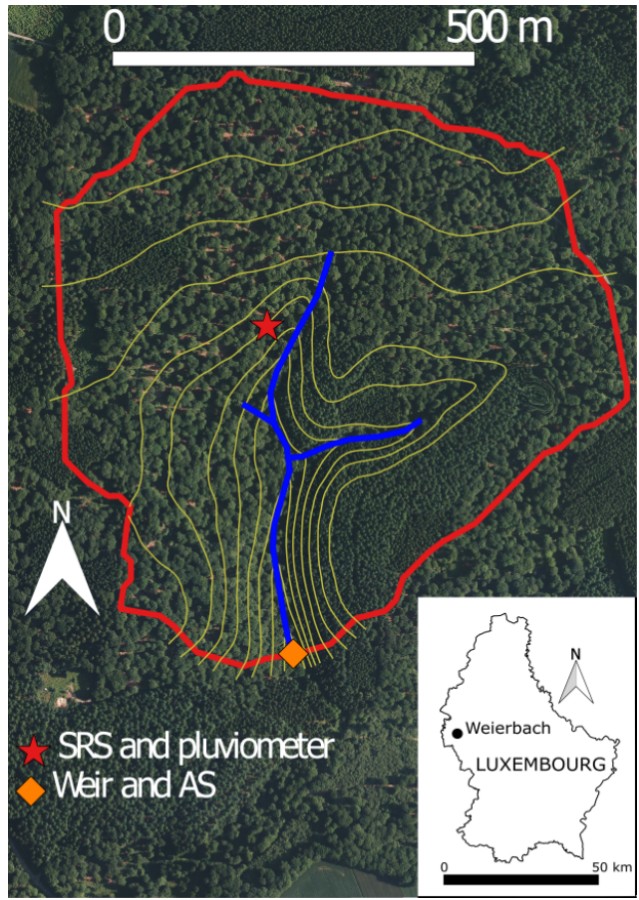

**Figure 1.** Map of the Weierbach catchment and its location in Luxembourg. The weir is located at coordinates (5°47'44" E, 49°49'38" N). SRS is the sequential rainfall sampler. AS is the stream autosampler. The elevation lines increase by 5 m from 460 m.a.s.l. downstream close to the weir location to 510 m.a.s.l. at the northern catchment divide.

## 2.2 Hydrometric and tracer data

In this study we use precipitation ($J$, in mm/h), $ET$ (mm/h), $Q$ (mm/h), and $\delta^2 H$ (‰) and $^3H$ (Tritium Units, T.U.) measurements in precipitation ($C_{P,2}$ and $C_{P,3}$ respectively) and streamflow ($C_{Q,2}$ and $C_{Q,3}$ respectively). Here the subscript 2 indicates deuterium ($^2H$) and the subscript 3 indicates tritium ($^3H$). The analysis in this study focuses on the period October 2015–October 2017 (Fig. 2). Details on the hydrometric data collection ($J$, $ET$, $Q$), and on the $^2H$ sample collection and analysis are given in Rodriguez and Klaus (2019).

The 1088 stream grab samples analyzed for $^2H$ were instantaneous samples collected manually or automatically with an autosampler (AS, Fig. 1), resulting in samples on average every 15 hours over October 2015–October 2017. These stream samples represent most flow conditions in the catchment in terms of frequency of occurrence (Fig. 3). The 525 precipitation samples analyzed for $^2H$ were collected approximately every 2.5 mm rain increment (i.e. on average every 23 hours) with

a sequential rainfall sampler (SRS) and in addition as cumulative bulk samples on an approximately bi-weekly basis (but ranging from 1 to 4 weeks in some occasions). Both the sequential rainfall samples and the cumulative bulk samples represent a precipitation-weighted average $\delta^2$H over different time intervals (approximately daily intervals for sequential rainfall samples and approximately bi-weekly intervals for bulk samples). The samples were analyzed at the Luxembourg Institute of Science and Technology (LIST) using an LGR Isotope Water Analyzer, yielding for $^2$H an analytical accuracy of 0.5 ‰ (equal to the LGR standard accuracy), and a precision maintained <0.5 ‰ (quantified as one standard deviation of the measured samples and standards).

The 24 stream samples analyzed for $^3$H were instantaneous grab samples selected from manual bi-weekly sampling campaigns to cover various flow ranges. The manual selection was not based on flows ranked by exceedance probabilities but rather on the streamflow time series itself. The selected samples represent various hydrological conditions (e.g. beginning of a wet period after a long dry spell, small but flashy streamflow responses), based on data available for this catchment (see Sect. 2 and Rodriguez and Klaus, 2019). The 24 tritium samples cover a wide portion of the flow frequencies (c.f. Fig. 3, all sampled flows conditions occurring more than 90% of the time). This number of $^3$H samples is one of the highest used in travel time studies (c.f., Maloszewski and Zuber, 1993; Uhlenbrook et al., 2002; Stewart et al., 2007; Gallart et al., 2016; Gabrielli et al., 2018; Visser et al., 2019), and it is limited by the analytical costs. The samples were analyzed by the GNS Science Water Dating Laboratory (Lower Hutt, New Zealand), which provides high precision tritium measurements using electrolytic enrichment and liquid scintillation counting (Morgenstern and Taylor, 2009). The precision of the stream samples varies from roughly 0.07 T.U. to roughly 0.3 T.U., but is usually around 0.1 T.U. $^3$H in precipitation was obtained for the Trier station (GNIP station 60 km away from the Weierbach) until 2016 from the WISER database of the International Atomic Energy Agency (IAEA) (IAEA and WMO, 2019; Stumpp et al., 2014). The 2017 values were obtained from the Radiologie group of Bundesanstalt für Gewässerkunde (Schmidt et al., 2020). $^3$H in precipitation before 1978 was calculated by regression with data from Vienna, Austria (Stewart et al., 2017). $^3$H in precipitation obtained from the IAEA corresponds to monthly integrated sampling made with an evaporation-free rain totalizer, as described in the GNIP station operations manual.

Since the stream grab samples were collected over a short time interval (seconds to minutes) using a weir, the associated concentrations $C_{Q,2}(t)$ and $C_{Q,3}(t)$ represent the instantaneous value of the deuterium and tritium concentrations in the stream at time $t$, equivalent to the concept of "flux concentrations" of Kreft and Zuber (1978) and Małoszewski and Zuber (1982). For both $^2$H and $^3$H, the time series of tracer in precipitation was interpolated between two consecutive samples (e.g. A and B) as being equal to the value of the next sample (i.e. B). This was necessary to obtain a continuous tracer input time series (required for Eq. 1 to work). For $^2$H, the signal obtained from cumulative bulk samples was continuous by design and thus used as a baseline representing "steps" of constant $\delta^2$H over two weeks on average. Then, the discontinuous signal with higher frequency variations provided by the sequential rainfall samples was inserted into the continuous baseline for the periods when sequential rainfall samples were available (this higher frequency signal is not continuous because of periods of absence of samples when the SRS failed). Therefore, in this study, $C_{P,2}(t)$ represents the instantaneous value of $\delta^2$H in precipitation at time $t$, equal to the precipitation-weighted average value over varying time intervals. Also, $C_{P,3}(t)$ represents the instantaneous value of $^3$H in precipitation at time $t$, equal to the precipitation-weighted average value over monthly intervals. Assuming uniform

precipitation over the catchment, $C_{P,2}(t)$ and $C_{P,3}(t)$ are also equivalent to "flux concentrations" (Małoszewski and Zuber, 1982). Since no measurements of $J$, $Q$, $ET$, and $C_{P,2}$ are available before 2010, we looped back their values of the period

October 2010–October 2015 periodically before 2010 as a best estimate of their past values (Fig. S16 and S17). We aggregated the input data ($J$, $ET$, $Q$, $C_{P,2}$, $C_{P,3}$) to a resolution $\Delta t = 4$ hours, which is small enough to capture the variability of flows and tracers in the input and simulate the variability of the flows and tracers in the output.

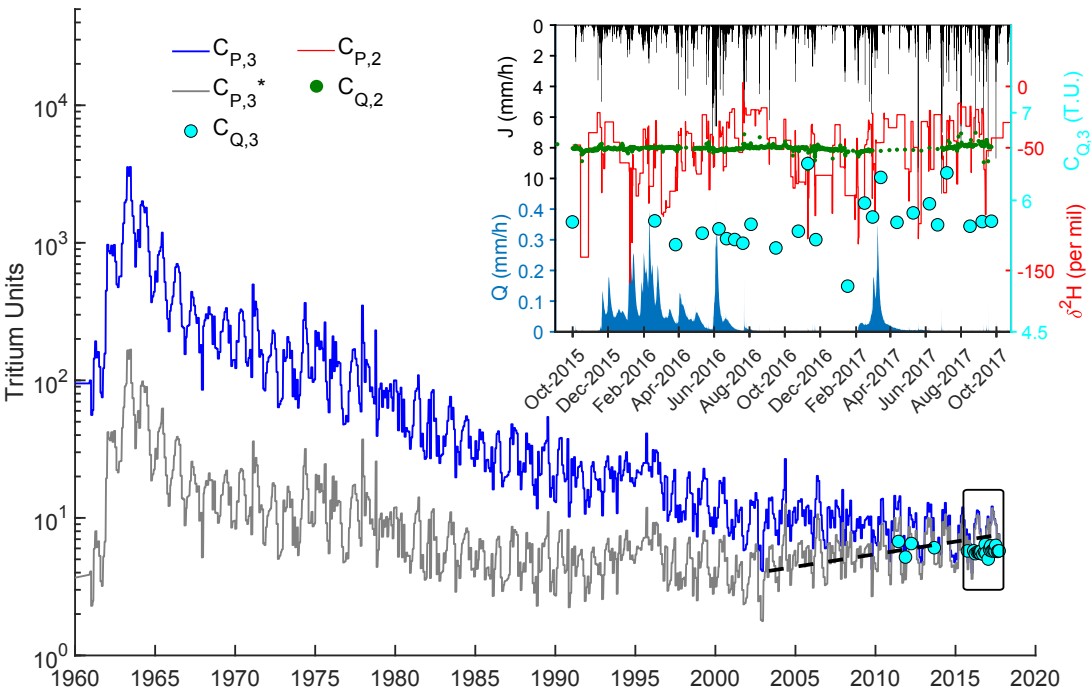

**Figure 2.** Data used in this study: $^3$H in precipitation ($C_{P,3}$), the corresponding tritium activities accounting for radioactive decay until 2017 ($C_{P,3}^*$), $\delta^2$H in precipitation $C_{P,2}$ (inset), precipitation $J$ (inset), streamflow $Q$ (inset), $^3$H measurements in the stream ($C_{Q,3}$ both plots), and $\delta^2$H in the stream ($C_{Q,2}$, inset). The period contained in the inset is represented as a rectangle in the bigger plot. The dashed line visually represents the increasing trend in $C_{P,3}^*$ that emerges as the effect of bomb peak tritium disappears (i.e. $C_{P,3}(t-T)$ stops decreasing around 2000 so $C_{P,3}^*(T,t) = C_{P,3}(t-T)\,e^{-\alpha T}$ starts decreasing with increasing $T$).

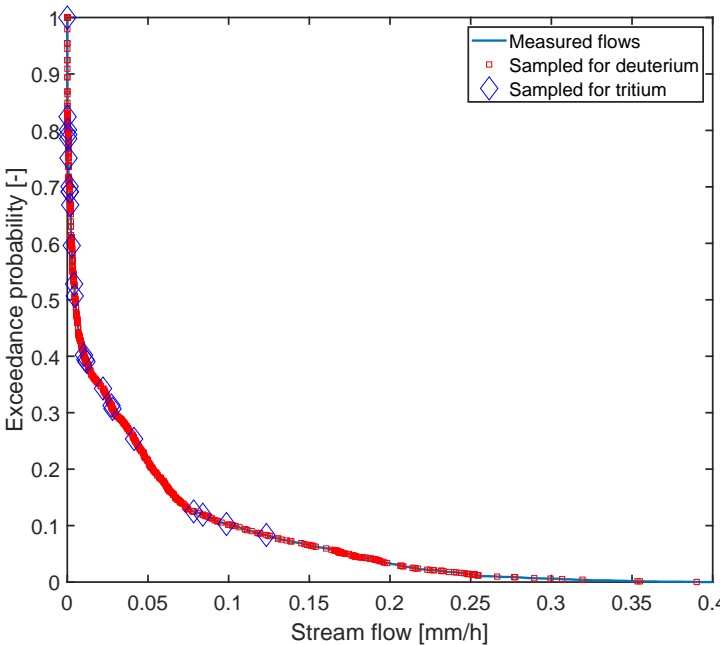

**Figure 3.** Distribution of stream samples ($^{3}$H and $\delta^{2}$H) along the flow exceedance probability curve defined as the fraction of stream flows exceeding a given value over 2015–2017.

## 2.3 Mathematical framework

Mathematically, the streamflow TTD is related to the stream tracer concentrations $C_Q(t)$ according to the following Eq. (1):

$$C_Q(t) = \int_{T=0}^{+\infty} C_P^*(T,t)\,\overleftarrow{p_Q}(T,t)\,dT \tag{1}$$

where $T$ is travel time (the age of water at the outlet), $t$ is time of observation, $C_Q(t)$ is the stream tracer concentration, $\overleftarrow{p_Q}$ (probability distribution function, p.d.f.) is the stream backward TTD (Benettin et al., 2015b), and $C_P^*(T,t)$ is the tracer concentration of the water parcel reaching the outlet at time $t$ with travel time $T$ (this parcel was in the inflow at time $t - T$). The concentrations in this equation need to be flux concentrations, i.e. representative of the tracer mass fluxes in inflows and outflows (see Sect. 2.2). This equation is always true for the exact (usually unknown) TTD, because it simply expresses the fact that the stream concentration is the volume-weighted arithmetic mean of the concentrations of the water parcels with different travel times at the outlet (the weighting of tracer concentrations by hydrological fluxes is thus implicit in $\overleftarrow{p_Q}(T,t)$). Thus, contrary to most of the past travel time studies using a steady-state version of equation 1, no weighting of the concentrations by fluxes is necessary because the time-varying TTD $\overleftarrow{p_Q}(T,t)$ already accounts for the time-varying fractions of precipitation not reaching the stream (due to either $ET$ or storage, see Sect. 2.4) and for time-varying streamflow rates. $C_P^*(T,t)$ depends on $T$

and $t$ as separate variables if the tracer concentration of a water parcel in the catchment changes between injection time $t-T$ and observation time $t$. For solutes like silicon and sodium, the concentration can increase with travel time (Benettin et al., 2015a). For $^3$H, radioactive decay with a constant $\alpha = 0.0563$ yr$^{-1}$ implies $C_{P,3}^*(T,t) = C_{P,3}(t-T)\,e^{-\alpha T}$, where $C_{P,3}(t-T)$ is the concentration in precipitation measured at $t-T$. For $^2$H, $C_{P,2}^*(T,t) = C_{P,2}(t-T)$. Thus, the streamflow TTD simultaneously verifies Eq. (2) and (3):

$$C_{Q,2}(t) = \int\limits_{T=0}^{+\infty} C_{P,2}^*(T,t)\,\overleftarrow{p_Q}(T,t)\,dT = \int\limits_{T=0}^{+\infty} C_{P,2}(t-T)\,\overleftarrow{p_Q}(T,t)\,dT \tag{2}$$

$$C_{Q,3}(t) = \int\limits_{T=0}^{+\infty} C_{P,3}^*(T,t)\,\overleftarrow{p_Q}(T,t)\,dT = \int\limits_{T=0}^{+\infty} C_{P,3}(t-T)\,e^{-\alpha T}\,\overleftarrow{p_Q}(T,t)\,dT \tag{3}$$

Practically, when measurements of $^2$H and $^3$H are used to inversely deduce the TTD by using Eq. 2 and 3, different TTDs may be found. These different TTDs may be called $\overleftarrow{p_{Q,2}}$ and $\overleftarrow{p_{Q,3}}$ for instance, referring to $^2$H and $^3$H, respectively. To avoid introducing more variables and to avoid confusion, we do not use the names $\overleftarrow{p_{Q,2}}$ and $\overleftarrow{p_{Q,3}}$ and we instead refer to the TTDs "constrained" by a given tracer, using a common symbol $\overleftarrow{p_Q}$. We do this also to stress that the exact (true) TTD must simultaneously verify both Eq. (2) and (3), and that two different TTDs $\overleftarrow{p_{Q,2}}$ and $\overleftarrow{p_{Q,3}}$ cannot physically exist. This is a fundamental difference from previous work that assumed two different TTDs, using for example Eq. (3) for $^3$H and another method for $^2$H (the sine-wave approach) (e.g. Małoszewski et al., 1983). The framework in this study also uses the fact that the same functional form of streamflow TTD needs to simultaneously explain both tracers to be valid, unlike previous work that used different TTD models for different tracers (Stewart and Thomas, 2008).

## 2.4 Transport model based on TTDs

Most of the previous travel time studies using tritium assumed steady-state flow conditions, an analytical shape for the streamflow TTD, and fitted the parameters of the analytical function using the framework described in 2.3. In this study, the TTDs are unsteady (i.e. time-varying or transient) and cannot be analytically described. Still, they can be calculated by numerically solving the "Master Equation" (Botter et al., 2011). This method has been applied in several recent studies (e.g. van der Velde et al., 2015; Harman, 2015; Benettin et al., 2017b), and is described in more details by Benettin and Bertuzzo (2018). The numerical method used to solve this equation in this study is described by Rodriguez and Klaus (2019). The biggest difference compared to many previous travel time studies is that time-varying TTDs can be obtained from the Master Equation without using tracer information (i.e., Equations 2 and 3). In this case, tracer equations 2 and 3 simply become a constraint on the solutions found by solving the Master Equation.

Essentially, the Master Equation is a water balance equation where storage and fluxes are labeled with age categories. The Master Equation is thus a partial differential equation. It expresses the fact that the amount of water in storage with a given residence time changes with calendar time. This change is due to new water introduced by precipitation $J(t)$, to water aging, and to losses to catchment outflows $ET(t)$ and $Q(t)$. Solving the Master Equation requires knowledge (or an assumption about

the shape) of the StorAge Selection (SAS) functions $\Omega_Q$ and $\Omega_{ET}$ of outflows $Q$ and $ET$, which conceptually represent how likely water ages in storage (residence times) are to be present in the outflows at a given time. Solving the Master Equation yields the distribution of residence times in storage at every moment, that can be represented in a cumulative form with age-ranked storage $S_T$, defined as the amount of water in storage (e.g. 10 mm) younger than $T$ (e.g. 1 year) at time t. $T \rightarrow S_T$ is just a mathematical change of variable, and it has no meaning respective to the location or depth of a water parcel with a certain residence time in the catchment. By definition $\lim_{T \rightarrow +\infty} S_T = S(t)$, where $S(t)$ is catchment storage. $\Omega_Q$ and $\Omega_{ET}$ are functions of $S_T$ and cumulative distributions functions (c.d.f.) for numerical convenience. SAS functions are closely linked to TTDs, such that one can be found from the other using the following expression (here for $Q$, but valid for other outflows):

$$\overleftarrow{p_Q}(T,t) = \frac{\partial}{\partial T}(\Omega_Q(S_T,t)) \tag{4}$$

The partial derivative with respect to travel time $T$ ensures the transition from c.d.f. to p.d.f. Assuming a parameterized form for $\Omega_Q$ and $\Omega_{ET}$ and calibrating their parameters using the framework defined in 2.3 yields time-varying TTDs constrained by the tracers in the outflows. In this study, the parameters of $\Omega_Q$ are directly calibrated by using Eq. 1 for $C_Q$. Since no tracer data $C_{ET}$ is available, the parameters of $\Omega_{ET}$ are indirectly deduced from Eq. 1 using the tracer measurements in streamflow only. This is made possible by the indirect influence of $\Omega_{ET}$ on the tracer partitioning between $Q$ and $ET$ and on the tracer mass balance (App. A2).

We assumed that $\Omega_{ET}$ is a function of only $S_T$ and it is gamma distributed with a mean parameter $\mu_{ET}$ (mm) and a scale parameter $\theta_{ET}$ (mm). Rodriguez and Klaus (2019) showed that in the Weierbach, a weighted sum of three components in the streamflow SAS function is more consistent with the superposition of streamflow generation processes (i.e. saturation excess flow, saturation overland flow, lateral subsurface flow, see Sect. 2.1) than a single component. This means that $\Omega_Q$ is written as a weighted sum of three c.d.f.s (see appendix A1) (Rodriguez and Klaus, 2019):

$$\Omega_Q(S_T,t) = \lambda_1(t)\,\Omega_1(S_T) + \lambda_2(t)\,\Omega_2(S_T) + \lambda_3(t)\,\Omega_3(S_T) \tag{5}$$

$\lambda_1(t)$, $\lambda_2(t)$, and $\lambda_3(t)$ are time-varying weights summing to 1. $\lambda_1(t)$ is parameterized to sharply increase during flashy streamflow events, using parameters $\lambda_1^*$, $f_0$, $S_{th}$ (mm), and $\Delta S_{th}$ (mm) (App. A1). $\lambda_2(t) = \lambda_2$ is calibrated, and $\lambda_3(t)$ just deduced by difference. $\Omega_1$ is a cumulative uniform distribution over $S_T$ in $[0, S_u]$ (with $S_u$ a parameter in mm). $\Omega_1$ represents the young water contributions associated with short flow paths during flashy streamflow events. We chose rather low values of $\lambda_1^*$ (see Table 1) such that $\lambda_1(t)$ is generally the smallest weight (because $\lambda_1(t) \leq \lambda_1^*$). The lower values of $\lambda_1(t)$ compared to other weights are consistent with tracer data suggesting limited contributions of event water to streamflow (Martínez-Carreras et al., 2015; Wrede et al., 2015). $\Omega_1$ corresponds to processes in the near stream area: saturation excess flow, saturation overland flow, and rain on the stream (Rodriguez and Klaus, 2019). $\Omega_2$ and $\Omega_3$ are gamma-distributed with mean parameters $\mu_1$ and $\mu_2$ (mm), and scale parameters $\theta_1$ and $\theta_2$ (mm) respectively. $\Omega_2$ and $\Omega_3$ represent older water that is always contributing to the stream. This older water consists of groundwater stored in the weathered bedrock that flows laterally in the subsurface.

Note that we used the same functional form of $\Omega_Q(S_T, t)$ for $^2$H and $^3$H to keep the functional form of the TTDs consistent between the tracers. Although composite SAS functions may considerably increase the complexity of the model compared to "traditional" SAS functions, they are necessary to account for different streamflow generation processes (Rodriguez and Klaus, 2019). These processes are potentially associated with contrasting flow path lengths and/or water velocities hence contrasting travel times. The accurate representation of these contrasting travel times is most likely vital for reliable simulations of stream chemistry (Rodriguez et al., 2020).

## 2.5 Model initialization and numerical details

Numerically solving the Master Equation requires an estimation of catchment mobile storage $S(t)$. Here, $S(t)$ represents the sum of "dynamic" (or "active") storage and "inactive" (or "passive") storage (Fenicia et al., 2010; Birkel et al., 2011; Soulsby et al., 2011; Hrachowitz et al., 2013). In this study the model is initialized with storage $S(t=0) = S_{ref} = 2000$ mm. This initial value is chosen large enough to sustain $Q$ and $ET$ during drier periods and to store water that is sufficiently old to satisfy Eq. (1). $S(t)$ is then simply deduced from the water balance as $S(t) = S_{ref} + \int_{x=0}^{t} (J(x) - Q(x) - ET(x))\, dx$. The initial residence time distribution in storage $p_S(T, t)$ is exponential with a mean of 1.7 years, the estimated Mean Residence Time (MRT) by Pfister et al. (2017). Initial conditions need not be specified for the SAS functions, since these are directly calculated from the initial state variables $(S_T(t=0) = S(t=0) \int_{x=0}^{+\infty} p_S(x, t=0)\, dx)$ assuming a parametric form and a set of parameter values. The model is then run with time steps $\Delta t = 4$ hours and age resolution $\Delta T = 8$ hours. This way the computational cost is balanced with the resolution of the simulations in $\delta^2$H. A 100-year spin-up is used to numerically allow the presence of water up to 100 years old in storage and to avoid a numerical truncation of the TTDs. This spin-up is also long enough to completely remove the impact of the initial conditions. This means that $S_{ref}$ and the initial residence time distribution in storage do not influence the results over October 2015–October 2017. $ET(t)$ is taken equal to potential evapotranspiration $PET(t)$ except that it tends non-linearly towards 0 (using a constant smoothing parameter $n$) when storage $S(t)$ decreases below $S_{ref} - S_{root}$ (mm), where $S_{root}$ is a parameter accounting for the water amount accessible by $ET$ (appendix A2).

## 2.6 Model calibration

The parameters of the SAS functions and the other model parameters were calibrated using a Monte Carlo technique. In total, 12 parameters were calibrated (Table 1). The initial ranges were selected based on parameter feasible values (e.g. $f_0$ between 0 and 1 by definition), on previous estimations (e.g. $S_{th}$), on hydrological data (e.g. $S_u$ and $\Delta S_{th}$ deduced from average precipitation depths), and on initial tests on the parameter ranges (e.g. $\mu$ and $\theta$). These ranges allow a wide range of shapes of SAS functions while minimizing numerical errors (occurring for example for $S_T > S(t)$).

Unlike our previous modeling work in this catchment (Rodriguez and Klaus, 2019), we fixed the initial storage in the model $S_{ref}$ (to 2000 mm). We did this to reduce the degrees of freedom when sampling the parameter space in order to limit the impact of numerical errors on the calibration. These errors are due to numerical truncation of $\Omega_Q(S_T, t)$ when a considerable part (e.g. a few percent) of its tail extends above $S(t)$. This occurs when parameters $\mu_2$, $\mu_3$, $\theta_2$, and $\theta_3$ are too large compared to $S_{ref}$ when the latter is also randomly sampled. Choosing a constant large value for $S_{ref}$ thus guarantees the absence of

**Table 1.** Model parameters

| Symbol | Type | Unit | Initial range | Description[a] |
|--------|------|------|---------------|----------------|
| $S_{th}$ | Calibrated | mm | $[20, 200]$ | Storage threshold relative to $S_{min}$ separating "dry" and "wet" periods |
| $\Delta S_{th}$ | Calibrated | mm | $[0.1, 20]$ | Threshold in short term storage changes identifying "first" peaks in hydrographs |
| $S_u$ | Calibrated | mm | $[1, 50]$ | Range of the uniformly distributed $\Omega_1$ |
| $f_0$ | Calibrated | – | $[0, 1]$ | Young water coefficient for the dry periods |
| $\lambda_1^*$ | Calibrated | – | $[0, 1]^b$ | Maximum value of the weight $\lambda_1(t)$ |
| $\lambda_2$ | Calibrated | – | $[0, 1]$ | Constant[c] value of the weight $\lambda_2(t)$ |
| $\mu_2$ | Calibrated | mm | $[0, 1600]$ | Mean parameter of the gamma distributed $\Omega_2$ |
| $\theta_2$ | Calibrated | mm | $[0, 100]$ | Scale parameter of the gamma distributed $\Omega_2$ |
| $\mu_3$ | Calibrated | mm | $[0, 1600]$ | Mean parameter of the gamma distributed $\Omega_3$ |
| $\theta_3$ | Calibrated | mm | $[0, 100]$ | Scale parameter of the gamma distributed $\Omega_3$ |
| $\mu_{ET}$ | Calibrated | mm | $[0, 1600]$ | Mean parameter of the gamma distributed $\Omega_{ET}$ |
| $\theta_{ET}$ | Calibrated | mm | $[0, 100]$ | Scale parameter of the gamma distributed $\Omega_{ET}$ |
| $S_{root}$ | Constant | mm | 150 | Water amount accessible by $ET$ |
| $m$ | Constant | – | 1000 | Smoothing parameter for the calculation of $\lambda_1(t)$ |
| $n$ | Constant | – | 20 | Smoothing parameter for the calculation of $ET(t)$ from $PET(t)$ |
| $\Delta t^*$ | Constant | hours | 8 | Width of the moving time window used to calculate short term storage variations $\overline{\Delta S(t)}$ |

[a] Details about the equations involving these parameters are given in appendix A1 and in Rodriguez and Klaus (2019)

[b] $\lambda_1^*$ is in fact uniformly sampled between 0 and $1 - \lambda_2 \leq 1$ to ensure that $\sum_{n=1}^{3} \lambda_k(t) = 1$. This also ensures that values close to 0 are more often sampled than values close to 1 for $\lambda_1^*$.

[c] $\lambda_1(t)$ varies, $\lambda_2$ is constant, and $\lambda_3(t)$ varies and it is deduced using $\lambda_3(t) = 1 - \lambda_2 - \lambda_1(t)$

truncation errors. $S_{ref}$ has little influence on the storage deduced from travel times since the ages sampled from storage by streamflow are governed only by $\mu_2$, $\mu_3$, $\theta_2$, and $\theta_3$. These parameters are independent of $S_{ref}$ as long at it allows sufficiently old water to reside in storage, which is ensured by its large value and by the long spin-up period we used (100 years).

The first step of the Monte Carlo procedure consisted in randomly sampling parameters from the uniform prior distributions with ranges defined in Table 1. 12,096 sets of the 12 calibrated parameters were sampled as a Latin Hypercube (LHS, Helton and Davis, 2003). This sampling technique has the advantages of a stratified sampling technique and the simplicity and objectivity of a purely random sampling technique (Helton and Davis, 2003). It was chosen to make sure that the parameter samples are as evenly distributed as possible despite their relatively small number with respect to the high number of dimensions (due to computational constraints enhanced by the required long spin-up period). The model was then run over the 100-year spin-up followed by October 2015–October 2017, and its performance was evaluated over October 2015–October 2017. We evaluated

model performance in a multi-objective manner, by using separate objective functions for $^2$H and $^3$H. For deuterium, we used the Nash-Sutcliffe Efficiency (NSE):

$$E_2 = 1 - \frac{\sum_{k=1}^{N_2}(C_{Q,2}(t_k) - \delta^2 H(t_k))^2}{\sum_{k=1}^{N_2}(\delta^2 H(t_k) - \overline{\delta^2 H})^2} \tag{6}$$

where $N_2 = 1,016$ is the number of deuterium observations in the stream. For tritium, we used the Mean Absolute Error:

$$E_3 = \sum_{j=1}^{N_3} |C_{Q,3}(t_j) - {}^3 H(t_j)| \tag{7}$$

where $N_3 = 24$ is the number of tritium observations in the stream. We used the MAE for tritium because it is common to report errors in T.U., and because of the limited variance of stream $^3$H (due to the limited number of samples and the low variability) making the NSE less appropriate (Gallart et al., 2016). The behavioral parameter sets that are used for uncertainty calculations and further analysis were selected based on threshold values $L_2$ and $L_3$ for the performance measures $E_2$ and $E_3$ respectively (Beven and Binley, 1992). Parameter sets were considered behavioral for deuterium simulations if $E_2 > L_2 = 0$, and behavioral for tritium simulations if $E_3 < L_3 = 0.5$ T.U. We subsequently refer to these parameter sets and corresponding simulations as "constrained by deuterium", "constrained by tritium", and as "constrained by both" when both performance criteria were used. We chose these constraints to get reasonable model fits to the data, to obtain a comparable number of behavioral parameter sets for $^2$H and $^3$H, and to maximize the amount of information gained about the parameters when adding a constraint on the model performance for a tracer. This information gain was assessed with the Kullback-Leibler Divergence $D_{KL}$ between the parameter distributions inferred from various combinations of constraints $L_2$ and $L_3$ (Sect. 2.7).

## 2.7 Information contents of $^2$H and $^3$H

Loritz et al. (2018, 2019) recently used information theory to detect hydrological similarity between hillslopes of the Colpach catchment, and to compare topographic indexes in the Attert catchment in Luxembourg. Thiesen et al. (2019) used information theory to build an efficient predictor of rainfall-runoff events. In this study we leverage information theory to evaluate our model parameter uncertainty (Beven and Binley, 1992), and to assess the added value of $\delta^2$H and $^3$H tracers for information gains on travel times. First, we calculated the expected information content of the prior and posterior parameter distributions using the Shannon entropy $\mathcal{H}$:

$$\mathcal{H}(X|^i H) = -\sum_{k=1}^{n_I} f(I_k) \log_2 f(I_k) \tag{8}$$

In this equation, the parameter $X$ (e.g. $\mu_1$) takes values (e.g. 125 mm) falling in intervals $I_k$ (e.g. $[100, 150]$ mm) that do not intersect each other and which union $\cup_{k=1}^{n_I} I_k$ equals $I_X$, the total interval of values on which $X$ is defined (e.g. $[50, 500]$ mm). The definitions of the $n_I$ intervals $I_k$ for each parameter depend on the binning of the parameter values, given in Table

2. The distribution $f$ defines the probability of the parameter $X$ to be in a certain state (i.e. to take a value falling in an interval $I_k$), when constrained by the criterion $E_2 > L_2 (i = 2)$ or $E_3 < L_3 (i = 3)$ (posterior distribution) or none of those (prior distribution). $f$ can also be calculated for a combination of these criteria ($\mathcal{H}(X|(^2H \cap {}^3H))$). When using the logarithm of base 2, $\mathcal{H}$ is expressed in bits of information contained in the distribution $f$. The uniform distribution over $I_X$ has the maximum possible entropy. Lower values of $\mathcal{H}$ thus indicate that the distribution is not flat, hence less uncertain than the uniform prior distribution. In general, lower values of $\mathcal{H}$ indicate less uncertain parameters. Lower values of $\mathcal{H}$ for the posteriors also indicate that information on travel times was extracted from the tracer time series. We used the Kullback-Leibler Divergence $D_{KL}$ to precisely evaluate the information gain from prior to posterior distributions:

$$D_{KL}(X|^iH, X) = \sum_{k=1}^{n_I} f(I_k) \log_2 \frac{f(I_k)}{g(I_k)} \tag{9}$$

where $f$ is the posterior distribution constrained by $E_2 > L_2$ and/or $E_3 < L_3$, and $g$ is the prior distribution. $D_{KL}$ is expressed in bits of information gained when the knowledge about a parameter distribution is updated by using tracer data. Summing the $D_{KL}(X|^iH, X)$ for all the parameters and for a given tracer ($i = 2$ or $i = 3$) yields the total amount of information learned on travel times from that tracer. We also used the Kullback-Leibler Divergence $D_{KL}$ to evaluate the gain of information when $^3H$ is used in addition to $^2H$ to constrain model predictions or vice versa:

$$D_{KL}(X|(^2H \cap {}^3H), X|^iH) = \sum_{k=1}^{n_I} f(I_k) \log_2 \frac{f(I_k)}{g(I_k)} \tag{10}$$

where $f$ is the posterior distribution constrained by $E_2 > L_2$ and $E_3 < L_3$, and $g$ is the posterior distribution constrained only by $E_2 > L_2 (i = 2)$ or only by $E_3 < L_3 (i = 3)$. $D_{KL}$ is expressed in bits of information gained when the knowledge about a parameter posterior distribution is updated by adding another tracer. Calculating $D_{KL}$ also requires binning the parameter values to define the intervals $I_k$ and calculate the distributions $f$ and $g$. The binning for each parameter (Table 2) was chosen such that the resulting histograms visually reveal the underlying structure of the parameter values, while avoiding uneven features and irregularities (e.g. very spiky histograms).

## 3 Results

### 3.1 Calibration results

148 parameter sets were behavioral for deuterium simulations, with $E_2$ ranging from $L_2 = 0$ to 0.24. 181 parameter sets were behavioral for tritium simulations, with $E_3$ ranging from 0.24 T.U. to $L_3 = 0.5$ T.U. Additionally, 16 parameter sets were behavioral for both tritium and deuterium simulations, with $E_2$ ranging from $L_2 = 0$ to 0.19 and $E_3$ ranging from 0.36 T.U. to $L_3 = 0.5$ T.U. These solutions show that a reasonable agreement between the model fit to $^2H$ and the model fit to $^3H$ can be found.

The behavioral posterior parameter distributions constrained by deuterium or tritium or by both generally had similar ranges than their prior distributions, except notably for $\mu_2$, $\theta_2$, $\mu_3$, and $\theta_3$ (Table 2). To assess the reduction of parameter uncertainty, we calculated and compared the entropy of the prior and of the posterior distributions (Table 2). A visual inspection of the posterior distributions was also made. Here, we show only the parameters $\mu_2$, $\theta_2$, $\mu_3$, and $\theta_3$ (Fig. 4) that directly control the range of longer travel times in streamflow, since they act mostly on the right-hand tail of the gamma components in $\Omega_Q$. These

parameters thus have a direct influence on the catchment storage inferred via age-ranked storage $S_T$. The distributions of $\mu_2$, $\theta_2$, $\mu_3$, and $\theta_3$ are clearly not uniform. The distributions of the other parameters are provided as a supplement (Fig. S12-S13). Most distributions are not uniform, indicating that the parameters are identifiable.

      Essentially, the results (Table 2 and Fig. 4) reveal that the parameter ranges decreased by adding information on [2]H or [3]H or both. This effect is particularly noticeable for $f_0$ and $\lambda_1^*$, which saw their upper boundary decrease, and for $\mu_2$ and $\mu_3$,

which saw their lower boundary increase considerably. These results also show that the posterior distributions depart from the uniform prior distributions when considering [2]H alone or [3]H alone (i.e. $\mathcal{H}(X|^{i}H) < \mathcal{H}(X)$ and $D_{KL}(X|^{i}H, X) > 0$ in Table 2). This effect is not very pronounced for most parameters, but clearly visible for $\lambda_1^*$, for $\mu_2$ and $\mu_3$ (e.g. uneven distributions of points in Fig. 4), and for $\mu_{ET}$. The posterior distributions become considerably narrower when both tracers are considered, since $\mathcal{H}(X|(^{2}H \cap {}^{3}H))$ is much lower than $\mathcal{H}(X)$, which is visually represented by the distribution of points

tending to cluster towards a corner in Fig. 4. Generally, more was learned about the likely parameter values by adding a constraint on [2]H simulations after constraining [3]H simulations than the opposite (i.e. generally $D_{KL}(X|(^{2}H\cap{}^{3}H), X|^{3}H) \geq D_{KL}(X|(^{2}H\cap{}^{3}H), X|^{2}H)$). Noticeable exceptions to this are the parameters $\mu_2$, $\theta_2$, and $\theta_3$, which are more related to the longer travel times in streamflow and to catchment storage than the other parameters.

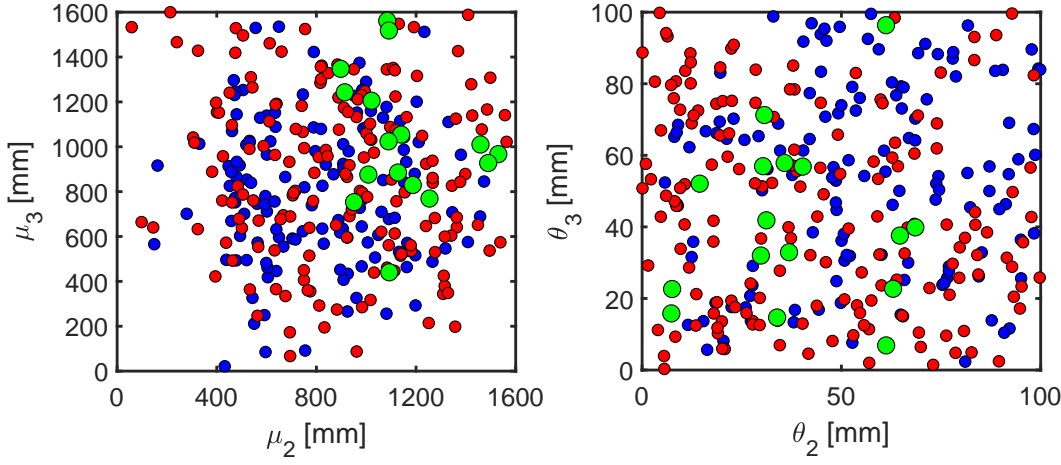

**Figure 4.** Distributions of SAS function mean ($\mu$, left panel) and scale ($\theta$, right panel) behavioral parameters directly controlling the selection of longer travel times by streamflow, constrained by deuterium (148 blue dots), or tritium (181 red dots), or both (16 green dots).

Simulations of stream $\delta^2$H captured both the slow and the fast dynamics of the observations when constrained by $E_2 > 0$
(blue bands and blue curve Fig. 5a), although some variability is not fully reproduced. The Nash Sutcliffe Efficiency ($E_2$)
is limited to 0.24 despite visually satisfying simulations (Sect. 4.4.2). Most flashy responses in $\delta^2$H (associated with flashy
streamflow responses) were reproduced to some extent by the behavioral simulations (the very thin peaks of the blue bands
in Fig. 5a, more visible in Fig. S1–S9). Nevertheless, about 3% of $\delta^2$H data points were visibly underestimated, pointing at
a partial limitation of the composite SAS functions to simulate the variability of the streamflow TTD at these few instances
(c.f. Sect. 4.4.2). Behavioral simulations that were selected using the other performance criterion instead ($E_3 < 0.5$ T.U., red
bands in Fig. 5) did not match well the $\delta^2$H observations. This shows that $^3$H contains some information on travel times that
is not in common with $^2$H. Yet, these behavioral simulations are able to match all observed $\delta^2$H flashy responses in amplitude,
suggesting that like $\delta^2$H, $^3$H contains information on young water contributions to streamflow (Sect. 4.3). Additionally, $\delta^2$H
simulations that were constrained by both criteria (green bands) have a smaller variability than those constrained only by
$E_2 > 0$, suggesting that $^3$H contains some information that is common with $^2$H.

Simulations of stream $^3$H generally matched the observations better in 2017 than before 2017 (red bands and red curve
in Fig. 6). Some simulations (red bands) nevertheless matched the observations before 2017 relatively well. Similar to $\delta^2$H
simulations, both the slow and the fast tracer responses seemed necessary to reproduce the variability in $^3$H observations
(especially in 2017), although additional stream samples would be needed to confirm that the model is accurate between the
current measurement points. The higher stream $^3$H values in 2017 that were better reproduced by the model correspond to an
extended dry period during which streamflow responses were mostly flashy and short-lasting hydrographs. The $^3$H values in
2017 were closer to precipitation $^3$H, mostly around 10 T.U (see also Fig. S15). The stream reaction to those higher values
suggests a considerable influence of recent rainfall events on the stream. Steady-state TTD models relying only on tritium decay
would probably struggle to simulate these fast responses. This also suggests a stronger influence of old water in 2016 than in
2017 (see Sect. 4.4.2). Simulations constrained by deuterium (blue bands) tended to overestimate stream $^3$H. Simulations
constrained by both criteria (green bands) worked well in 2017, but they overestimated stream $^3$H before 2017. Similar to $\delta^2$H
simulations, this suggests that $^2$H and $^3$H have common but also distinct information contents on travel times. The tendency
of the model constrained by deuterium and/or by tritium to overestimate the tritium content in streamflow suggests an non-
negligible influence of the isotopic partitioning of inputs between $Q$ and $ET$ (Sect. 4.4.2, App. A2, and Fig. S15).

## 3.2 Storage and travel time results

For each behavioral parameter set, we calculated $\overleftarrow{P_Q}(T)$, the average streamflow TTD weighted by $Q(t)$ (over 2015–2017) in
cumulative form (Fig. 7). Visually, there are no striking differences between $\overleftarrow{P_Q}(T)$ constrained by deuterium or by tritium,
except a slightly wider spread for simulations constrained by tritium. The $\overleftarrow{P_Q}(T)$ constrained by both tracers clearly differ.
The associated curves (Fig. 7c) show a much narrower spread. The travel time uncertainties are thus visually much lower than
when using each tracer individually, highlighting the benefit of using both tracers together. The $\overleftarrow{P_Q}(T)$ constrained by both
tracers are also slightly shifted towards higher travel times. We calculated various statistics of the $\overleftarrow{P_Q}(T)$ constrained by the

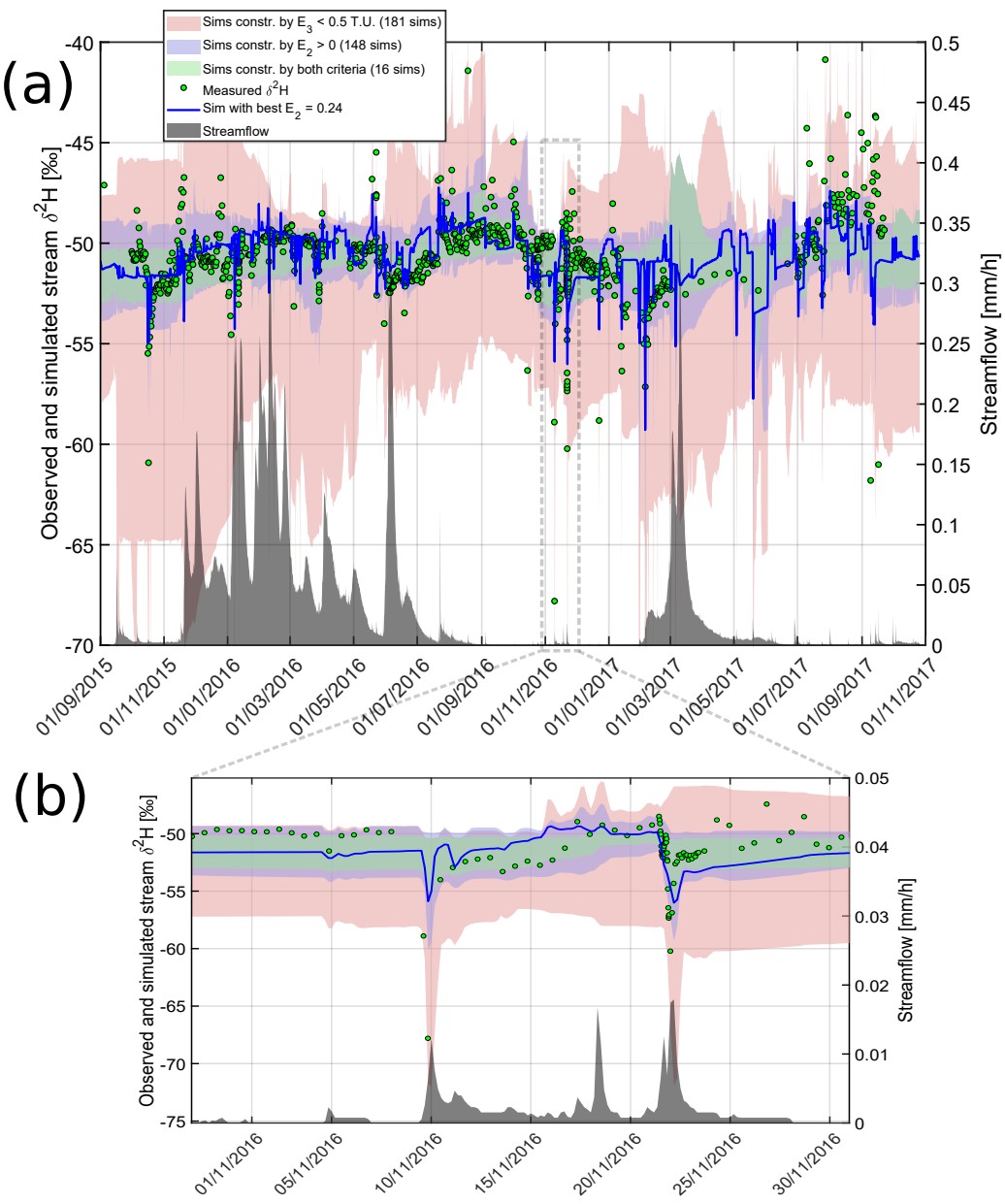

**Figure 5.** Simulations in deuterium. $E_2$ is the Nash-Sutcliffe efficiency in deuterium, and $E_3$ is the Mean Absolute Error in tritium units.

different performance criteria to quantitatively compare the distributions (Table 3). This showed that the $\overleftarrow{P_Q}(T)$ constrained only by tritium systematically correspond to higher travel times (and lower young water fractions) than those constrained only by deuterium. A Wilcoxon rank sum test revealed that some statistically significant differences exist between the $\overleftarrow{P_Q}(T)$

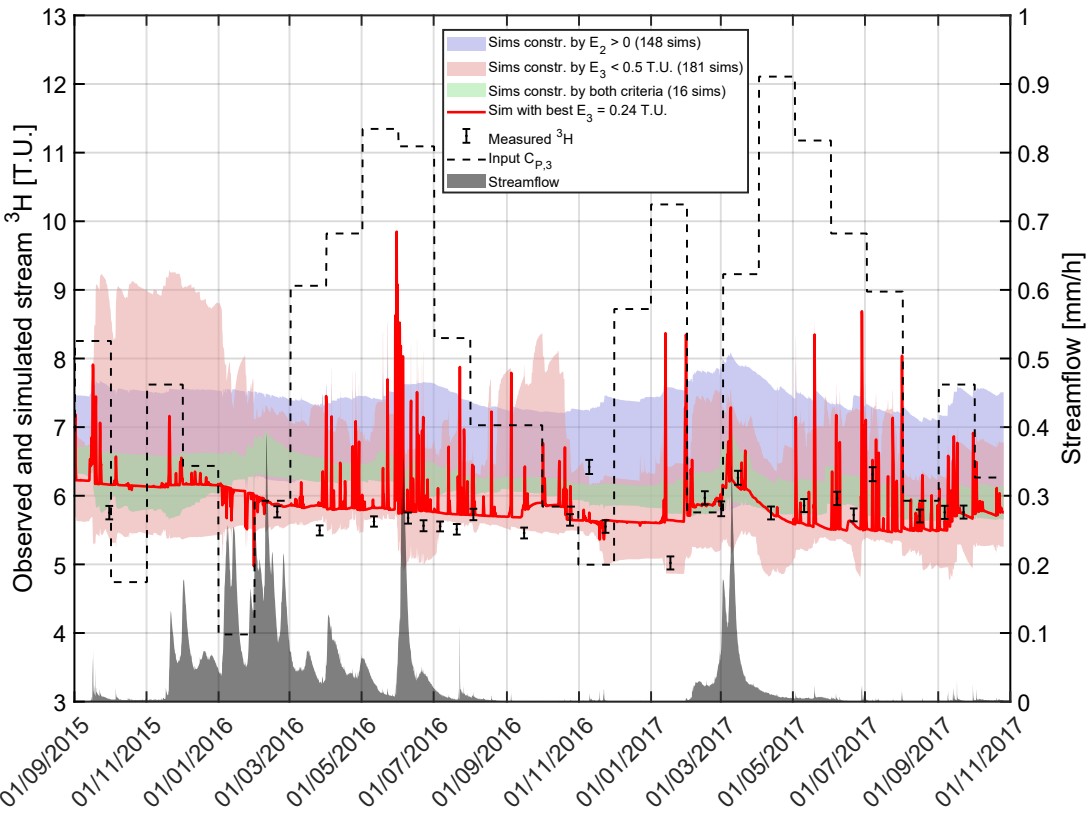

**Figure 6.** Simulations of stream concentrations in tritium compared to observations and to the variability in precipitation.

constrained by deuterium and the $\overleftarrow{P_Q}(T)$ constrained by tritium (App. B). Even if these differences are statistically significant, they remain lower than in previous studies (Sect. 4.1). In addition, the youngest water fractions and the oldest water fractions of $\overleftarrow{P_Q}(T)$ did not significantly differ according to the Wilcoxon rank sum test (App. B).

We defined the right-hand tail of the streamflow SAS function $\Omega_{tail}$ as the weighted sum of the two gamma components in $\Omega_Q$:

$$\Omega_{tail}(S_T) = \frac{1}{\lambda_2 + \lambda_3^*} \left( \lambda_2\,\Omega_2(S_T) + \lambda_3^*\,\Omega_3(S_T) \right) \tag{11}$$

where $\lambda_3^* = 1 - \lambda_2 - \lambda_1^*$. $\Omega_{tail}$ thus allows us to study in detail the asymptotic behavior of the function $\Omega_Q$. In particular, this asymptotic behavior is time-invariant when plotted against $S_T$, because $\Omega_2$ and $\Omega_3$ are functions of $S_T$ only. The behavioral parameter sets were thus directly used to calculate the curves $(S_T, \Omega_{tail}(S_T))$. These curves show similar differences for $^2$H and $^3$H than the curves $(T, \overleftarrow{P_Q}(T))$ (Fig. 8): a slightly wider spread is observed for $\Omega_{tail}$ constrained by tritium than deuterium

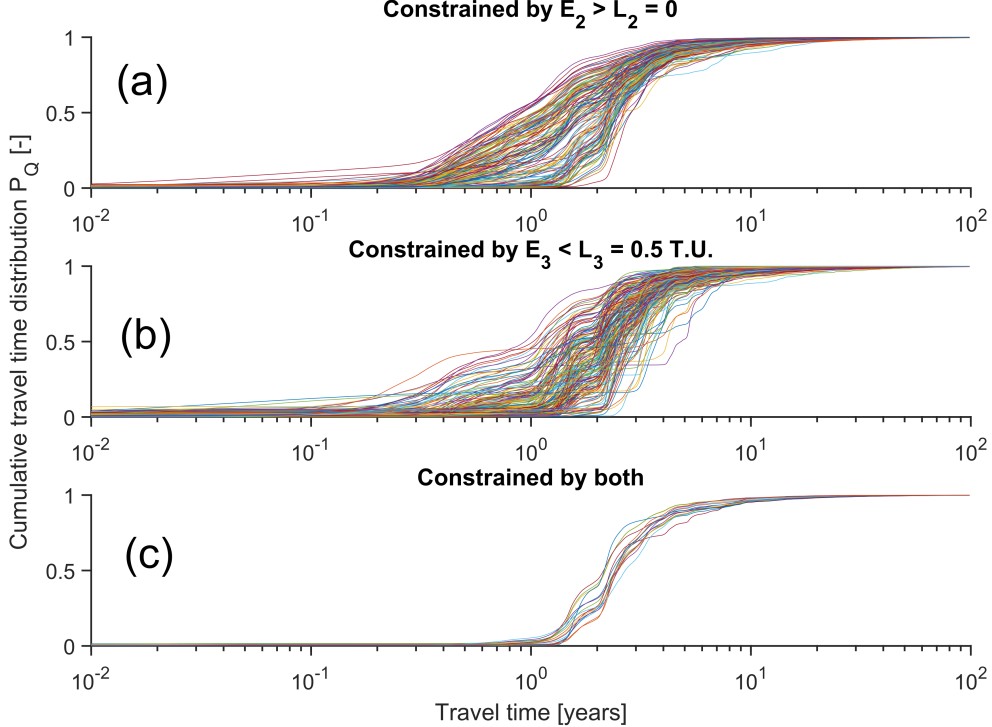

**Figure 7.** Flow weighted (2015–2017) cumulative stream TTDs for the behavioral parameter sets constrained by $^{2}$H (a), by $^{3}$H (b), and by both (c).

(Fig. 8b), and the $\Omega_{tail}$ constrained by both tracers tend to converge to a narrow envelope of curves slightly shifted towards higher storage values (Fig. 8c).

To quantitatively study the implications of different $\Omega_{tail}$ for storage estimations, we computed statistics of a storage measure derived from these curves (Table 4). The 95$^{th}$ percentile of $\Omega_{tail}$, called $S_{95P}$ (black crosses in Fig. 8) allows for estimating total mobile storage $S(t)$ from $\Omega_{tail}$. In average, the $\Omega_{tail}$ constrained by tritium or by both tracers yielded significantly higher mobile storage $S(t)$ and smaller spread in $S(t)$ (Fig. 8, Table 4, Table B1). Yet, the mobile storage $S(t)$ values estimated from the tracers are mutually consistent when considering the uncertainties.

## 4 Discussion

### 4.1 Consistency between TTDs derived from stable and radioactive isotopes of H

Our work shows that streamflow TTDs and the related catchment mobile storage $S(t)$ can be estimated in unsteady conditions by using "ranked" SAS functions $\Omega(S_T, t)$ (Harman, 2015). Similar to Visser et al. (2019), we propose to coherently use the measurements of stream $^{2}$H and $^{3}$H to calibrate the parameters of the SAS functions, here defined in the age-ranked domain

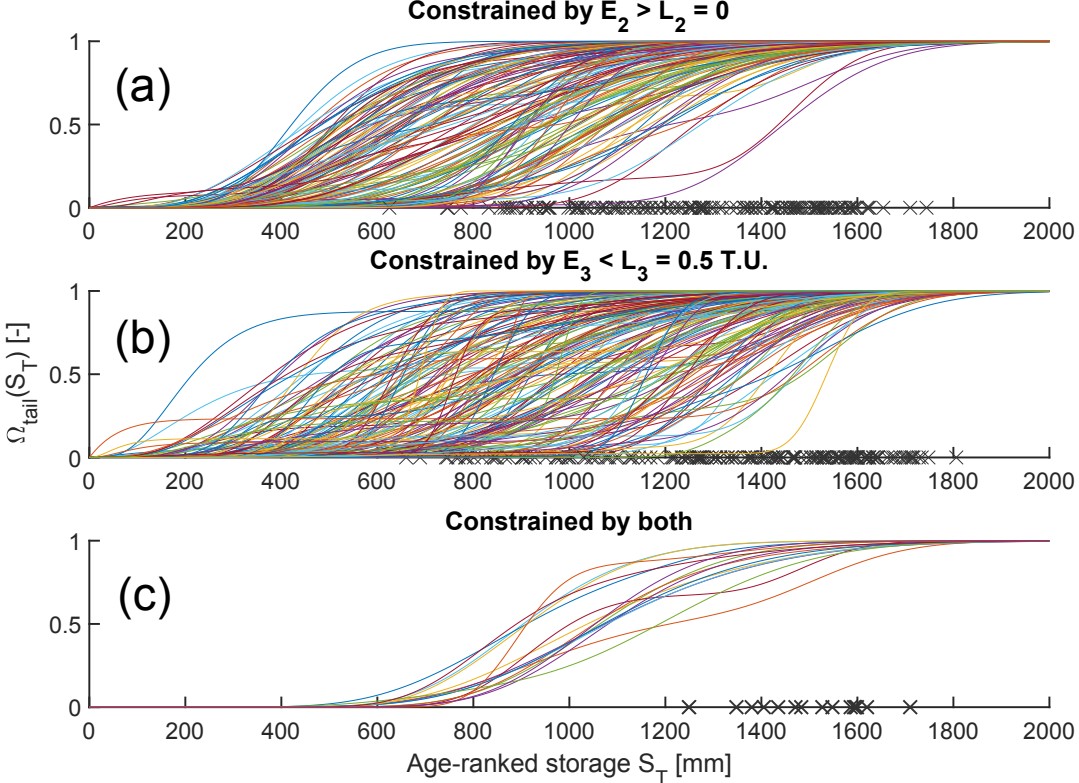

**Figure 8.** Cumulative right-hand tail $\Omega_{tail}$ of streamflow SAS functions for the behavioral parameter sets constrained by $^2$H (a), by $^3$H (b), and by both (c). $\Omega_{tail}$ is defined as the weighted sum of the two gamma components in $\Omega_Q$. The black crosses indicate $S_{95P}$ for each curve, i.e. the 95[th] percentile of $\Omega_{tail}$

$S_T \in [0, +\infty[$ instead of the cumulative residence time domain $P_S \in [0,1]$. The calibrated tail of the streamflow SAS function $\Omega_Q$ (called here $\Omega_{tail}$) could thus be used to approximate mobile storage $S(t)$ instead of defining the value a priori. The SAS functions also allowed us to estimate the unsteady TTDs defined in the travel time domain $T$ and their statistics (e.g.,

mean, median, young water fraction). There were statistically significant differences between some TTD measures (e.g. mean, median) constrained by deuterium or by tritium (Wilcoxon rank sum test, App. B). Yet, the statistical significance may be questioned due to the contrasting number of $^3$H samples (24) compared to $\delta^2$H (> 1000), which is not accounted for in the statistical test. The Wilcoxon rank sum test only compares an equivalent number of accepted simulations (148 for deuterium against 181 for tritium) regardless of data considerations. The TTDs obtained from each tracer were broadly consistent in

shape, and the travel time differences were considerably smaller (i.e., <1 yr) in the Weierbach than in a previous comparison study in four catchments from Germany and New Zealand (up to 5 yr, Stewart et al., 2010). This is particularly true for the MTT (only 8% difference in this study), which was the only travel time measure compared in the previous study (up to 200%

difference in MTT for Stewart et al., 2010). In addition, our travel time differences were smaller for the 75th and 90th percentiles of the TTD than for the 10th and 25th percentiles. The 90th percentile difference was not statistically significant. This is not consistent with previous statements (Stewart and Morgenstern, 2016) that tritium would reveal the long tails of the TTD that remain undetected by stable isotopes. Finally, our travel time differences were smaller than the calculation uncertainties. The storage estimates derived from $^2$H or $^3$H were also statistically different, but the differences were also small compared to the calculation uncertainties.

These results emerged for a number of reasons. First, we treated $^2$H and $^3$H equally by calculating TTDs using a coherent mathematical framework for both tracers (i.e. same method and same functional form of SAS function). However, sampling frequency and model efficiency criteria needed tracer-specific adaptations (see Sect. 4.4.2 and 4.4.3). Second, we did not derive the travel times solely based on the radioactive decay of tritium in order to avoid biases due to mixing at the outlet (Bethke and Johnson, 2008) and in order to avoid the travel time ambiguity caused by tritium from nuclear tests (Stewart et al., 2012). Moreover, we did not use multiple control volumes having different TTDs determined by tracer measurements in their input and output (Małoszewski et al., 1983; Uhlenbrook et al., 2002; Stewart et al., 2007; Stewart and Thomas, 2008). This way, we avoided uncertainties related to difficulties in characterizing end members and gathering representative samples (Delsman et al., 2013). Third, we explicitly accounted for unsteady flow conditions, which has been done in only one previous study using tritium (Visser et al., 2019). This allowed us to estimate realistic average TTDs corresponding to the catchment inflows, outflows, and internal flows that are highly time variant. Fourth, our tritium stream sampling was not focused solely on baseflow hence not biased towards old water. Fifth, we considered the entire TTDs by using various percentiles and statistics, and not only the MTT which is highly influenced by the improbable extreme values of $T$. This means that even if there is water older than e.g. 1,000 years in streamflow, it can be neglected if it represents less than e.g. 0.000001 % of the volume. Finally, we explicitly accounted for parameter uncertainty. This is important because absolute values without an uncertainty estimate are difficult to interpret.

## 4.2 Does tritium help revealing the presence of older water?

$^3$H systematically resulted in higher travel time and storage estimates (tables 3 and 4). Isotopic effects on the transport of water molecules containing deuterium or tritium (i.e., on different isotopologues) seem insufficient to explain these travel time differences, because the self diffusion of these isotopologues in water are nearly equal (Devell, 1962), and their advective velocities are the same. However, flow paths in the relatively small Weierbach catchment are probably too short to allow travel time differences due to isotopic effects on self diffusion coefficients.

It seems likely that the higher storage, the higher travel times, and the larger uncertainties for tritium are related to the lack of high-resolution data. Tritium simulations included many small peaks corresponding to flashy streamflow responses associated with young water (Fig. 6). Only few simulated flashy peaks could be confirmed by the presence of stream $^3$H measurements, especially in 2016. More stream $^3$H samples during flashy $^3$H events may further validate these simulations of young water in streamflow and shift the TTDs constrained by tritium towards younger water. This is consistent with the larger travel time

differences found for the 10[th] and 25[th] percentiles of the TTDs. The limited tritium sampling resolution (bi-weekly) covered most of the flow probabilities (Fig. 3), but it may still be slightly biased towards hydrological recessions during which the youngest water fractions are absent by definition (Sect. 4.4.3). Tritium and stable isotopes of O and H synchronously sampled at high resolution would pave the way for further research on stream water travel times from a multi-tracer perspective.

The travel time and storage measures estimated from a joint use of $^2$H and $^3$H are higher than with individual tracers (tables 3 and 4). These measures are not intermediate values (i.e., the average of the results from the individual tracers) because deuterium and tritium have information in common about longer travel times (i.e., the simulations constrained by both tracers are a specific selection among all accepted simulations, see Fig. 4). Tritium may thus have helped revealing the presence of old water in streamflow. However, it did so only when combined with deuterium. It is commonly assumed that $^3$H carries more
information on old water because of radioactive decay that relates lower tritium activities to increasing travel times (Stewart et al., 2010). However, as shown by Stewart et al. (2012) and in Fig. 2, current tritium values of the water recharged in 1980–2000 are similar to the tritium values of the water recharged today. Thus, the younger water disrupts the relationship between travel times and tritium values. It seems that using the high frequency $\delta^2$H measurements reduced the ambiguity of tritium-derived travel times by helping to discriminate young and old water contributions to streamflow. The travel times being below
∼5 years in the Weierbach (Table 3) could be another reason for the limited information of $^3$H on older water. $^3$H decays by only about 25 % in 5 years, meaning that all the tritium activities of the water in the Weierbach have varied by at most ∼2 T.U. since water entered the catchment. This is much lower than the 10 T.U. amplitude of tritium variations in precipitation. Thus in catchments with relatively short residence times, radioactive decay may give information that is redundant with the natural variability of the tracer in precipitation. In a few decades, water recharged in 1980–2000 may have completely left many
catchments or may be a negligible part of storage, such that the $\log(^3H)$ of stored water may increase linearly with residence time (see the recent increasing trend in $C^*_{P,3}$ in Fig. 2). Thus in a few decades, tritium could be even more informative about old water contributions because there may be no travel time ambiguity anymore. Furthermore, the oscillations of tritium in precipitation over long time scales (>10 years) recently detected and related to cycles of solar magnetic activity (Palcsu et al., 2018) may give stream tritium concentrations even more age-specific meaning. Therefore it is important to re-iterate the call
of Stewart et al. (2012) to start sampling tritium in streams now and for the next decades to use it in travel time analyses.

### 4.3   Travel time information contents of stable and radioactive isotopes

Sampling deuterium and tritium jointly provided substantial additional information besides similar travel time and storage measures derived using each tracer alone. Combining both tracers yielded a non-negligible information gain of ∼10% of the initial $\mathcal{H}(X)$ for most parameters. In total, 12.7 bits of information on travel times were learned by combining the two tracers.
This is more than twice the amount learned from each individual tracer (around 4 bits, see paragraph below). This amount of information can be calculated for a given tracer by summing $D_{KL}(X|(^2H \cap ^3H), X)$ for all parameters (see Sect. 2.7, Table 2). Combining the tracers also resulted in lower uncertainties (lowest entropy $\mathcal{H}(X|(^2H \cap ^3H))$ in Table 2, narrower groups of curves in Fig. 7 and 8, lower standard deviations in tables 3 and 4). This information gain on travel times was possible because

composite SAS functions (Eq. (5)) allowed us to constrain three nearly-independent components ($\Omega_1$, $\Omega_2$, $\Omega_3$) of the same streamflow TTD with one tracer or the other. This reduced the potential trade-offs between the shapes suggested by one tracer or the other. These three components are formally related only by the requirement to have $\lambda_1(t) + \lambda_2(t) + \lambda_3(t) = 1$. Thus all their other parameters are independent.

With deuterium alone, we learned 4.08 bits of information with 1385 samples. With tritium, we learned 4.47 bits of information with only 24 samples. Thus, tritium was overall more informative than deuterium about travel times, even with a lower number of samples. This is because tritium considerably informed us about the travel times in $ET$. Tritium constrained the posterior of $\mu_{ET}$ even better than deuterium (Table 2). The particularly large information gains on $\mu_{ET}$ and $\theta_{ET}$ with tritium reveal a stronger influence of $\Omega_{ET}$ on the accuracy of stream tracer simulations than for deuterium, via an indirect influence on isotopic partitioning (App. A2). This also highlights the importance of explicitly considering $ET$ in streamflow travel time calculations (van der Velde et al., 2015; Visser et al., 2019; Buzacott et al., 2020). However, $^2$H resulted in lower uncertainties for nearly all other parameters (e.g. lower Shannon entropy $\mathcal{H}(X|^2H)$, Table 2). This is most likely due to the much higher sampling frequency for deuterium that allows for constraining the simulations better than with bi-weekly tritium measurements (see the simulation envelopes Fig. 5 and 6). From our experience in the Weierbach catchment, we estimate that for $^2$H, a weekly sampling to cover the damped variations of $\delta^2$H (i.e. about 100 samples over 2015–2017) complemented with an event-based high-frequency sampling (every 15 hours) of the flashy responses (i.e. about 300 samples over 2015–2017) could have given us as much information as the complete time series. This suggests that a more strategic sampling of $^2$H may outperform $^3$H. The amount of information learned from the isotopic data necessarily grows with an increasing number of samples. Yet, we do not know whether it scales linearly or non-linearly and whether it quickly reaches a plateau as the number of observation points grows (Fig. S14). In the future, it would be useful to further use information theory (e.g. entropy conditional on sample size) to know how how this information scales and how many measurements are enough and when to sample isotopes for maximum information gain on travel times. This would imply artificially re-sampling a higher-frequency isotopic time series using various strategies (e.g. Pool et al., 2017; Etter et al., 2018) and re-calibrating the model many times, which would involve much subjectivity and come with an exorbitant computational price. Finally, the information contents on travel times that we have derived depend on our model structure (number of control volumes and SAS functional form). More work is needed in developing "model-free" (e.g., data-based) unsteady TTD estimation methods in order to reduce the dependence of the results on modeling assumptions.

Overall, stable and radioactive isotopes of H had different information contents on travel times. The positive $D_{KL}$ values are not simply due to different performance measures for deuterium and for tritium (c.f. Table S1) but due to non-redundant information contents on travel times for each tracer. Performance measures $E_2$ and $E_3$ are not identical but are both based on minimizing a sum of residuals and thus do not considerably influence what can be learned from tracer data (c.f. Table S2). Moreover, the parameters corresponding to the best simulations in $^2$H did not correspond to those for $^3$H and vice versa. Yet, stable and radioactive isotopes had some information in common on young water. This is consistent with early tritium studies that tried to show its potential for detecting young water contributions to streamflow (Hubert et al., 1969; Crouzet et al.,

1970; Dinçer et al., 1970; Klaus and McDonnell, 2013). This has been overlooked in recent travel time studies because of the sampling focused on periods outside events (Stewart et al., 2010). The theoretical span of 0–4 years pointed out in Stewart et al.

(2010) should however not be taken as the only range of travel times where $^{18}$O, $^2$H, and $^3$H may have redundant information. As clearly written by Stewart et al. (2010), this limit corresponds to a steady-state exponential TTD only, while other TTD shapes (or unsteady TTDs) could yield much higher limits. More importantly, this limit can be lowered by the seasonality of the input function (see Stewart et al., 2010, p. 1647). Finally, stable and radioactive isotopes had some information in common on old water as well. This is clearly shown by the increased travel time and storage measures when both tracers are used, which

also highlights that they can give similar results.

## 4.4 Limitations and way forward

### 4.4.1 Hydrometric- versus tracer-inferred storage

The storage value derived from unsteady travel times constrained by tracer data (Table 4, ∼1200–1700 mm) is noticeably larger than the maximum storage ($\simeq$ 250 mm) estimated from point measurements of porosity and water content (Martínez-

Carreras et al., 2016), from water balance analyses (Pfister et al., 2017), water balance analyses combined with recession techniques (Carrer et al., 2019), and from a distributed hydrological model ($\leq$ 700 mm, Glaser et al., 2016, 2020). Our storage value is more consistent with the ∼1600 mm derived from depth to bedrock and porosity data used for the Colpach catchment (containing the Weierbach) that was modeled with CATFLOW (Loritz et al., 2017). Large differences between hydrometrically-derived and tracer-derived storage estimates are not uncommon (Soulsby et al., 2009; Fenicia et al., 2010; Birkel et al., 2011)

and in fact highlight the ability of tracers to reveal the existence of stored water that is not directly involved in streamflow generation (Dralle et al., 2018; Carrer et al., 2019). This "hydraulically disconnected" storage is nevertheless important to explain the long residence times in catchments (Zuber, 1986). More research is needed for improving the conceptualization of storage and unifying storage terminology and the various estimates obtained from tracers or other techniques. The storage value we found is not in complete contradiction with the previous estimates if we consider their uncertainties. Hydrological

measurements ($J$, $Q$, and especially $ET$) are highly uncertain (Waichler et al., 2005; Graham et al., 2010; Buttafuoco et al., 2010; McMillan et al., 2012; McMahon et al., 2013) and their errors are accumulated in long term water balance calculations. An explicit consideration of those uncertainties in the future could reconcile the different storage estimates. Furthermore, it is worth remembering that simplifying storage from a complex spatially-distributed quantity to a simple compact 1D water column neglects the importance of subsurface heterogeneity, surface topography, and bedrock topography for the storage and

release of water. As a result, upscaling local point measurements of storage capacity that are not representative of the whole subsurface is very likely to under or overestimate the true storage capacity of the whole catchment. This is even more true if the new techniques used to scan the subsurface over larger areas such as Electrical Resistivity Tomography (ERT) are themselves associated with uncertainties, requiring adaptations and site-specific independent knowledge (Parsekian et al., 2015).

### 4.4.2 Model performance and uncertainty

The visually satisfactory tracer simulations enhance our confidence that the model accurately simulates travel times in the Weierbach. Still, the performance in $\delta^2$H or in $^3$H could be improved in the future by testing other models of composite SAS functions. The best NSE for deuterium simulations ($E_2$) was 0.24, which is lower than several other using SAS functions (van der Velde et al., 2015; Harman, 2015; Benettin et al., 2017b). $E_2$ is penalizing for the $\delta^2$H time series in the Weierbach because the observed stream $\delta^2$H has many more points corresponding to damped seasonal fluctuations (Fig. 5a) compared

to the large flashy fluctuations (Fig. 5b). $E_2$ also overemphasizes the timing errors, even if the shape of the simulation is perfect (Klaus and Zehe, 2010; Seibert et al., 2016). In addition, $E_2$ is not an absolute measure of model performance allowing comparisons between different studies (Seibert, 2001; Schaefli and Gupta, 2007; Criss and Winston, 2008). Future work needs to develop more appropriate objective functions for $\delta^2$H, especially with respect to the information gained from model calibration. This implies accounting for expert knowledge, intuition, and visual experience with simulations in a customized

performance measure (Ehret and Zehe, 2011; Seibert et al., 2016), or finding an adequate benchmark model for $\delta^2$H (Schaefli and Gupta, 2007), or correctly defining the statistical properties of the model errors (Schoups and Vrugt, 2010). The best MAE for tritium simulations (called $E_3$) was 0.24. This is slightly higher than values of RMSE (close to 0.10) reported in a number of studies using tritium (Stewart et al., 2007; Stewart and Thomas, 2008; Duvert et al., 2016). However these studies had only a few stream samples, while Gusyev et al. (2013) report for instance a RMSE of 1.62 T.U. for 15 stream samples. Stream $\delta^2$H

seems to suggest larger fraction of young water than the simulations (c.f. underestimation of flashy events in Fig. 5). Stream $^3$H data seems to suggest larger fractions of old water than the simulations (c.f. overestimation of tritium activities over March– September 2016 in Fig. 6). A model passing through all observation points may thus show larger differences between the TTDs constrained by deuterium and the TTDs constrained by tritium. It is important to recall that there are less $^3$H stream samples compared to $^2$H, thus a comparison of the TTDs from this hypothetical ideal model could be misleading. Furthermore, the

different scaling for the units for $\delta^2$H and $^3$H may also mislead the visual comparisons and interpretations on young water contributions based on the different amplitude of flashy tracer responses. We believe that a higher resolution of stream $^3$H would unambiguously show the potential of tritium for revealing young water in the stream, as shown in the early tritium studies (Hubert et al., 1969; Crouzet et al., 1970; Dinçer et al., 1970). Our choice of performance measures ($E_2$=NSE and $E_3$=MAE) and selection criteria ($L_2 = 0$ and $L_3 = 0.5$) resulted in slightly more TTDs constrained by tritium than TTDs constrained by

deuterium (148 curves for $E_2 > 0$ against 181 curves for $E_3 < 0.5$). These numbers are highly sensitive to performance thresholds, and our choices represent the closest match in the number of accepted solutions for each tracer, while considering only meaningful performance criteria variations (i.e., $\geq 0.1$) and acceptable model performance. This guarantees a similar treatment of the two tracers (i.e. it avoids biases in travel times for a given tracer), while accepting only satisfying simulations for both tracers. Future work could assess the sensitivity of travel time differences between tracers for other performance measures and

thresholds, and for contrasting numbers of accepted solutions.

The isotopic simulations were better for decreasing $\delta^2$H than for increasing $\delta^2$H (better simulations of the flashy events in $\delta^2$H pointing downwards, Fig. 5). This is probably because the increases in $\delta^2$H generally correspond to drier periods,

during which $C_{Q,2}$ starts reacting stronger to $C_{P,2}$ indicating that young water fractions (controlled by $\lambda_1(t)$ in the model) are higher than expected. $C_{P,2}$ can explain only about 30% of the variations of $C_{Q,2}$, but this can increase to 44% during drier periods (Fig. S10 and S11). The low explanatory power of $C_{P,2}$ is linked to the larger influence of groundwater for streamflow responses in the Weierbach (conceptualized with $\Omega_2$ and $\Omega_3$ having larger weights $\lambda_2$ and $\lambda_3$). During drier periods, we expect an increase in the non-linearity of the processes delivering young water to the stream. For example, the decreasing extent of the stream network and of saturated areas observed in the Weierbach during drier conditions (Antonelli et al., 2020a, b) is likely caused by decreasing groundwater levels (Glaser et al., 2020) and it could reduce the amounts of young water reaching the stream (c.f. van Meerveld et al., 2019). However, streamflow is lower during drier conditions, thus the fractions of young water can still increase because of a less pronounced dilution of the young water in streamflow compared to wet periods. On the other hand, preferential flow observed in the soils of the Weierbach catchment and in the direct vicinity (Jackisch et al., 2017; Angermann et al., 2017; Scaini et al., 2017, 2018) may become more relevant during drier conditions and could increase the amount of young water contributing to streamflow, especially because precipitation intensities can be much higher in summer (due to thunderstorms) than in winter. The parameterization of streamflow SAS functions via $\lambda_1(t)$ (Eq. (A5)) includes—to some extent—the effect of wet vs. dry conditions and the role of precipitation intensity, but it seems not to fully capture how these factors influence young water fractions in the stream. Testing other parameterizations of $\lambda_1(t)$ or including additional information such as soil moisture or groundwater levels in the current parameterization of $\lambda_1(t)$ may improve the simulations. Finally, the uncertainty of precipitation $\delta^2 H$ could be higher during drier periods, because precipitation amounts can be too small (e.g. < 1 mm) over several weeks or because the precipitation intensities can be too high (e.g. > 5 mm/h) to be captured efficiently by the sequential rainfall sampler. This may lead to inaccuracies in the input data and thus to the inability of the model to simulate the corresponding flashy events in stream $\delta^2 H$. The representation of precipitation $\delta^2 H$ should be improved in the future by using more recent sampling techniques (e.g. Michelsen et al., 2019).

The tendency of the model to yield higher average tritium values than the observations in streamflow over 2015–2017 (Fig. 6) and lower average tritium values than precipitation (see Fig. S15 where this is more visible) seems related to either not enough tritium residing in storage or removed by $ET$. The latter mechanism is only indirectly controlled by $\Omega_{ET}$ which loosely acts on the isotopic partitioning between $Q$ and $ET$ (App. A2). Unfortunately, no tracer data in $ET$ can be used to close the tracer mass balance and to draw firm conclusions on the correct mechanism. In any case, an accumulation of tritium in storage to decrease the average stream tritium content is not a realistic behavior in the long term. The average stream $^3 H$ is higher for the simulations constrained by $E_2 > L_2$ than $E_3 < L_3$ probably because of the lower resolution of $^3 H$ measurements. The simulations overestimated $^3 H$ in the stream particularly in 2015–2016 compared to 2017 (Fig. 6). In 2017 the simulations were better because the model used more young water (<7 days old, using $\Omega_1$) to simulate the variability and the higher values of stream $^3 H$ than in 2016. The lower $^3 H$ in 2015–2016 could be caused by an increased travel time in the older water components in 2015–2016 compared to 2017, due to changes in the importance of different subsurface flow paths in the Weierbach caused by a wetter period. The old water components $\Omega_2$ and $\Omega_3$ (Eq. (5)) represent subsurface flow paths likely occurring in the lower soils and following bedrock topography (Glaser et al., 2016; Rodriguez and Klaus, 2019) and potentially in weathered bedrock fractures (Scaini et al., 2018) or in the bedrock (Angermann et al., 2017; Loritz et al., 2017). We used functions of $S_T$ only for

$\Omega_2$ and $\Omega_3$, meaning that the ranges of ages they select do not change considerably with time (because the distribution of $S_T$ is rather stable). Including an explicit dependence on time for $\Omega_2$ and $\Omega_3$ could help to better represent deeper flow paths in the catchment and improve $^3$H simulations in 2015–2016. Eventually, the monthly resolution of $^3$H in precipitation is coarser than the biweekly sampling in the stream, which can hinder accurate simulations. An increase in sampling resolution of tritium in precipitation will be necessary in the future (Rank and Papesch, 2005).

Finally, parameter distributions (Fig. 4 and S12-S13) and information measures (Table 2) suggest that some parameters are not strongly constrained by tracer data (but they are not unidentifiable either). This may result from the larger number of parameters than traditional SAS functions. Nevertheless, all these parameters are necessary to represent the array of non-linear and time-varying processes leading to the selection of particular ages from storage (numerically represented by $\sim 10^5$ control volumes) to generate both outflows $Q$ and $ET$. This is essential to not neglect certain travel times that may become important for accurate water chemistry simulations (Rodriguez et al., 2020). Other methods to explore parameters (using Markov Chains) such as DREAM (Vrugt, 2016) or PEST (Doherty and Johnston, 2003) could yield narrower posterior distributions. Nevertheless, these more advanced algorithms would need to be adapted to allow parameter constraints, numerically-diverging solutions (typically for randomly selected combinations of parameters values that are incompatible), and multi-objective calibration.

### 4.4.3 Data constraints

The highest flows that were not sampled for tritium (Fig. 3) represent about 50% of the water that left the catchment via streamflow over 2015–2017. The high flows are mostly "second" delayed streamflow peaks in this catchment where double-peaked hydrographs occur in wet conditions (Sect. 2.1). Previous studies in the Weierbach using various tracers suggest that second peaks are likely composed of older water than first peaks (Wrede et al., 2015; Martínez-Carreras et al., 2015). Nevertheless the high flows in the second peaks may be associated with shorter travel times than low flows. Loritz et al. (2017) described the subsurface of the Weierbach catchment as highly permeable and hypothesized that it is able to rapidly transmit large amounts of young water during high streamflow events. This may explain the higher tritium-derived travel times due to the limited $^3$H sampling in this study (e.g., 25% difference in median travel time). For deuterium, the highest flows are associated with 40 samples (about 4% of the samples) which represent about 20% of the water leaving via streamflow over 2015–2017 (Fig. 3). It is important to notice that weighting the available stream samples by streamflow in the calibration (i.e., calibrating on tracer loads instead of concentrations) would not compensate for this relative absence of samples during high flow conditions. In addition, it would bias the calibrated TTDs towards high flow conditions, while our goal is to have TTDs which accurately represent the functioning of the catchment over all flow conditions (the whole 2015–2017 study period). An adaptive sampling frequency based on accumulated flows (e.g., one sample every dozen m$^3$) could improve the representativity of the samples with respect to the flow volumes. This would not improve the results because the TTDs already account for the flow volumes by definition and because the larger water mass not sampled for tritium is not leading to a strong bias towards young or old water compared to deuterium. The latter is shown by the good agreement between the TTDs constrained by deuterium and the TTDs constrained by tritium. Flow-proportional sampling would also lead to a much larger number of samples, rapidly

exceeding the current field and laboratory capacities. This is why nearly-continuous in situ measurements would be preferable (e.g., Pangle et al., 2013; von Freyberg et al., 2017). Nevertheless, in situ measurements are currently not available for tritium.

We found much lower deviations for the travel time and storage measures constrained by deuterium and tritium together (tables 3 and 4). However, it has to be acknowledged that there are only few accepted solutions (16), while there about 10 times more when using $^2$H alone or $^3$H alone. We should expect a higher standard deviation due to a lower number of accepted solutions to calculate this statistic using both tracers. On the contrary, the associated TTDs (Fig. 7c and 8c) fall close to each other, resulting in lower deviations that clearly point to lower uncertainties. A lower number of accepted solutions is in the end inevitable as it is an inherent consequence of using several performance measures independently as opposed to using a combined objective function (e.g. Hrachowitz et al., 2013; Rodriguez et al., 2018). Fewer accepted simulations are also an advantage to identify behavioral parameter sets (Klaus and Zehe, 2010). Less strict threshold criteria for behavioral solutions could increase the number of accepted solutions but they would accept less accurate simulations, which could lead to misleading conclusions. More stream $^3$H measurements would on the other hand allow the use of more advanced objective functions, which could lead to more accepted solutions. The input data measured over 2010–2017 and used to spin up the model from 1960 to 2010 ($J$, $ET$, $Q$, and $C_{P,2}$) could be unrepresentative of the real hydrometeorological and isotopic conditions of 1960–2015 due for instance to nonstationarity or climate change. These changing conditions could affect the modeled residence times in storage and thus the estimated streamflow travel times (Wilusz et al., 2017). Different methods to spin up the model could be tested in the future (Hrachowitz et al., 2011), especially to assess the effect the effect of changing hydrometeorological and isotopic conditions on the estimation of travel times. For this, isotope tracer records that span several decades like the ones that can be reconstructed from pearl mussels shells (Pfister et al., 2018, 2019) represent a crucial asset. Eventually, the precipitation tritium samples were taken about 60 km away from the catchment and may introduce some uncertainty.

## 5 Conclusions

Stable isotopes of O and H and tritium are indispensable tracers to infer the streamflow TTD and derive storage estimates in catchments. Our study addressed an emerging concern about the possible limitations of stable isotopes to infer the whole streamflow TTD compared to tritium. We went beyond previous data and methodological limitations and we did not find that stable isotopes are blind to old water fractions as suggested by earlier travel time studies. We found statistically significant differences between some travel times measures derived from each tracer, but these differences were considerably smaller than in previous studies. The differences we found can most likely be attributed to a higher number of stable isotope samples compared to tritium due to different analysis techniques. Based on the results in our experimental catchment in Luxembourg, we conclude that the perception that stable isotopes systematically truncate the tails of TTDs may not be valid. Instead, our results highlight that stable isotopes and tritium have different information contents on travel times but they can still result in similar TTDs. In fact, inferring the streamflow TTD from a joint use of both tracers better exploits their information contents, which results in lower uncertainties and higher information gains. Although $^3$H appeared to be slightly more informative than $^2$H even with fewer samples, a different sampling strategy of the stable isotopes could outperform tritium. Future work could

additionally compare streamflow TTD and storage from the two tracers in larger catchments where older water is expected,

to give tritium more time to decay and better leverage its ability to point the presence of very old water out. More work is also needed to compare the information contents of the tracers on travel times using data-based approaches in order to avoid a dependence on model structure. We therefore recommend to: (1) keep sampling tritium in as many places as possible, as emphasized by Stewart et al. (2012); but also (2) to sample tritium at the highest frequency possible and synchronously with stable isotopes if possible. This is particularly important for the isotopic measurements in precipitation that drive all model

simulations, regardless of functional forms of TTD and their parameter values. Overall, our work shows that more tracer data is naturally better to gather more information about the catchments functions of storage and release.

*Data availability.* The tritium input data until 2016 used in this study can be obtained from the WISER database portal of the International Atomic Energy Agency (values for 2017 will be accessible there too in the future, please ask Axel Schmidt from Bundesanstalt für Gewässerkunde in the meantime). The rest of the data used in this study is the property of the Luxembourg Institute of Science and

740 Technology (LIST) and can be obtained by request to the corresponding author after approval by LIST.

## Appendix A: Model equations

### A1 Parameterization of the SAS functions

In this section we provide further details on the equations used in the model. The composite streamflow SAS function $\Omega_Q$ used in this study is:

$$745 \quad \Omega_Q(S_T, t) = \lambda_1(t)\,\Omega_1(S_T) + \lambda_2(t)\,\Omega_3(S_T) + \lambda_3(t)\,\Omega_1(S_T) \tag{A1}$$

$\Omega_1(S_T)$ is a cumulative uniform distribution for $S_T$ in $[0, S_u]$, where $S_u$ (mm) is a calibrated parameter representing the amount of stored young water potentially contributing to flashy streamflow responses. Thus:

$$\Omega_1(S_T) = \begin{cases} \frac{S_T}{S(t)}, & S_T \in [0, S_u] \\ 1, & S_T > S_u \end{cases} \tag{A2}$$

$\Omega_2(S_T)$ and $\Omega_3(S_T)$ are direct functions of $S_T$ and are gamma-distributed:

$$750 \quad \Omega_2(S_T) = \frac{1}{\Gamma(\frac{\mu_2}{\theta_2})}\,\gamma(\frac{\mu_2}{\theta_2}, \frac{S_T}{\theta_2}) \tag{A3}$$

$$\Omega_3(S_T) = \frac{1}{\Gamma(\frac{\mu_3}{\theta_3})}\,\gamma(\frac{\mu_3}{\theta_3}, \frac{S_T}{\theta_3}) \tag{A4}$$

where $\Gamma$ is the gamma function, $\gamma$ is the lower incomplete gamma function, $\mu_2$ and $\mu_3$ (mm) are mean parameters (calibrated), and $\theta_2$ and $\theta_3$ (mm) are scale parameters (calibrated).

$\lambda_1(t)$, $\lambda_2(t)$, and $\lambda_3(t)$ sum to 1. These are simply time-varying weights giving each component (i.e. c.d.f. $\Omega$) a dynamic contribution to streamflow generation. In particular, $\lambda_1(t)$ is made highly time-variant to represent the flashy hydrographs that have an on-off type of response to precipitation. $\lambda_2(t)$ is considered constant and calibrated to keep the parameterization parsimonious. $\lambda_3(t) = 1 - \lambda_2 - \lambda_1(t)$ is deduced by difference for parsimony as well. Since $\Omega_1(S_T)$ represents young water contributions and previous studies in the Weierbach showed that event water contributions depend on the catchment wetness and on precipitation intensity (Wrede et al., 2015; Martínez-Carreras et al., 2015), $\lambda_1(t)$ was parameterized using storage $S(t)$ and a proxy storage variations $\overline{\Delta S(t)}$ (see Rodriguez and Klaus (2019) for more details):

$$\lambda_1(t) = \lambda_1^* \left[ f(t) + (1 - f(t))\, g(t) \right] \tag{A5}$$

where $\lambda_1^* \in [0, 1]$ (no units) is a calibrated parameter representing the maximum value of $\lambda_1(t)$, and $f(t) \in [0, 1]$ and $g(t) \in [0, 1]$ are given by:

$$f(t) = f_0 \left( 1 - \tanh \left[ \left( \frac{S(t)}{S_{min} + S_{th}} \right)^m \right] \right) \tag{A6}$$

$$g(t) = 1 - \exp \left( -\frac{\overline{\Delta S(t)}}{\Delta S_{th}} \right) \tag{A7}$$

$f_0 \in [0, 1]$ (no units) is a calibrated parameter guaranteeing a minimum for $\lambda_1(t)$ during dry periods; $S_{min} = \min(S(t))$; and $S_{th}$ (mm, calibrated parameter) is a storage threshold relative to the minimum storage $S_{min}$ separating wet ($S(t) > S_{min} + S_{th}$) from dry periods ($S(t) < S_{min} + S_{th}$). $m = 1000$ is a fixed parameter used to smooth the function $f$ with respect to $S(t)$. $\overline{\Delta S(t)}$ is a proxy of storage variations calculated as a moving average of storage variations over a time window $\Delta t^* = 2\,\Delta t$:

$$\overline{\Delta S(t)} = \max \left( \frac{1}{3} \sum_{j=0}^{2} \Delta S(t - j\Delta t), 0 \right) \tag{A8}$$

with $\Delta S(t) = \Delta t \left( J(t) - Q(t) - ET(t) \right)$. $\overline{\Delta S(t)}$ essentially increases during precipitation events and decreases when $Q(t)$ or $ET(t)$ are high. $\Delta S_{th}$ is a threshold in $\overline{\Delta S(t)}$ above which $g(t)$ tends to 1, allowing $\lambda_1(t)$ to increase and decrease sharply during flashy streamflow events.

## A2 Actual evapotranspiration and tracer partitioning between Q and ET

Actual evapotranspiration $ET(t)$ is calculated from potential evapotranspiration $PET(t)$ using the formula:

$$ET(t) = PET(t) \tanh\left[\left(\frac{S(t)}{S_{root}}\right)^n\right] \tag{A9}$$

where $S_{root} = S_{ref} - 150$ is a fixed parameter (mm) representing the storage threshold $S(t) = S_{root}$ below which $ET(t)$ starts decreasing from $PET(t)$ towards 0. A similar strategy was employed for instance by Fenicia et al. (2016) and Pfister et al. (2017) in the Weierbach and neighboring Luxembourgish catchments. This decrease is smoothed by the fixed coefficient

$n = 20$. $S_{root}$ accounts for the water available for evaporation and plant transpiration until the capillary forces offer too much resistance. This formula thus represents the decrease in water losses to the atmosphere under water limited conditions.

In the model, this equation is the only explicit partitioning condition of the tracer influx $J \times C_P$ between evaporative losses $ET \times C_{ET}$ and streamflow $Q \times C_Q$. An implicit partitioning nevertheless exists for the following reason. The tracer mass balance equation is:

$$\frac{dM}{dt}(t) = J(t)\,C_P(t) - Q(t)\,C_Q(t) - ET(t)\,C_{ET}(t) \tag{A10}$$

where $M(t)$ is the tracer mass in the catchment and $C_P(t)$ is the tracer concentration in precipitation at time $t$. $J(t)\,C_P(t)$ is given by the input data, and $Q(t)\,C_Q(t)$ and $ET(t)\,C_{ET}(t)$ are partly determined by the SAS functions $\Omega_Q$ and $\Omega_{ET}$. For $Q(t)\,C_Q(t)$, $Q(t)$ is measured data, and $C_Q(t)$ is directly related to $\Omega_Q$ through the related TTD $\overleftarrow{p_Q}$ (Eq. 1 and 4). The parameters of $\Omega_Q$ are thus directly determined by the fit of the simulations to observed $C_Q(t)$. Tracer data for $C_{ET}(t)$ is

not available. Thus, the parameters of $\Omega_{ET}$ cannot be directly determined from data similarly to $\Omega_Q$. Still, the parameters of $\Omega_{ET}$ need to yield $C_{ET}$ values which satisfy the tracer mass balance (Eq. A10) in the long term (when $\frac{dM}{dt}(t)$ becomes negligible). If the parameters of $\Omega_{ET}$ do not allow the closure of the tracer mass balance, the simulations in $C_Q(t)$ will be affected and will not match the observations. Therefore, the fit between observed and simulated $C_Q(t)$ can be used also to indirectly deduce the parameters of $\Omega_{ET}$, using the implicit tracer partitioning $\Omega_{ET}$ exerts. This partitioning is only indirect

(or implicit) because there is no one-to-one relationship between $T$ and $C_P^*(T,t)$ (Eq. 1), meaning that age selection patterns expressed by the SAS functions do not uniquely determine the average values of $Q(t)\,C_Q(t)$ and $ET(t)\,C_{ET}(t)$. Tritium was more informative on travel times than deuterium due to its stronger constraint on the parameter values of $\Omega_{ET}$, $\mu_{ET}$ and $\theta_{ET}$. Based on the reasoning above, this is simply due to the fact that the relationship between $T$ and $C_P^*(T,t)$ is clearer for tritium due to its radioactive decay than for deuterium, for which there is essentially no relationship between travel time and tracer

concentrations. In conclusion, information on the parameters of $\Omega_{ET}$ exists in the time series of $C_Q(t)$ and can be extracted by calibrating the model based on SAS functions, particularly from using tritium.

## Appendix B:  Statistical significance of travel time and storage differences

The obtained differences in travel time and storage measures (Tables 3 and 4) were further compared to assess their statistical significance (Table B1). For this, we used a Wilcoxon rank sum test (also known as the Mann-Whitney U-test) for each of the time-averaged (flow-weighted over 2015–2017) statistics (e.g., the $10^{th}$ percentile) of the distributions calculated from $^2$H (148 distributions) or $^3$H (181 distributions) and shown in Fig. 7(a,b) and 8(a,b). This tested the null hypothesis that the two underlying median TTDs or SAS functions obtained from each tracer are equal (i.e., the distribution obtained as the median of all the flow-weighted time-averaged distributions over 2015–2017 corresponding to the behavioral parameter sets for a given tracer). We chose this test because it is non-parametric, and because it allows taking into account the travel time and storage uncertainties by including all the behavioral distributions. All tests were made at the 5% significance level.

The results show significant differences (at the 5% level) between all measures except two. According to the statistical test, the youngest fractions of water (younger than ∼2 months) and the oldest fractions of water ($90^{th}$ percentile, older than about 4 years) are most likely drawn from a common TTD, regardless of the tracer used. Despite significant differences of all other measures, this test suggests that the truncation of the long TTD tail when using only deuterium is not statistically plausible.

*Author contributions.*  LP and JK designed the project and obtained the funding for this study. NR and JK provided the experimental design for the study. NR carried out the field and lab work. NR and JK performed the modeling part. NR, LP, EZ, and JK jointly structured the manuscript, and NR wrote the manuscript with contributions from JK, EZ, and LP

*Competing interests.*  The authors declare no competing interests.

*Code availability.*  The codes implementing the composite SAS-based model for deuterium and tritium simulations can be found on the LIST Gitlab (https://git.list.lu/catchment-eco-hydro/composite_sas_model_2h_3h_weierbach).

*Data availability.*  Most of the data used in this study was uploaded to Zenodo (https://zenodo.org/record/4061554#.X4gz69AzaUk). The rest of the data is the property of Luxembourg Institute of Science and Technology and can be obtained by request to the corresponding author.

*Acknowledgements.*  We thank Uwe Morgenstern from GNS Science and Axel Schmidt from Bundesanstalt für Gewässerkunde (BfG) for providing access to the 2017 precipitation tritium data. We thank Laurent Gourdol for his help with the preparation of the tritium input data, and for useful discussions about estimating TTDs with tritium measurements. We thank Uwe Ehret for providing Matlab scripts to compute

information theory measures ($\mathcal{H}$ and $D_{KL}$). Nicolas Rodriguez, Julian Klaus, and Laurent Pfister (FNR/CORE/C14/SR/8353440/STORE-AGE) acknowledge funding for this study from the Luxembourg National Research Fund (FNR). This study also contributes to and benefited from the "Catchments as Organized Systems" (CAOS FOR 1598, INTER/DFG/14/9476192/CAOS2) research unit funded by DFG, FNR, FWF. We thank Jérôme Juilleret for his help to collect tritium samples in the field. The authors acknowledge support by the state of Baden-Württemberg through bwHPC. We thank Barbara Glaser for her help with the input data (PET).

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

**Table 2.** Parameter ranges and information measures before and after calibration to isotopic data

| Parameter | $S_{th}$ | $\Delta S_{th}$ | $S_u$ | $f_0$ | $\lambda_1^*$ | $\lambda_2$ | $\mu_2$ | $\theta_2$ | $\mu_3$ | $\theta_3$ | $\mu_{ET}$ | $\theta_{ET}$ |
|---|---|---|---|---|---|---|---|---|---|---|---|---|
| Unit | mm | mm | mm | – | – | – | mm | mm | mm | mm | mm | mm |
| Range for $E_2 > L_2$ | [21, 196] | [0.14, 20] | [1.4, 50] | [0, 1] | [0, 0.76] | [0, 1] | [149, 1530] | [6, 100] | [21, 1561] | [2, 100] | [51, 926] | [1, 99] |
| Range for $E_3 < L_3$ | [20, 200] | [0.1, 20] | [1, 50] | [0, 1] | [0, 0.92] | [0, 1] | [58, 1564] | [0, 100] | [67, 1600] | [0, 100] | [0, 959] | [1, 100] |
| Range for $E_2 > L_2$ and $E_3 < L_3$ | [25, 177] | [0.17, 20] | [11, 49] | [0, 0.82] | [0, 0.76] | [0.1, 1] | [897, 1530] | [7, 69] | [440, 1561] | [7, 96] | [51, 120] | [3, 98] |
| Binning[a] | [20:20:200] | [0:2:20] | [0:5:50] | [0:0.1:1] | [0:0.05:1] | [0:0.1:1] | [0:100:1600] | [0:10:100] | [0:100:1600] | [0:10:100] | [0:50:1600] | [0:10:100] |
| $\mathcal{H}(X)$[b] | 3.17 | 3.32 | 3.32 | 3.32 | 3.71 | 3.32 | 4 | 3.32 | 4 | 3.32 | 5 | 3.32 |
| $\mathcal{H}(X|^2H)$ | 3.12 | 3.27 | 3.29 | 3.22 | 2.97 | 3.2 | 3.44 | 3.25 | 3.59 | 3.23 | 2.77 | 3.24 |
| $\mathcal{H}(X|^3H)$ | 3.1 | 3.3 | 3.31 | 3.3 | 3.23 | 3.24 | 3.73 | 3.25 | 3.83 | 3.25 | 1.53 | 3.22 |
| $\mathcal{H}(X|(^2H \cap ^3H))$ | 2.52 | 2.95 | 2.53 | 2.91 | 2.65 | 2.91 | 2.48 | 2.18 | 2.91 | 2.75 | 0.7 | 2.48 |
| $D_{KL}(X|^2H, X)$ | 0.05 | 0.05 | 0.04 | 0.10 | 0.27 | 0.12 | 0.56 | 0.07 | 0.41 | 0.10 | 2.22 | 0.09 |
| $D_{KL}(X|^3H, X)$ | 0.07 | 0.02 | 0.01 | 0.02 | 0.13 | 0.08 | 0.27 | 0.07 | 0.17 | 0.07 | 3.45 | 0.10 |
| $D_{KL}(X|(^2H \cap ^3H), X|^2H)$ | 0.64 | 0.37 | 0.76 | 0.42 | 0.60 | 0.41 | 1.52 | 1.14 | 1.10 | 0.57 | 4.30 | 0.84 |
| $D_{KL}(X|(^2H \cap ^3H), X|^2H)$ | 0.39 | 0.36 | 0.75 | 0.3 | 0.36 | 0.27 | 1.23 | 1.03 | 0.75 | 0.48 | 1.29 | 0.72 |
| $D_{KL}(X|(^2H \cap ^3H), X|^3H)$ | 0.48 | 0.36 | 0.8 | 0.42 | 0.33 | 0.34 | 1.08 | 0.91 | 0.84 | 0.37 | 2.04 | 0.75 |

[a] Binning is indicated as $[a : b : c]$, where $a$ is the left edge of the first bin, $b$ is the bin width, and $c$ is the right edge of the last bin.

For instance, $[7 : 2 : 11]$ indicates data sorted with the two bins $[7, 9]$ and $[9, 11]$

[b] $\mathcal{H}$ and $D_{KL}$ are expressed in bits.

**Table 3.** Statistics of $\overleftarrow{P_Q}(T)$ constrained by deuterium or tritium

| Travel time statistics | $^2$H ($E_2 > 0$) [mean ± std] | $^3$H ($E_3 < 0.5$ T.U.) [mean ± std] | $^3$H–$^2$H differences Absolute difference | $^2$H and $^3$H [mean ± std] |
|---|---|---|---|---|
| 10$^{th}$ percentile [years] | 0.78 ± 0.49 | 1.10 ± 0.57 | 0.32 years | 1.44 ± 0.11 |
| 25$^{th}$ percentile [years] | 1.16 ± 0.56 | 1.54 ± 0.59 | 0.38 years | 1.85 ± 0.22 |
| Median [years] | 1.77 ± 0.55 | 2.19 ± 0.64 | 0.42 years | 2.38 ± 0.15 |
| 75$^{th}$ percentile [years] | 2.78 ± 0.61 | 3.07 ± 0.74 | 0.29 years | 3.26 ± 0.39 |
| 90$^{th}$ percentile [years] | 4.64 ± 1.27 | 4.79 ± 1.41 | 0.15 years | 5.19 ± 0.86 |
| Mean [years] | 2.90 ± 0.54 | 3.12 ± 0.59 | 0.22 years | 3.45 ± 0.28 |
| $F_{yw}$[a] [%] | 1.5 ± 1.6 | 1.8 ± 2.3 | 0.3% | 0.61 ± 0.53 |
| F(T < 6 months) [%] | 10 ± 8.6 | 6.3 ± 8.2 | -3.7% | 0.75 ± 0.58 |
| F(T < 1 year) [%] | 24 ± 17 | 11 ± 12 | -13% | 2.1 ± 1.5 |
| F(T < 3 years) [%] | 77 ± 8.5 | 71 ± 16 | -6% | 70 ± 6.6 |

The mean and standard deviations are calculated from all retained behavioral solutions for a given criterion. [a] Fraction of "young water" (Kirchner, 2016), younger than 0.2 years

**Table 4.** Storage estimate $S_{95P}$ constrained by deuterium or tritium

| Statistics of $S_{95P}$ | $^2$H ($E_2 > 0$) | $^3$H ($E_3 < 0.5$ T.U.) | $^2$H and $^3$H |
|---|---|---|---|
| Mean ± st. dev. [mm] | 1275 ± 245 | 1335 ± 279 | 1488 ± 135 |
| Median ± st. dev. [mm] | 1281 ± 245 | 1392 ± 279 | 1505 ± 135 |
| Min [mm] | 625 | 660 | 1249 |
| Max [mm] | 1744 | 1806 | 1710 |

$S_{95P}$ is calculated as the 95$^{th}$ percentile of $\Omega_{tail}$ (eq. 11)

**Table B1.** Results from the Wilcoxon rank sum test comparing the travel time and storage measures between $^2$H and $^3$H behavioral solutions. The null hypothesis is that the measures are extracted from the same underlying distribution for both tracers.

| Travel time or storage measure | Decision about the null hypothesis | p-value |
|---|---|---|
| 10$^{th}$ percentile | Rejected | $3.3 \times 10^{-6}$ |
| 25$^{th}$ percentile | Rejected | $5.9 \times 10^{-8}$ |
| Median | Rejected | $1.5 \times 10^{-8}$ |
| 75$^{th}$ percentile | Rejected | $1.1 \times 10^{-3}$ |
| 90$^{th}$ percentile | Accepted | 0.30 |
| Mean | Rejected | $3.5 \times 10^{-5}$ |
| $F_{yw}$[a] | Accepted | 0.37 |
| $F(T < 6 \text{ months})$ | Rejected | $5.3 \times 10^{-6}$ |
| $F(T < 1 \text{ year})$ | Rejected | $2.7 \times 10^{-10}$ |
| $F(T < 3 \text{ years})$ | Rejected | $2.5 \times 10^{-3}$ |
| $S_{95P}$ | Rejected | $1.4 \times 10^{-2}$ |

All tests were made at the 5% significance level.

[a] Fraction of "young water" (Kirchner, 2016), younger than 0.2 years