# Peer review of "A comparison of catchment travel times and storage deduced from deuterium and tritium tracers using StorAge Selection functions"

_Hydrology and Earth System Sciences, 2019_

## Referee Comment (RC1) · Francesc Gallart (Referee) · 29 Nov 2019

The authors propose a very interesting piece of work that may shed light on the future joint use of deuterium and tritium isotopes on water age studies. The volume of the original analytical information is outstanding, the text is a little verbose but clear, the graphs are explicative and the rationale and methods are well explained although a relevant part is not described as it is under review in another journal.

Nevertheless, there are a few methodological issues that should be fixed or justified before the manuscript is acceptable for publication.

[Figure]

General comments.

The procedure used by the authors to test the "truncation" hypothesis "that streamflow TTDs calculated using only deuterium (2H) or only tritium (3H) are different" does not follow the established methods for hypothesis testing. As a rule, for rejecting a null hypothesis it is necessary to verify that its probability is lower than a prefixed assumable error risk, typically p<0.01. High uncertainty of the results is not sufficient for rejecting a null hypothesis.

The authors found that "differences between the various statistics of the TTDs were smaller than the uncertainties of the calculations when comparing the results obtained with 2H alone and with 3H alone". But the authors also state that "even though the uncertainties are sufficient to account for the differences between 2H - and 3H -derived age and storage measures, it is worth noticing that 3H systematically gave higher estimates". Therefore, even if the authors did not estimate the probability of the null (truncation) hypothesis, this last sentence suggests that its probability was not sufficiently low for rejecting it, so the result of this work is that the authors cannot reject the "truncation" hypothesis.

Furthermore, this hypothesis testing exercise had other issues. Indeed, although the authors "treated 2H and 3H equally by calculating TTDs using a coherent mathematical framework for both tracers (i.e. same method and same functional form of TTD)" they did not treat these isotopes with similar sampling strategies. Indeed, nearly 30 stream samples of 3H collected during highly varying flow conditions cannot be compared with the 1088 stream samples of 2H collected every 15 hours on average, even if the period was the same. Among the diverse causes that can explain the modest differences found between the results obtained with deuterium and tritium, the potential role of the different sampling strategy must be taken into account (differences respect to the sample number and flow representativeness, as also suggested by the authors in the discussion). The test performed by the authors compares the results and potentials of both isotopes when used under the current state of the art but not their own potentials.

[Figure]

A rigorous test for comparing the own potentials of both isotopes would need to use an equal number of samples taken simultaneously for both.

This leads to another relevant issue on the sample treatment. The authors, as commonly made, weighted the isotopic signal of rainfall waters with the respective rainfall depths. But nothing is sated on the weighting of stream samples, as regrettably also recurrently made. So the reader has to assume that the raw (unweighted) isotopic signals of stream samples were used for constraining the model.

My point is that this approach, if actually used, will provide a set of model parameters adequate to describe the isotopic signal of the samples as they are in the record, but not to simulate the isotopic mass balances, i.e. the main rationale of the model. If the isotopic mass balances are sought, it is necessary to weight the isotopic signal of every sample with the associated flow (time span X discharge). Furthermore, looking to Figure 4, it seems that the most highly scattered 2H samples were taken during low flows, so it could happen that the, really low, efficiency of the model would improve by flow-weighting the stream samples.

Another associated question is the representativeness of the stream samples of the diverse flow ranges in the catchment. In the discussion, the authors sensibly wonder if "tritium ... may still be biased towards hydrological recessions" and "how many measurements are enough and when to sample isotopes for maximum information gain on water ages". If the stream samples must represent the mass flow of water and tracers and a detailed flow record is available, it is possible to compare the distribution functions of both flow records (only measured versus measured and sampled) for assessing the degree of representativeness of the sampling designs. This kind of analysis should be customary in all catchment environmental tracing studies, particularly for small catchments where the flow duration curve is usually highly skewed.

Detailed comments:

Lines 12-13: The truncation (null) hypothesis cannot be rejected from the work results.

Line 122: "phyllade" is a French geological term. The closest English term, as far as I know, is "phyllite"

Line 330: ... This is not the case for d3H...

Line 380: The truncation (null) hypothesis cannot be rejected from the work results.

The model calibration method that consists of using a range of parameter sets instead of an 'optimal' parameter set was developed by Beven & Binley (1992). I suggest to cite this work also because it, as far as I know, was the first using the Shannon entropy for analysing the value of additional data in the calibration of a model.

Beven, K., & Binley, A. (1992). The future of distributed models: model calibration and uncertainty prediction. Hydrological processes, 6(3), 279-298.

---

## Referee Comment (RC2) · Anonymous Referee #2 · 10 Dec 2019

**1   General comments**

The manuscript tested the hypothesis that $^3$H tracer provides information over longer transit times than $^2$H. The authors calibrated the StorAge Selection (SAS) function model for each tracer and examined information gain using the posterior distributions of the model parameters. They rejected the hypothesis based on their results. Nevertheless, they concluded that $^3$H tracer is more informative and cost-efficient compared to $^2$H.

e topic is timely and very interesting.  However, the manuscript needs substantial re-

vision. First, I do not think that the results presented in this study support most of their conclusions. Their SAS function-based model performed poorly even with 12 parameters, and it is not clear how much we can learn from the poorly performing and not well-constrained model. Second, I have several issues with their analysis and the hypothesis test. These points are described in more detail in what follows.

**2  Major comments**

**2.1  Not enough dataset and the poorly performing complex model**

The model has an unusually large number of parameters (12 parameters; e.g., Line 249) compared to the previous SAS function-based modeling studies. I believe that the authors illustrated the need for more parameters well in their previous study, which is now published in WRR. However, the model does not perform well even with the 12 parameters (with the maximum NSE 0.24 for $^2$H), and I am not sure what we can learn from the poorly-performed model. The large number of parameters also causes several issues described below. Also, the dataset is very limited, and it is not clear if the limited number of samples and the limited sampling period support their conclusions.

First, it is not clear if the $^3$H dataset is enough. The number of samples is too limited to constraint 12 parameters. I can easily guess that the parameters are not well-constrained. Thus, it is obscure how much information we can extract from the time series, the posterior distributions of those parameters, the TTDs, and the SAS functions, which were used to test the hypothesis and examine if those tracers contain non-redundant information to each other. For example, the authors stated that "stable and radioactive isotopes have information in common about young water" in Lines 472-475. However, the argument cannot be supported by those 24 samples. Furthermore, how much information we can extract from the 2-years of $^2$H dataset? Can we talk

about transit time longer than 2 years (at the maximum) based on the model results?

Second, I think that their Latin hypercube sampling (Line 262) suffers the curse of dimensionality. They sampled 12,096 parameter sets from the 12-dimensional parameter space. It can be easily guessed that those samples are very sparsely distributed in the 12-dimensional parameter space (i.e., $12^4 > 12,096$), and the sparse sampling can potentially limit their ability to construct well-constrained posterior distributions of those parameters.

Lastly, the poor performance of the model leads me to think that maybe their model structure is not adequate, and any discussion based on the model results should be conducted more carefully. It is clear that the model fails to reproduce short time-scale dynamics. Figure 4 shows that their $^2$H-based model cannot capture the observed large fluctuation. It seems that the large fluctuation is, in part, due to the high correlation between $C_{p,2}$ and $C_{Q,2}$ especially when the system is dry, and It implies that short time-scale dynamics are not captured by the model (as they mentioned in Lines 512-513). The fluctuation seems much more pronounced in the $^2$H time series. Thus, if we have a better model that captures the short time-scale dynamics, it may contradict the authors' argument in Line 472: "stable and radioactive isotopes have information in common about young water."

**2.2  Analysis and Hypothesis test**

The use of the Kullback-Leibler Divergence $D_{KL}$ in the hypothesis test seems inappropriate. Throughout the manuscript, the authors stated that using both tracers together is valuable since $D_{KL} > 0$ (e.g., in Lines 435-436 and Lines 468-470). However, the criterion $D_{KL} > 0$ cannot determine whether the criterion is met because multiple tracers are used or because there is just any additional information. For example, $D_{KL}$ between the model constrained by, let's say, 100 $^2$H data and the model constrained by the rest of the $^2$H data will be greater than zero.

Moreover, different performance measures were used for their models (Lines 265-270), and it makes the use of $D_{KL}$ even more inappropriate. The authors used the NSE for the $^2$H-based model and used the MAE for the $^3$H-based model. Thus, the difference between the posterior distributions estimated by those behavioral models can be, in part, explained by the choice of performance measure. For example, if the authors estimate the posterior distributions using the $^2$H dataset based on the MAE, the posterior distributions would differ from those estimated based on the NSE. Then, $D_{KL}$ would be greater than zero. Thus, it is not hard to follow their argument that using both tracers together is valuable (e.g., in Lines 331-333, Lines 435-436, Lines 478-470, and Lines 580-581).

Furthermore, I disagree with their cost analysis (in Lines 445-451), which led them to conclude that $^3$H tracer is more cost-effective (e.g., Line 17). As described in Lines 462-463, "The amount of information learned from the isotopic data probably scales non-linearly and probably reaches a plateau as the number of observation points grows." However, they assumed "linearity" in their cost analysis. Thus, the analysis is not valid.

Lastly, it seems that the ET SAS functions are very important in this study but rarely explained. One of its parameters, $\mu_{ET}$ is the most valuable parameter in terms of the information gain in this study (see Table 2). However, no explanation is provided why it is the most valuable and how it affects their interpretation of the results. For example, Figure 5 is one of the most important figures that clearly illustrates the difference between the $^2$H-based model and the $^3$H-based model. The simulated $^3$H concentration using the $^2$H-based model, in general, is higher than that simulated using the $^3$H-based model. It means that tracer mass partitioned into discharge is smaller in the $^3$H-based model during the period. Since there is no explanation on the difference, I had to guess that either more $^3$H tracer mass is stored in the system in the $^3$H-based model or more $^3$H tracer was partitioned into evapotranspiration in the model. Overall, it seems that the partitioning is one of the most important differences between the two models. Thus, the partitioning should be explained in more detail.

**3   Minor comments**

1. The posterior distribution of the parameters should be presented in this manuscript. It is hard to grasp most of the authors' arguments without those distributions.

2. Line 375: Typo in "[0,$\infty$["

3. Line 224: It is stated that $\lambda_1(t)$ is the smallest weight. However, it is not clear how that was constrained in the model calibration.

4. Lines 236-237: $S_{ref}$ is chosen not calibrated, so probably introducing the chosen value here would be better, rather than introducing it in the next section, 2.6 Model calibration.

5. The initial condition for the SAS function model is not described. If there was a spin-up for the SAS function model (like the storage estimation), what tracer time series were used?

6. Lines 404-405: How this comparison between 2016 finding and 2017 finding helps readers to understand the higher age estimated using the $^3$H-based model?

7. Lines 437-439: Those parameters are not independent. Thus, those were not independently constrained.

---

## Author Comment (AC1) · 17 Dec 2019

FG: The authors propose a very interesting piece of work that may shed light on the future joint use of deuterium and tritium isotopes on water age studies. The volume of the original analytical information is outstanding, the text is a little verbose but clear, the graphs are explicative and the rationale and methods are well explained although a relevant part is not described as it is under review in another journal.

Nevertheless, there are a few methodological issues that should be fixed or justified
before the manuscript is acceptable for publication.

Authors: We thank Francesc Gallart (FG) for the overall positive reception and constructive evaluation of our work. Please note that the mentioned study is now published (open access) in WRR as:

Rodriguez, N. B., & Klaus, J. (2019). Catchment travel times from composite StorAge Selection functions representing the superposition of streamflow generation processes. *Water Resources Research*, 55. https://doi.org/10.1029/2019WR024973

We are grateful for FG's relevant suggestions and we will provide appropriate modifications in order to improve to the manuscript accordingly.

FG: The procedure used by the authors to test the "truncation" hypothesis "that streamflow TTDs calculated using only deuterium ($^2$H) or only tritium ($^3$H) are different" does not follow the established methods for hypothesis testing. As a rule, for rejecting a null hypothesis it is necessary to verify that its probability is lower than a prefixed assumable error risk, typically p<0.01. High uncertainty of the results is not sufficient for rejecting a null hypothesis.

Authors: We understand that strong statements such as "We found equal TTDs and equal mobile storage between the $^2$H- and $^3$H-derived estimates" and our use of the words "hypothesis", "reject", or "testing" in the title could be interpreted as if we applied some statistical test in the traditional framework of hypothesis testing. Our intention was not to conclude on the statistical significance of the results, but rather to show that the potential water age differences obtained with the two tracers are not as drastic as generally expected since the study of Stewart et al. (2010). Our goal was thus to show a counterexample to the conjecture that the tails of the TTDs are systematically truncated when using seasonal tracers. We will thus revise the manuscript accordingly, to avoid misinterpretations. Notably, we will change the word "testing" in the title with "assessing". Moreover, the scientific method does not rely only on statistical hypothesis testing to move forward, for various reasons (Pfister and Kirchner, 2017). Important hydrological conjectures, such as the idea that streamflow is made only of overland flow, were proven wrong without a probability criterion because new experimental data (e.g. strong damping of stable isotopic signatures) provided clear evidence in favor of alternative explanations (Kirchner, 2003).

We did not mean to use the parameter uncertainties as a criterion to assess if the water age differences can be considered statistically significant or not. Instead, we simply pointed out that the observed differences are small. Since "small" is always subjective, we compared these age differences to what we had available, i.e. the parameter uncertainties. This comparison raised the question whether the age differences can be confidently interpreted as representative of a TTD truncation issue or not. We will revise this part of the discussion to make it clearer that the age differences are in fact smaller than what was expected based on the study of Stewart et al. (2010), and that this is actually the main reason why we doubt that the TTD tails are systematically truncated when using only deuterium as a tracer.

FG: The authors found that "differences between the various statistics of the TTDs were smaller than the uncertainties of the calculations when comparing the results obtained with $^2$H alone and with $^3$H alone". But the authors also state that "even though the uncertainties are sufficient to account for the differences between $^2$H- and $^3$H -derived age and storage measures, it is worth noticing that $^3$H systematically gave higher estimates". Therefore, even if the authors did not estimate the probability of the null (truncation) hypothesis, this last sentence suggests that its probability was not sufficiently low for rejecting it, so the result of this work is that the authors cannot reject the "truncation" hypothesis

Authors: We thank FG for pointing out this potential interpretation issue that can be addressed with a proper statistical analysis. We will therefore add a Wilcoxon rank sum test to the revised manuscript. The results

show that there is a statistically significant difference in most (but not all) of the age measures shown in table 3 (e.g. median age, mean age). We will include these results in the appendix and refer to them in the discussion.

However we believe that these results do not change the core message of the study, for various reasons. First, as mentioned above, the age differences are small compared to those suggested by Stewart et al. (2010) and subsequently assumed by many researchers working with tritium. For example, the largest age difference we found (41%) was actually for the youngest water fractions, while our mean travel times differed only by <7%. In contrast, the mean travel times compared by Stewart et al. (2010) can for example differ by a factor of nearly 200%. Second, as written in the discussion, we think that these age differences can be mostly explained by the large difference in the number of tritium samples (24) compared to deuterium samples (>1000). Although the statistical analysis suggests a significant difference between $^2$H- and $^3$H- derived water ages, it is really important to remember that this analysis does not take into account the large difference in the number of tracer samples! Let's imagine the opposite situation: 24 samples for deuterium and >1000 for tritium, especially keeping in mind figures 6a and 6b. How would behavioral simulations look then? It is then difficult to say a priori whether the corresponding TTDs would be similar to those found now, and whether the TTDs would then be consistent between $^2$H and $^3$H. We believe that currently, with only 24 tritium samples compared to >1000 deuterium samples, it is very unlikely that the consistency we found between the TTDs is a simple coincidence.

We will carefully reformulate the abstract, the discussion, and the conclusion, to include the statistical results, and to soften the claim that the TTDs are equal. Rather, we will present that the $^2$H- and $^3$H- derived TTDs are mostly consistent in terms of shape and percentiles (e.g. mean). We will also add in the discussion another potential physical interpretation about water age differences with respect to the self-diffusion of HDO and HTO in water.

FG: Furthermore, this hypothesis testing exercise had other issues. Indeed, although the authors "treated $^2$H and $^3$H equally by calculating TTDs using a coherent mathematical framework for both tracers (i.e. same method and same functional form of TTD)" they did not treat these isotopes with similar sampling strategies. Indeed, nearly 30 stream samples of $^3$H collected during highly varying flow conditions cannot be compared with the 1088 stream samples of $^2$H collected every 15 hours on average, even if the period was the same. Among the diverse causes that can explain the modest differences found between the results obtained with deuterium and tritium, the potential role of the different sampling strategy must be taken into account (differences respect to the sample number and flow representativeness, as also suggested by the authors in the discussion). The test performed by the authors compares the results and potentials of both isotopes when used under the current state of the art but not their own potentials. A rigorous test for comparing the own potentials of both isotopes would need to use an equal number of samples taken simultaneously for both.

Authors: Given the measurement techniques limitations and price, we are not sure that the concept of "own potential" can be clearly defined if the tracer signals are not continuous (i.e. with an infinite number of points). Indeed, each tracer will always be associated with a given (finite) number of samples, and this number of samples for $^3$H will most likely be much lower than the number of $^2$H samples unless the sampling for deuterium is voluntarily coarse. One may think that it could be useful to restrict the number of $\mathbf{\delta}^2$H samples to match the number and/or the timing of $^3$H samples in order to define this "own potential". The first issue is that it would correspond to ignoring the facts (the measurements we already have), i.e. ignoring the true variability of $\delta^2$H in favor of that of $^3$H, which appears conceptually wrong to us. We know that $\delta^2$H varies in such a way and there is information (quantifiable, see section 2.7) to gain from it. Ignoring samples can only reduce the amount of information extracted from the tracer data, or worse, support the wrong interpretations. Moreover, in our case there are already more than $10^{48}$ ways to select 24 samples among 1088. It is nearly impossible to test all combinations. Even by being more strategic, for example by using a flow duration curve (FDC) to select 24 deuterium samples among 1088, there is still a lot of subjectivity involved. For instance, selecting samples distributed along the FDC implies a hidden assumption of a one-to-one relationship between a given flow value and streamflow generation processes or catchment state variables such as soil moisture, groundwater levels, or catchment storage. This means that one can never be sure that all "end members" or "wetness states" or "streamflow generation processes" are accurately represented in the selected tracer data set with such a method, and that there may always be a sampling bias. Finally, we did try to compare the "own potentials" in the discussion (4.3) by showing the amount of water

age information gained per deuterium/tritium sample or per €. This normalization per price or per number of samples allowed us to take some perspective on the results and to quantify to what extent tritium seems more age-informative than deuterium for our current number of samples, without having to ignore any deuterium measurement.

FG: This leads to another relevant issue on the sample treatment. The authors, as commonly made, weighted the isotopic signal of rainfall waters with the respective rainfall depths. But nothing is stated on the weighting of stream samples, as regrettably also recurrently made. So the reader has to assume that the raw (unweighted) isotopic signals of stream samples were used for constraining the model.

Authors: We did not state in the manuscript that we weighted the isotopic signal of precipitation with respect to precipitation amounts. We will clarify in sections 2.2 and 2.3 (especially equations 1, 2, and 3) that it is the unweighted signals (for stream and precipitation samples) that are used. Weighting functions for the input signal were introduced in travel time theory in early studies that considered only groundwater systems because these could reasonably be assumed to be at steady-state (Maloszewski and Zuber, 1982). In this case, the input function of groundwater systems is not described well by the precipitation signal because of mixing due to the complexity of flow paths in the unsaturated zone and because of water losses to the atmosphere via ET. It is not necessary to use an input weighting function with time-varying TTDs that consider the whole catchment and obtained with SAS functions, because the effect of ET is implicitly taken into account in the Master Equation (Botter et al. 2010), and because the effect of mixing in the unsaturated zone is included in the definition of the streamflow TTD. We will add this information in section 2.3.

FG: My point is that this approach, if actually used, will provide a set of model parameters adequate to describe the isotopic signal of the samples as they are in the record, but not to simulate the isotopic mass balances, i.e. the main rationale of the model. If the isotopic mass balances are sought, it is necessary to weight the isotopic signal of every sample with the associated flow (time span X discharge). Furthermore, looking to Figure 4, it seems that the most highly scattered $^2$H samples were taken during low flows, so it could happen that the, really low, efficiency of the model would improve by flow-weighting the stream samples.

Authors: The isotopic mass balance takes the following form (Rodriguez and Klaus, 2019):

$$dM/dt = J C_P - Q C_Q - ET\, C_{ET}$$

With our model described in section 2.4, we are able to numerically calculate all the terms of the right hand side of the equation, hence the term on the left hand side as well. However, the main objective of the model is not to simulate the isotopic mass balance, but only to simulate the isotopic signal in a given outflow, here $C_Q$ for which we have tracer observations. This is sufficient to show that the transport from precipitation to the stream is correctly modelled and that the streamflow travel times are correct. Solving the isotopic mass balance is useful only to know in addition how the isotopic tracer mass in the catchment changes with time. We do not focus on this term because we do not have representative tracer data for the ET flux. This means that we are unable to compare our simulated $C_{ET}$ to any observation. Without appropriate tracer data for ET, both the flux term corresponding to ET (ET times $C_{ET}$) and the "mass change term" (dM/dt) cannot be verified against experimental data, and thus depend on each other. We will emphasize again on this point in section 2.4.

Moreover, we think that focusing on the flow-weighted isotopic signal is problematic for the goals of our study. The flow signal varies considerably more than the isotopic signals. The variations of the product signal (flux times isotope) therefore mostly depend on the flow variations. Although calibrating a model to such flux-weighted signal could improve the performance measures thanks to this, it would also overlook the isotopic variations. Our goal is not only to improve performance measures, but to accurately simulate the variable of interest, here the unweighted tracer signal, that carries most of the information about travel times (while water fluxes in themselves do not). We discussed in our related paper (Rodriguez and Klaus, 2019) the relevance of these unusually low values of NSE for deuterium and the issues with this objective function in our particular case. To avoid overlap with this study, we will refer the reader to this paper for more details on the choice of objective function.

FG: Another associated question is the representativeness of the stream samples of the diverse flow ranges in the catchment. In the discussion, the authors sensibly wonder if "tritium... may still be biased towards hydrological recessions" and "how many measurements are enough and when to sample isotopes for maximum information gain on water ages". If the stream samples must represent the mass flow of water and tracers and a detailed flow record is available, it is possible to compare the distribution functions of both flow records (only measured versus measured and sampled) for assessing the degree of representativeness of the sampling designs. This kind of analysis should be customary in all catchment environmental tracing studies, particularly for small catchments where the flow duration curve is usually highly skewed.

Authors: This is a good remark. We will include the following figure showing the distribution of isotopic samples along a flow exceedance probability curve in section 2.2. Our sampling scheme covered flows with exceedance probabilities going down to 2e-4 for deuterium and down to 0.09 for tritium. This makes the sampling scheme rather representative of all flow conditions. Note however that we did not select the 24 tritium samples based on this curve, but based on the streamflow time series. We selected samples at different flow conditions representing interesting hydrological events (e.g. beginning of a wet period after a long dry period, small but flashy streamflow responses), based on our experimental knowledge of this catchment and on our previous experience with deuterium data (Rodriguez and Klaus, 2019). We will add this detail to section 2.2. Comparing the histograms of measured vs sampled flow records is not very meaningful for tritium because there are only 24 measured values (against more than 4000 for flow alone).

[Figure]

FG: Lines 12-13: The truncation (null) hypothesis cannot be rejected from the work results.

Authors: See the answer to the general comments. This is correct, the statistical hypothesis cannot be rejected. However, one has to keep in mind that the point of our work was not to conclude on the statistical significance of the age differences we found. Our point was rather to show that the TTDs are not so drastically different, which acts as a counterexample to the conjecture of Stewart et al. (2010) that seasonal tracers systematically truncate the long tails of the TTDs. Moreover, the current lack of high-resolution

tritium data means that it cannot be safely concluded from the simple statistical analysis of these results that the TTDs are truly different. We will revise the manuscript to make this aspect clearer.

FG: Line 122: "phyllade" is a French geological term. The closest English term, as far as I know, is "phyllite"

Authors: We thank FG for pointing this out. We will change it as suggested.

FG: Line 330: ... This is not the case for d3H...

Authors: We suppose FG thought that we meant "$^3$H" and not what is currently written, "$\delta^2$H". We really meant $\delta^2$H. We will rewrite this to avoid any confusion.

FG: The model calibration method that consists of using a range of parameter sets instead of an 'optimal' parameter set was developed by Beven & Binley (1992). I suggest to cite this work also because it, as far as I know, was the first using the Shannon entropy for analysing the value of additional data in the calibration of a model.

Authors: We thank FG for the relevant suggestion, and we will add this reference.

Authors: We will also modify figure 5 to better represent the standard error (1 standard deviation of measurements) above and below the points. The current figure shows only half a standard deviation above and below the points.

Kirchner, J. W. (2003). A double paradox in catchment hydrology and geochemistry. Hydrological Processes, 17, 871-874. https://doi.org/10.1002/hyp.5108

Pfister, L., & Kirchner, J. W. (2017). Debates—Hypothesis testing in hydrology: Theory and practice. Water Resources Research, 53, 1792–1798. https://doi.org/10.1002/2016WR020116

---

## Author Comment (AC2) · 10 Jan 2020

Authors: Francesc Gallart (FG) reacted to our reply to his referee comments. He sent us some additional thoughts of improvement by email because the manuscript is currently in the "author comments only" phase. We obtained FG's approval to reply in HESSD by reproducing his comments below.

*FG: The discussion through HESSd is over, but I wanted to shortly react to your kind response to my referee comment.*

*- Sampling.*
*Thanks for the flow duration curve. It confirms my worries: tritium sampling is partial and deuterium sampling is biased.*
*As usually, you plotted the curve of discharges respect to time although this not the relevant variable with such a skewed distribution, but the relative cumulated flow. I made some gross calculations of the area (flow\*relative time) and found that the tritium sample for the highest discharge (exceeded in time only 0.09) was exceeded in flow about 0.45: Your tritium sampling discarded about 45% of the highest flows, so it is not only biased but really partial.*
*The figures are fortunately much better for deuterium, but my gross estimate is that the 40 samples taken for the highest flows represent about 23% of the cumulated highest flows: 4% of samples represent 23% of highest flows: your sampling is much biased.*
*The implications are that: (i) my objections on the way you compare your deuterium results with tritium ones are highlighted (ii) your sampling is not representative of the stream flows. For deuterium you must flow-weight your samples in order to compensate the biased sampling.*

*- sample weighting.*
*I am very surprised by your answer. After Botter et al. (2011) "the residence time distribution describes the pdf of the ages of all **water particles stored** inside a catchment/hillslope transport volume at a given time, and plays a key role in describing **the catchment storage of water and pollutants**". A water particle is a mass element. Concentrations cannot be stored. Your SAS functions select the ages in the catchment store to be output by runoff or ET and these (relative) mass outputs are updated in the catchment store. Your goal may not be the water mass balance, but you need the tracer mass balance to simulate the outputs of the system, and this cannot be made without mass weighting isotope inputs and outputs.*
*Indeed, flow varies much more than concentrations, but this is the real hydrological world. One hour of high discharge may transport more water and tracer mass than several weeks of low flows.*
*You may argue that your model should predict any unweighted stream water isotopic sampling. This might be true for a 'perfect' model if the precipitation isotopy was mass-weighted, but not for a model that has so much unexplained variance. For a non-perfect model, the result of the NSE depends on the samples you use, so you can try how diverse sets of samples give different NSE results and different behavioural parameters, but, frankly speaking, I would prefer to use precipitation and flow-weighted concentrations for a sound simulation.*

*I hope that these thoughs will be useful for a better revision of your nice paper.*
*All the best*
*Francesc*

Authors: We sincerely thank FG for the additional remarks. Regarding the sampling, we found similar numbers. The highest flows that were not sampled for tritium represent about 50% of the water that left the catchment via streamflow over 2015-2017. For deuterium, the highest flows associated with 40 samples (about 4% of the samples) represent about 20% of the water leaving via streamflow over 2015-2017.

In brief, this is what we will emphasize on in the revised manuscript (we nevertheless wrote more details below):

a) The employed sampling technique is not designed to measure the tracer masses, but their concentrations. Only nearly-continuous sampling or time-integrated samples can measure the tracer masses.

b) The limited number of $^3$H samples compared to $\delta^2$H samples does not allow a comparison of the exported tracer masses for each isotope, but it still allows a comparison of the stream water ages for each isotope.

c) Flow-weighting the stream samples will not compensate for the potential lack of samples during high and/or low flows.

d) Simulating only the tracer concentrations is sufficient to validate the TTDs.

e) Time-varying TTDs already implicitly account for the catchment-scale mass balance, no additional flow-weighting of the input and/or output tracer signal is necessary.

Here are additional details on the reasoning:

a) Our sampling is based mostly on fixed time intervals generally larger than a few hours. Thus, it should not be a surprise that the water mass is not proportionally represented in the sampling scheme. For this, an adaptive sampling frequency based on accumulated flows needs to be implemented (e.g., one sample every dozen m$^3$). In our case this would nevertheless lead to a much larger number of samples, exceeding the available field and lab resources. With more frequent samples during higher flows and less frequent samples during low flows, the mass of water flowing out of the catchment would of course be better represented. However, this would imply that the samples are not evenly distributed in time (hence along the FDC), which could also be criticized for being unrepresentative of all hydrological times of the year (i.e., over-representation of wet and cold conditions). It appears that choosing a type of sampling scheme (i.e. flow-proportional vs. fixed time intervals) will not allow to have the samples evenly distributed in time AND representative of all the water mass leaving via streamflow, unless streamflow is constant. Only nearly-continuous in-situ measurements that are currently available for stable isotopes can avoid these limitations (e.g., von Freyberg et al., 2017). Alternatively, a time-integrative sampling technique (that implicitly uses flux-weigthing) should be used for streamflow if the goal of the work is to simulate the exported tracer mass and compare it to the observations (this is not our goal). Note that the precipitation tracer measurements are time-integrative by design.

b) Even with the time-based sampling scheme and the limited number of tritium samples, the good agreement between TTDs constrained by deuterium and the TTDs constrained by tritium shows that the large water mass not sampled for tritium is not creating a strong bias towards young or old water compared to deuterium. This was different in previous tritium studies that focused on baseflow, where perhaps 90% of the water mass leaving the catchment via streamflow was not sampled for tritium and contained all the young water fractions. Our tritium data set most likely contains a rather representative selection of young and old stream water, even if not all water mass was not sampled.

c) Our goal is to accurately estimate the streamflow TTD at all times of the year. Accurately simulating the tracer mass flux in streamflow will not help reach this goal better than accurately simulating the tracer concentrations only. This is for the reasons outlined below. To put it more quantitatively, our model errors take the form:

$$\varepsilon_C\left(t_{obs}\right) = C_{modelled}\left(t_{obs}\right) - C_{observed}\left(t_{obs}\right)$$

where only the times corresponding to stream samples $t_{obs}$ are used (this avoids interpolating $C_{observed}$). Minimizing $\varepsilon_C$ at all times when we have observations allows us to constrain the TTDs to the most accurate estimate given our current tracer data set. If we were to flow-weight the tracer samples, the model errors would take the form:

$$\varepsilon_{QC}(t_{obs}) = Q_{observed}(t_{obs}) C_{modelled}(t_{obs}) - Q_{observed}(t_{obs}) C_{observed}(t_{obs})$$

because measured streamflow is used as an input in our model (there is no $Q_{modelled}$). Note again that only the times $t_{obs}$ when we have measurements $C_{observed}$ can be used. This is why flow-weighting the stream samples will not compensate for the lack of higher resolution tracer data over 2015-2017. There will still be some missing knowledge about the true variability of the tracer concentrations and the true tracer mass flux in streamflow. Furthermore:

$$\varepsilon_{QC} = Q_{observed} \varepsilon_C$$

This means that minimizing $\varepsilon_{QC}$ by adjusting model parameters is similar to minimizing $\varepsilon_C$ (because $Q_{observed}$ does not depend on parameters), and it yields the same TTDs. The NSE does not try to minimize each individual error but a squared sum of errors normalized by the observed variance. For $\varepsilon_{QC}$ this would give much more weight to periods with high flows, and the TTDs during drier periods would not be accurate anymore. Now, the variance of QC is much bigger than that of C, which can also "artificially" allow higher NSE values. Therefore, with flow-weighting, the "performance" of the model would improve, but this would lead to less reliable constraints on the TTDs because NSE>x for $\varepsilon_{QC}$ is clearly less strict than NSE>x for $\varepsilon_C$. The intuitive interpretation is that flows Q do not contain considerable information about the time scales of transport, only tracer concentrations do. Including the flows in the calibration can only reduce the information learned about stream water ages.

d) & e) Moreover, the convolution integral implicitly includes flow-weighting. We agree that "concentrations cannot be stored". Our approach does not store only concentrations, but also the associated particle volumes and thus mass. As written in section 2.3, Equation 1 expresses the fact that the stream concentration is the volume-weighted arithmetic mean of the concentrations of the water parcels with different travel times at the outlet. Let's imagine a streamflow grab sample represented below:

[Figure]

Each water particle k (there are n=4 particles represented here) has a given volume $V_k$ and a given concentration $C_k$. The measured tracer concentration of the sample is:

$$C_{obs} = \frac{\sum_{k=1}^{n} C_k V_k}{\sum_{k=1}^{n} V_k}$$

which can be rewritten:

$$C_{obs} = \sum_{k=1}^{n} C_k \frac{V_k}{\sum_{k=1}^{n} V_k} = \sum_{k=1}^{n} C_k p_k$$

where $p_k$ is the fraction of streamflow volume associated with particle k. Now, if we label each particle k with its age relative to the precipitation input, $C_k$ and $V_k$ simply become the corresponding past (time-varying) precipitation amounts and concentrations, and $p_k$ simply becomes the backward TTD.

Equation 1 in the manuscript is simply the continuous version of the equation above, for n tending to infinity. Therefore, the backward TTD needs no additional flow-weighting with respect to precipitation because it already includes it (the time-varying $V_k$). Furthermore, if an unsteady TTD is used, the stream flow variations are already included in its definition (by the time-varying denominator $\sum_{k=1}^{n} V_k(t) = Q(t)\Delta t$), and no flow-weighting of $C_{obs}$ is needed to correctly deduce the TTD from the convolution integral.

From this equation we now easily guess the data requirements of the approach, sufficient to estimate the TTDs and to respect the mass balance. In terms of tracer: a continuous tracer concentration input signal, and a time series of tracer concentrations in the outflow. The finer the resolution of the time series of the output concentration, the less uncertainty there should be about the TTD, because fewer weighted combinations of all the $C_k$ will closely match all the $C_{obs}$ simultaneously. In terms of hydrometric measurements: precipitation rates, and stream flows. In addition, to calculate the TTD from the Master Equation, storage needs to be deduced from the catchment-scale water balance equation. This requires actual ET to be calculated as well.

von Freyberg, J., Studer, B., and Kirchner, J. W.: A lab in the field: high-frequency analysis of water quality and stable isotopes in stream water and precipitation, Hydrol. Earth Syst. Sci., 21, 1721–1739, https://doi.org/10.5194/hess-21-1721-2017, 2017.

---

## Author Comment (AC3) · 10 Jan 2020

R2: The manuscript tested the hypothesis that $^3$H tracer provides information over longer transit times than $^2$H. The authors calibrated the StorAge Selection (SAS) function model for each tracer and examined information gain using the posterior distributions of the model parameters. They rejected the hypothesis based on their results. Nevertheless, they concluded that $^3$H tracer is more informative and cost-efficient compared to $^2$H.

The topic is timely and very interesting. However, the manuscript needs substantial revision. First, I do not think that the results presented in this study support most of their conclusions. Their SAS function-based model performed poorly even with 12 parameters, and it is not clear how much we can learn from the poorly performing and not well-constrained model. Second, I have several issues with their analysis and the hypothesis test. These points are described in more detail in what follows.

Authors: We thank the reviewer (R2) for the detailed assessment of the work and for suggestions of improvement. Regarding the hypothesis testing, we were not clear in our writing. We did not intend to test the statistical significance of the water age differences derived from different tracers, but rather wanted to prove that the age differences are much smaller than previously shown (Stewart et al., 2010) and assumed in most following tracer studies. As a consequence of the comments from Francesc Gallart (FG) and R2, also written in more detail in our reply to FG, we will now also include a statistical test in the revised manuscript.

We note R2's concerns about our model and data. We detailed below why we think that we can still derive robust conclusions from the modelling exercise. We will modify the manuscript to clarify this and to address R2's comments.

R2: The model has an unusually large number of parameters (12 parameters; e.g., Line 249) compared to the previous SAS function-based modeling studies. I believe that the authors illustrated the need for more parameters well in their previous study, which is now published in WRR. However, the model does not perform well even with the 12 parameters (with the maximum NSE 0.24 for $^2$H), and I am not sure what we can learn from the poorly-performed model. The large number of parameters also causes several issues described below.

Authors: We understand R2's concern that the model does not perform sufficiently well despite the large number of parameters it has. We will rephrase parts of the discussion to stress that the model is of course not in perfect agreement with the observations, and that a better model may change the interpretation of the results to some extent. We already proposed some suggestions of improvement of the model for future studies (section 4.4 and our answer to a comment further below). We agree with R2 that the NSE cannot be considered high, but we disagree with R2's interpretation that the model is performing poorly. In our previous study (Rodriguez and Klaus, 2019), we detailed why such a complex model structure is adequate for this catchment, even if the NSE appears unusually low. We also emphasized on the fact that 12 parameters is a small number to constrain the vast array of time-varying processes leading to the selection of particular water ages by Q and ET from anywhere in catchment storage (represented here in the Master Equation by ~$10^5$ "age control volumes" and their associated age fluxes). We previously detailed the limitations of the NSE for evaluating model performance with such complex tracer time series (see also 4.4, the NSE assumes normally distributed, uncorrelated, and homoscedastic errors). Other performance measures have been proposed (e.g., Schoups and Vrugt, 2010; Ehret and Zehe, 2011), but they either require more parameters, or they are not designed for tracer time series but only for hydrographs.

Furthermore, the evaluation of model performance usually involves expert knowledge (Gharari et al., 2015; Hrachowitz et al., 2014) that cannot be expressed via the traditionally used objective functions (Seibert and McDonnell, 2002). The Weierbach $\delta^2$H time series has unusually damped seasonal dynamics, while at the same time unusually strong flashy events occur. A close look at the behavioral simulations (see figure 4) reveals that some runs were actually able to match the flashy $\delta^2$H dynamics quite well. A zoom on figure 4 allows to see the short-term simulation capabilities of the model (the very thin peaks of the simulation envelopes). We will add an inset with a zoom on particular peak in figure 4. We will add figures (see a few examples below) in the supplement showing more details about the behavioral simulations. In these figures, it is remarkable that only several dozen data points among the more than 1000 were not captured by

behavioral simulations in deuterium. These points are almost all during summer 2016 and summer 2017 (drier periods). The other interesting aspect is that behavioral simulations in tritium were able to match many of these extreme values. We believe that this is because the behavioral simulations in tritium were not penalized by the limitations imposed by the NSE, and were thus allowed to have more extreme variations.

[Figure]

Figure: δ$^2$H simulations in Nov-Dec 2015

[Figure]

Figure: δ$^2$H simulations in Jul-Oct 2016

[Figure]

Figure: $\delta^2$H simulations in winter 2016

Although higher NSE values were reported in the past for other $\delta^2$H time series simulated with transient TTDs (e.g. NSE > 0.5; Benettin et al., 2017; Harman, 2015; van der Velde et al., 2015), we disagree to state that our model performs poorly simply because the NSE values are not as high. The NSE of the behavioral simulations is not closer to 1 partly because of the underlying assumptions about model residuals in the NSE (Rodriguez and Klaus, 2019). Care should be taken in interpreting the NSE values. The NSE does not allow a reliable performance comparison between different studies and it is not an absolute measure of model performance, because it implicitly uses the mean observed value as a benchmark model. This benchmark model is not always the best choice, as stressed in several studies (Seibert, 2001; Schaeffli and Gupta, 2007; Criss and Winston, 2008). In our particular case, the mean observed value is particularly penalizing because the $\delta^2$H time series has many more points corresponding to very damped seasonal fluctuations than points corresponding to the large flashy fluctuations. Within tracer hydrology and modelling there is an urgent need for better ways of summarizing model efficiency. Yet, this is beyond the scope of this study, especially because it focuses the calibration task while our goal is to focus on what can be learned from the isotopic data set in terms of water ages. We will add these points to section 4.4 in the discussion.

R2: Also, the dataset is very limited, and it is not clear if the limited number of samples and the limited sampling period support their conclusions. First, it is not clear if the [3]H dataset is enough. The number of samples is too limited to constraint 12 parameters.

Authors: The [3]H data set has, with the study of Visser et al. (2019), one of the highest number of stream samples analyzed for [3]H and used for travel time analysis. We understand that this may appear as a small number to constrain 12 parameters in the more general context of environmental modelling studies, but this is very common in travel time studies involving tritium. Many previous studies had about as many parameters as tritium samples or a just a few samples per parameter (Maloszewski and Zuber, 1993; Uhlenbrook et al., 2002; Stewart et al., 2007; Stewart and Thomas, 2008; Stewart and Fahey, 2010; Morgenstern et al., 2010; Cartwright and Morgenstern, 2016a, 2016b; Duvert et al., 2016; Gallart et al., 2016; Gusyev et al., 2016; Gabrielli et al., 2018). We will cite some of these studies and mention this point in sections 2.2, 2.6, and 4.4. Future studies may present a higher number of tritium samples if the analyses become more affordable.

R2: I can easily guess that the parameters are not well-constrained. Thus, it is obscure how much information we can extract from the time series, the posterior distributions of those parameters, the TTDs, and the SAS functions, which were used to test the hypothesis and examine if those tracers contain non-redundant information to each other.

Authors: We will include the parameter posterior distributions (see below) in a supplementary file. Most distributions are not flat (i.e. not uniform), indicating that the parameters are identifiable to some extent. We also note that all the parameters directly related to the shape of the SAS functions hence the TTDs ($\mu_2$, $\theta_2$, $\mu_3$, $\theta_3$, $\mu_{ET}$, $\theta_{ET}$) are visually clearly not uniform. We initially used Shannon's entropy H and the Kullback-Leibler Divergence DKL concepts for parameter identifiability instead of these figures to have a more objective and more quantifiable uncertainty assessment. We note that "how much information we can extract from time series, the posterior distributions of those parameters..." is exactly quantified via equations 8 and 9. We will explain these concepts in more detail in section 2.7 and add a line in table 2 corresponding to the DKL between prior and posterior distributions for each parameter.

[Figure]

Figure: Posterior distributions constrained by deuterium

[Figure]

Figure: Posterior distributions constrained by tritium

R2: For example, the authors stated that "stable and radioactive isotopes have information in common about young water" in Lines 472-475. However, the argument cannot be supported by those 24 samples. Furthermore, how much information we can extract from the 2-years of $^2$H data set? Can we talk about transit time longer than 2 years (at the maximum) based on the model results?

Authors: We are not sure what is meant exactly by "the argument cannot be supported by those 24 samples" and thus how to cope with this comment. As indicated in the following sentences (lines 472-475) we believe that the high variability of stream tritium concentrations, that follow the variations of precipitation concentrations, indicates that it is very likely the effect of young water contributions. This was unobserved before due to a focus on baseflow sampling, except for rare studies showing high tritium variability during short-term hydrological events (Hubert et al., 1969; Crouzet et al., 1970; Dinçer et al., 1970). Tritium has therefore been generally considered to be informative only about old water (we will emphasize on this detail in the corresponding paragraph). However, tritium can be used and has been used to detect young water contributions, for example in the first studies using hydrograph separation (Klaus and McDonnell, 2013).

Moreover, as it can be seen in table 3, we have travel times above 2 years (e.g. mean > 2). We have travel times up to about 100 years (see figure 6). This is possible due to the 100 year spin-up period (1915-2015) that we systematically used before evaluating each simulation over 2015-2017. We will add a sentence to clarify this in section 2.5.

R2: Second, I think that their Latin hypercube sampling (Line 262) suffers the curse of dimensionality. They sampled 12,096 parameter sets from the 12-dimensional parameter space. It can be easily guessed that those samples are very sparsely distributed in the 12-dimensional parameter space (i.e., $12^4$ > 12,096), and the sparse sampling can potentially limit their ability to construct well-constrained posterior distributions of those parameters.

Authors: We understand that 12,096 parameter samples for a 12 dimensional space can be less than one may hope for. We also understand that it would be ideal if we had several more orders of magnitudes in the number of samples. However, we are currently limited by computational time (more than 1 hour) to run the model with each parameter sample, despite the use of a highly parallelized code with a high performance computer. This computational time is so large because of the need to spin-up the model for 100 years (see above). Without this spin-up, a numerical truncation of the TTDs will occur.

As suggested by R2, the parameter sets are thus likely to be sparsely distributed. The LHS technique was thus employed to make sure that the samples are distributed as evenly as possible in this high-dimensional space (each parameter range is divided in 12,096 equal intervals that each contain at least one point). This technique has the advantages of a stratified sampling technique, while keeping the simplicity and objectivity of a pure random sampling technique (Helton and Davis, 2003). We will emphasize on this aspect in section 2.6.

Finally, we want to point out that the posterior distributions from our approach using a simple Monte Carlo technique and a Latin Hypercube Sampling scheme are naturally more likely to appear less constrained than when using Markov-chain-based algorithms such as DREAM (Vrugt, 2016) or PEST (Doherty and Johnston, 2003). This is a visual effect. Our approach is similar to a global optimizer that tries to find the absolute optimum point by exploring the widest space as evenly as possible (especially when using LHS), say [0, 1] to make it simple. In contrast, Markov Chain Monte Carlo algorithms tend to quickly converge on "interesting areas" (say [0.05, 0.1]) and tend to stay confined there on several local optima. This means that the resulting posteriors appear naturally more constrained with MCMC algorithms because they only show values in the explored region of interest, say [0.05, 0.1], out of the total initial space ([0, 1]). We could not use MCMC algorithms for numerical reasons. For example, MCMC algorithms are poorly suited to systematically enforce parameter constraints (such as the sum of SAS function weights λ being 1).

R2: Lastly, the poor performance of the model leads me to think that maybe their model structure is not adequate, and any discussion based on the model results should be conducted more carefully. It is clear that the model fails to reproduce short time-scale dynamics. Figure 4 shows that their $^2$H-based model cannot capture the observed large fluctuation. It seems that the large fluctuation is, in part, due to the high correlation between $C_{p,2}$ and $C_{Q,2}$ especially when the system is dry, and It implies that short time-scale dynamics are not captured by the model (as they mentioned in Lines 512-513). The fluctuation seems much more pronounced in the $^2$H time series. Thus, if we have a better model that captures the short time-scale

dynamics, it may contradict the authors' argument in Line 472: "stable and radioactive isotopes have information in common about young water."

Authors: Please see our related answer about model performance above. We will stress in the discussion that a better model may change the interpretation of the results to some extent. We don't think that a model performing better would change the conclusions of our study. Furthermore, in our model, the flashy events (that we assume to be young water contributions) are conceptualized in a novel way via $\lambda_1$ and its parameterization depending both on storage and a proxy of storage variations. In the discussion of the original manuscript, we proposed suggestions for improvement in future studies regarding this part of the model (Lines 518-538). Yet, we disagree with R2. Behavioral simulations were able to match the flashy dynamics of $\delta^2 H$ to a degree. We will supply figures as a supplement (see above) that will allow the readers to visually identify this aspect better (see one example below). As R2 points out, these flashy events occur mostly during drier periods, but not only. During winter 2016, flashy variations in $\delta^2 H$ can also be observed (figure 4 of the original manuscript). The flashy variations tend to follow the variations of precipitation $\delta^2 H$, and suggest the influence of young water contributions to the stream. However there is not a perfect correlation between $C_{P,2}$ and $C_{Q,2}$, even during dry periods (e.g. for Q < 0.02 mm/h) when the relationship seems visually clearer. This is most likely because of a strong annual groundwater contribution, conceptualized with the two gamma components in the SAS function (Rodriguez and Klaus, 2019). $C_{P,2}$ can thus explain only about 45% of the variations of $C_{Q,2}$ during dry periods. We will provide a figure showing this in the supplement of a revised manuscript, and include these comments in the discussion, section 4.4.

[Figure]

Figure: Simulations of $\delta^2 H$ in May-June 2016

The flashy variations appear more pronounced for $\delta^2 H$, because there are many more samples compared to $^3 H$, and because the unit scaling is different. We think that these flashy events would be similar for tritium if we had more than 1000 samples. One of such flashy events was already captured with the 24 samples and can be observed in November 2016 for $^3 H$. Re-scaling the time series to be able to include the precipitation signal (as this was done for tritium in figure 5) makes the flashy events appear much less pronounced. For instance, compare the inset of figure 2 with figure 4 for $\delta^2 H$. The inset in figure 2 makes the tritium variations appear stronger than deuterium variations. Finally, as we detailed in the discussion (lines 515-517), a model passing through all observation points would still not allow to draw firm conclusions of the "own potentials" of each tracer in terms of water ages, because the number of samples for each tracer is not comparable. We think that high-frequency tritium observations would unambiguously show that young water contributions are as visible in tritium time series than in the $\delta^2 H$ plot (e.g. Crouzet et al., 1970). The point of our work is to argue that there is only one streamflow TTD, and that an observed age difference between the

tracers can be due to sampling limitations in one or the other tracer or to erroneous assumptions (e.g. steady-state). We will insist further on this point in section 4.4 of the discussion.

R2: The use of the Kullback-Leibler Divergence DKL in the hypothesis test seems inappropriate. Throughout the manuscript, the authors stated that using both tracers together is valuable since DKL > 0 (e.g., in Lines 435-436 and Lines 468-470). However, the criterion DKL > 0 cannot determine whether the criterion is met because multiple tracers are used or because there is just any additional information. For example, DKL between the model constrained by, let's say, 100 $^2$H data and the model constrained by the rest of the $^2$H data will be greater than zero.

Authors: This is an interesting point. However, it is not only because DKL > 0 that we concluded that using both tracers together is valuable. As stated lines 436-437, using both tracers together reduced the entropy of the posterior distributions compared to prior distributions. Combining both tracers also allowed narrower groups of TTD curves in figure 6 and 7, and yielded lower standard deviations of the age and storage measures in table 3 and 4 despite having fewer samples. Second, DKL is strictly positive if and only if the compared probability distribution functions (pdfs) differ, meaning that they contain different information about the population(s) they describe. It does not matter for DKL whether the pdfs come from sampling different populations (in our case the posteriors constrained either by one tracer or two tracers) or from sampling the same population several times with different methods (e.g. using two distinct objective functions to constrain the parameters using only one tracer). In any case, DKL being strictly positive tells us that the posteriors are not equal, thus we learned something about the parameters and the water ages. The statement "DKL between the model constrained by, let's say, 100 $^2$H data and the model constrained by the rest of the $^2$H data will be greater than zero" may unfortunately be wrong. If the additional $\delta^2$H data points do not visibly change the posterior pdfs compared to the initial 100 points, meaning that they do not bring considerably more information about the parameters hence the water ages, DKL can be close or equal to 0. We found DKL values about 10 times smaller than the maximum Shannon entropies corresponding to uniform prior distributions (table 2). This roughly 10% additional knowledge gained by adding one tracer is therefore not negligible. We will add these comments in section 4.3.

R2: Moreover, different performance measures were used for their models (Lines 265-270), and it makes the use of DKL even more inappropriate. The authors used the NSE for the $^2$H-based model and used the MAE for the $^3$H-based model. Thus, the difference between the posterior distributions estimated by those behavioral models can be, in part, explained by the choice of performance measure. For example, if the authors estimate the posterior distributions using the $^2$H dataset based on the MAE, the posterior distributions would differ from those estimated based on the NSE. Then, DKL would be greater than zero. Thus, it is not hard to follow their argument that using both tracers together is valuable (e.g., in Lines 331-333, Lines 435-436, Lines 478-470, and Lines 580-581).

Authors: This is also an interesting remark. We therefore conducted additional analyses. Before we answer this comment, we want to mention that these additional analyses helped us realize that we mistakenly multiplied all the values in table 2 by $\log_2(10)$. This means that we will correct all the values shown in table 2 and mentioned in the text by dividing them by $\log_2(10)$. It is important to notice that this changes absolutely nothing to all the reasoning we applied and to what we wrote in the manuscript, since the values are all changed by exactly the same proportionality factor.

Following R2's suggestions, we recalculated table 2, using the criteria MAE < 1.3‰ for $\delta^2$H and MAE < 0.5 T.U. for $^3$H. We used the threshold 1.3‰ for deuterium to obtain a similar number of behavioral simulations (here, 149) than with NSE > 0 (148 solutions). We obtained similar results than for NSE > 0 and MAE < 0.5 T.U. Only minor differences can be observed for some parameters. We carefully checked and found that all our reasoning and our conclusions based on table 2 remain intact (lines 321-328 and discussion section 4.3). We will nevertheless include these additional results in the supplement. Following R2's comment that DKL would be greater than 0 if we used both MAE and NSE constraints on $\delta^2$H, we went further and calculated the DKL between posteriors constrained by NSE > 0 or by MAE < 1.3‰ and posteriors constrained by the combination {NSE > 0 and MAE < 1.3‰}. All the DKL values we found were below 0.02 bits. This information gain is negligible compared to what was learned by adding one tracer after another. It is not a

surprise because the NSE and the MAE are both based on minimizing a sum of residuals (squared or not), making them almost equivalent. It would be very different if we used a measure based on residuals and another based for example on a correlation measure (Legates and McCabe, 1999). Thus, in our case, the use of the DKL clearly shows that the information gain is not due specifically to the choice of distinct objective functions for $^2$H and $^3$H, but instead to the additional information contained in the other tracer.

R2: Furthermore, I disagree with their cost analysis (in Lines 445-451), which led them to conclude that $^3$H tracer is more cost-effective (e.g., Line 17). As described in Lines 462-463, "The amount of information learned from the isotopic data probably scales nonlinearly and probably reaches a plateau as the number of observation points grows." However, they assumed "linearity" in their cost analysis. Thus, the analysis is not valid.

Authors: We thank R2 for this remark. The reviewer is right, that we would (most likely) not have concluded that tritium is more cost-effective, if we had more samples and if these samples did not bring more information about parameters and water age. The lines above the quoted statement (lines 458-462) and the conclusion (lines 574-575) also say that $^2$H could have been more cost-effective with a smarter sampling, which could reduce the number of δ$^2$H samples hence the total analytical price. We will anyway remove the parts of the manuscript that mentioned cost-efficiency, to avoid misinterpretations.

Finally, we only hypothesized that "The amount of information learned from the isotopic data probably scales nonlinearly and probably reaches a plateau as the number of observation points grows". We will rewrite this sentence to make this clearer. We do not know if there is linearity or not. The only thing we know with our samples are the two points shown in the figure below (that we will include in the supplement). How information scales with the number of samples could be any of the dashed curves that represent very different scenarios. The other thing we are sure of is that the true curve can never decrease: there is no information lost by adding new samples. In the worst case, nothing is learned, and the information gain is 0. This means that no matter how many tritium samples we add in our case, tritium will always stay more informative in the absolute sense (14.85 > 13.55) than deuterium. We will thus only keep our statement that tritium was overall more informative.

[Figure]

Figure: Information learned about water ages from each tracer (points) and potential relationships between the number of samples and the (necessarily) growing information content (dashed lines).

By simply dividing the total amount of information by the number of samples or by the total analytical price, we only applied some sort of normalization that does not assume linearity or nonlinearity. It would be different if we used a normalized value (e.g. 0.619 bits per sample for tritium) to extrapolate how much information we could learn in the future by gathering more samples. This would correspond to drawing the unknown curves towards the right-hand side of the points in the figure above. We did not test in what way the amount of information grows with increasing number of samples, as detailed lines 463-467. We will rewrite this part to make sure this is clear, and so that future studies may look into this aspect. As we also detailed in our reply to FG, this test would introduce some subjectivity because not only the number of samples that is used would matter for this analysis, but also the way those samples would be selected among all that we have.

R2: Lastly, it seems that the ET SAS functions are very important in this study but rarely explained. One of its parameters, $\mu_{ET}$ is the most valuable parameter in terms of the information gain in this study (see Table 2). However, no explanation is provided why it is the most valuable and how it affects their interpretation of the results. For example, Figure 5 is one of the most important figures that clearly illustrates the difference between the $^2$H-based model and the $^3$H-based model. The simulated $^3$H concentration using the $^2$H-based model, in general, is higher than that simulated using the $^3$H-based model. It means that tracer mass partitioned into discharge is smaller in the $^3$H-based model during the period. Since there is no explanation on the difference, I had to guess that either more $^3$H tracer mass is stored in the system in the $^3$H-based model or more $^3$H tracer was partitioned into evapotranspiration in the model. Overall, it seems that the partitioning is one of the most important differences between the two models. Thus, the partitioning should be explained in more detail

Authors: We thank R2 for this excellent remark. ET is critical for travel time studies. The water mass balance reads:

$$dS/dt = J - Q - ET$$

In the study we only have one water partitioning condition in the model and that is to decrease ET from PET to 0 when storage S drops below a certain threshold ($S_{root}$) (see appendix A2). This threshold conceptualizes the strongly increasing capillary forces that prevent water from being taken up by plant roots or directly evaporated at lower soil water contents (Rodriguez and Klaus, 2019). A similar strategy was employed for instance by Fenicia et al. (2016) and Pfister et al. (2017) in the Weierbach and neighboring Luxembourgish catchments. The choice of the SAS functions $\Omega_Q$ and $\Omega_{ET}$ has only an indirect link with the isotopic partitioning of J between Q and ET. The SAS functions represent only a preference of a given outflow for certain stored water ages. Since there is no one-to-one relationship between the stored water age at a given moment and the past tracer concentrations in the input (e.g. the age ambiguity of tritium, see figure 2), there is no explicit partitioning of isotopic concentrations in the model based on the SAS functions. We will add these details to the methods (2.4) and to appendix A2.

We did not focus on the parameters of the SAS function of ET because our study deals with streamflow travel times, and because we do not have tracer data representative of the ET flux that could be used to directly constrain its SAS function parameters. Instead, we indirectly constrained these parameters to the tracer data in streamflow. Similar to Van der Velde et al. (2015) and Visser et al. (2019), we found that the parameters of the ET SAS function have a non-negligible influence on the simulations of stream isotopic tracers. We agree with R2 that this relative importance of $\mu_{ET}$ was observed because of the long term isotopic partitioning of precipitation between streamflow and ET. We will include the figure below in the supplement. It shows, as suggested by R2, that the simulations constrained by $^2$H generally yielded more tritium mass in streamflow over 2015-2017 than the simulations constrained by $^3$H.

[Figure]

Figure: Simulated flow-weighted concentrations in the stream for the behavioral model runs constrained by deuterium samples or by tritium samples.

As R2 points out, this means that for the [3]H-based model, more tritium was stored, or ET removed more tritium from storage compared to the [2]H-based model (or both effects together). We do not have the necessary tracer observations (such as isotope samples in ET or isotope samples representative of storage) to say what mechanism happened in the catchment. In that instance, we cannot determine if the model used the correct mechanism or not. However, we can discriminate the solution based on the long term isotopic mass balance.

Tritium accumulation in modelled storage to momentarily decrease $C_Q$ is only a short-term solution, because the stored tritium concentration cannot continuously increase in a physically realistic model. The only solution to reduce the stream tritium content in the long term (e.g. over 2 years like here, or longer) is to evacuate the excess tritium by ET. The posterior distribution of $\mu_{ET}$ constrained by tritium observations (see above) tends to lower values, indicating a stronger preference for younger water in ET compared to $\mu_{ET}$ constrained by deuterium observations. If we restrict our point of view from years 2000 to 2017, current precipitation generally has a higher tritium content than the water recharged before (see figure 2). Thus, by preferentially removing younger water, ET partly contributes to removing tritium from the system and to keeping the simulated stream tritium concentrations low over 2015-2017. This is why the information gain about $\mu_{ET}$ is so high with tritium data. It is interesting to see that the same mechanism must be occurring with stable isotopes because the information gain about $\mu_{ET}$ is also high with stable isotopes, and the figure above shows that behavioral solutions for deuterium also have a lower stream tritium content than current precipitation.

We think that the lack of high-resolution tritium data explains why the simulations constrained by tritium observations tend to have a lower stream tritium content than simulations constrained by stable isotopes. On the one hand, with only monthly measurements of precipitation taken 60 km away from the study site, our knowledge of the true tritium content of local precipitation has some uncertainty. It is possible that we overestimate the flux-weighted tritium concentration of precipitation (see the red cross in the figure above). The same remark applies to the stream tritium content. The 24 samples probably do not fully represent the flux-weighted tritium concentration in the stream. It is thus possible that we underestimate the true value, and that more samples during hydrological events (such as flashy peaks) would increase the estimated value. We will condense and add this information to the results, section 3.1. We also think that this really points to a

critical limitation in many hydrological studies: the lack of appropriate sampling schemes for tracers in ET in space and time.

R2: Line 375: Typo in "[0,∞["

Authors: We think that R2 means that the open squared bracket "[" should be a parenthesis "(". If that is the case, we observed that both notations exist, and we prefer to keep the one already used. If that is not the case, we are sorry but we do not see the typo.

R2: Line 224: It is stated that $\lambda_1(t)$ is the smallest weight. However, it is not clear how that was constrained in the model calibration.

Authors: Essentially, $\lambda_1(t) < \lambda_1^*$ and $\lambda_1^*$ is sampled between 0 and $1-\lambda_2$ (hence between 0 and 1) to have $\lambda_1+\lambda_2+\lambda_3=1$ (table 1 footnotes). This means that $\lambda_1^*$ is sampled more often close to 0 than close to 1, and $\lambda_1(t)$ is generally the smallest weight. We did this because large values of $\lambda_1(t)$ generally corresponded to poor simulation fits in initial tests, and because it is necessary to impose at least one relationship between two $\lambda$ coefficients to be able to randomly select three $\lambda$ verifying $\lambda_1+\lambda_2+\lambda_3=1$. We will add more details about this and rephrase the sentence to avoid misinterpretations.

R2: Lines 236-237: $S_{ref}$ is chosen not calibrated, so probably introducing the chosen value here would be better, rather than introducing it in the next section, 2.6 Model calibration.

Authors: We will do as suggested.

R2: The initial condition for the SAS function model is not described. If there was a spin-up for the SAS function model (like the storage estimation), what tracer time series were used?

Authors: We will add this information to the paragraph.
We detailed in section 2.2 that we periodically looped back the 2010-2015 input data to create the spin-up time series (1915-2015). The initial condition corresponds to an exponential distribution of residence times (RTD) with a mean of 1.7 years. The initial SAS functions and TTDs are then calculated based on their chosen functional form and their parameters, using this initial RTD.

R2: Lines 404-405: How this comparison between 2016 finding and 2017 finding helps readers to understand the higher age estimated using the $^3$H-based model?

Authors: We will rewrite this sentence to make it clearer.

R2: Lines 437-439: Those parameters are not independent. Thus, those were not independently constrained.

Authors: What we really meant is that the shapes of the components of the streamflow SAS function were constrained independently. The only imposed relationship between the parameters of three components is $\lambda_1+\lambda_2+\lambda_3=1$. This does not affect their shape, nor their location on the age axis (or age-ranked storage axis). We will rewrite the sentence to reflect this better.

Authors: We noticed a typo in figures 4 and 6, where we wrote 141 behavioral simulations in deuterium instead of 148 (correct value, as stated in the text). We will correct this.

Cartwright, I., & Morgenstern, U. (2016a). Contrasting transit times of water from peatlands and eucalypt forests in the Australian Alps determined by tritium: implications for vulnerability and the source of water in upland catchments. Hydrology and Earth System Sciences, 20, 4757-4773. doi:10.5194/hess-20-4757-2016.

Cartwright, I., & Morgenstern, U. (2016b). Using tritium to document the mean transit time and sources of water contributing to a chain-of-ponds river system: Implications for resource protection. Applied Geochemistry, 75, 9-19, http://dx.doi.org/10.1016/j.apgeochem.2016.10.007.

Crouzet, E., Hubert, P., Olive, P., Siwertz, E., Marce, A., 1970. Le tritium dans les mesures d'hydrologie de surface. Determination experimentale du coefficient de ruissellement. J. Hydrol. 11 (3), 217–229.

Doherty, J. and Johnston, J.M. (2003), METHODOLOGIES FOR CALIBRATION AND PREDICTIVE ANALYSIS OF A WATERSHED MODEL. Journal of the American Water Resources Association, 39: 251-265. doi:10.1111/j.1752-1688.2003.tb04381.x

Duvert, C., Stewart, M. K., Cendón, D. I., and Raiber, M.: Time series of tritium, stable isotopes and chloride reveal short-term variations in groundwater contribution to a stream, Hydrology and Earth System Sciences, 20, 257–277, https://doi.org/10.5194/hess-20-257-2016, 2016

Ehret, U. and Zehe, E.: Series distance – an intuitive metric to quantify hydrograph similarity in terms of occurrence, amplitude and timing of hydrological events, Hydrol. Earth Syst. Sci., 15, 877–896, https://doi.org/10.5194/hess-15-877-2011, 2011.

Fenicia, F., D. Kavetski, H. H. G. Savenije, and L. Pfister (2016), From spatially variable streamflow to distributed hydrological models: Analysis of key modeling decisions, Water Resour. Res., 52, doi:10.1002/2015WR017398.

Gabrielli, C. P., Morgenstern, U., Stewart, M. K., & McDonnell, J. J. (2018). Contrasting groundwater and streamflow ages at the Maimai watershed. Water Resources Research, 54, 3937–3957. https://doi.org/10.1029/2017WR021825

Gallart, F., Roig-Planasdemunt, M., Stewart, M. K., Llorens, P., Morgenstern, U., Stichler, W., Pfister, L., and Latron, J.: A GLUE-based uncertainty assessment framework for tritium-inferred transit time estimations under baseflow conditions, Hydrological Processes, 30, 4741–4760, https://doi.org/10.1002/hyp.10991, 2016.

Gusyev, M. A., Morgenstern, U., Stewart, M. K., Yamazaki, Y., Kashiwaya, K., Nishihara, T., Kuribayashi, D., Sawano, H., and Iwami, Y.: Application of tritium in precipitation and baseflow in Japan: a case study of groundwater transit times and storage in Hokkaido watersheds, Hydrol. Earth Syst. Sci., 20, 3043–3058, https://doi.org/10.5194/hess-20-3043-2016, 2016.

Helton, J. and Davis, F.: Latin hypercube sampling and the propagation of uncertainty in analyses of complex systems, Reliability Engineering & System Safety, 81, 23 – 69, https://doi.org/10.1016/S0951-8320(03)00058-9, 2003

Hubert, P., Marin, E., Meybeck, M., Ph. Olive, E.S., 1969. Aspects Hydrologique, Géochimique et Sédimentologique de la Crue Exceptionnelle de la Dranse du Chablais du 22 Septembre 1968. Arch. Sci (Genève) 3, 581–604.

Legates, D. R., and McCabe, G. J. (1999), Evaluating the use of "goodness‐of‐fit" Measures in hydrologic and hydroclimatic model validation, *Water Resour. Res.*, 35( 1), 233– 241, doi:10.1029/1998WR900018.

Maloszewski, P., and Zuber, A. (1993). Principles and practice of calibration and validation of mathematical models for the interpretation of environmental tracer data in aquifers. Advances in Water Resources, 16: 173-190

Morgenstern, U., Stewart, M. K., and Stenger, R.: Dating of streamwater using tritium in a post nuclear bomb pulse world: continuous variation of mean transit time with streamflow, Hydrol. Earth Syst. Sci., 14, 2289–2301, https://doi.org/10.5194/hess-14-2289-2010, 2010.

Pfister, L., Martínez-Carreras, N., Hissler, C., Klaus, J., Carrer, G. E., Stewart, M. K., and McDonnell, J. J.: Bedrock geology controls on catchment storage, mixing, and release: A comparative analysis of 16 nested catchments, Hydrological Processes, 31, 1828–1845, https://doi.org/10.1002/hyp.11134, 2017.

Schoups, G., and Vrugt, J. A. ( 2010), A formal likelihood function for parameter and predictive inference of hydrologic models with correlated, heteroscedastic, and non‐Gaussian errors, *Water Resour. Res.*, 46, W10531, doi:10.1029/2009WR008933.

Stewart, M. K. and Fahey, B. D.: Runoff generating processes in adjacent tussock grassland and pine plantation catchments as indicated by mean transit time estimation using tritium, Hydrol. Earth Syst. Sci., 14, 1021–1032, https://doi.org/10.5194/hess-14-1021-2010, 2010.

Stewart, M. K., Mehlhorn, J., and Elliott, S.: Hydrometric and natural tracer (oxygen-18, silica, tritium and sulphur hexafluoride) 900 evidence for a dominant groundwater contribution to Pukemanga Stream, New Zealand, Hydrological Processes, 21, 3340–3356, https://doi.org/10.1002/hyp.6557, 2007.

Stewart, M. K. and Thomas, J. T.: A conceptual model of flow to the Waikoropupu Springs, NW Nelson, New Zealand, based on hydrometric and tracer (18O, Cl,3H and CFC) evidence, Hydrology and Earth System Sciences, 12, 1–19, https://doi.org/10.5194/hess-12-1-2008, 2008.

Uhlenbrook, S., Frey, M., Leibundgut, C., and Maloszewski, P.: Hydrograph separations in a mesoscale mountainous basin at event and seasonal timescales, Water Resources Research, 38, 31–1–31–14, https://doi.org/10.1029/2001WR000938, 2002.

Vrugt, J. A. (2016). Markov chain Monte Carlo simulation using the DREAM software package: Theory, concepts, and MATLAB implementation. Environmental Modelling & Software, 75, 273-316. http://dx.doi.org/10.1016/j.envsoft.2015.08.013

---

## Referee Report (RR1)

The new version of the manuscript entitled 'A comparison of catchment travel times and storage deduced from deuterium and tritium tracers using StorAge Selection functions' by Nicolas B, Rodriguez et al. clearly improves the quality of the previous version in several aspects.

From my point of view, this manuscript may be accepted for publication in HESS if minor changes are done for improving its clarity and scientific soundness.

Major comments:

- In the abstract it is resolutely claimed that deuterium and tritium provided similar aging of waters in Weierbach, but the time span of the results is not stated, so the reader may erroneously understand that this result is valid for any catchment with any MTT value. It is necessary to clearly state there that one of the conditions for this result is that "in catchments with limited residence times, radioactive decay may give information that is redundant with the natural variability of the tracer in precipitation" (line 481).

- line 142: the sentence "The model's ability to simulate stream $^2$H dynamics helped to further confirm that these flow processes are active in the Weierbach" is not acceptable. "Model performances measure the correctness of estimates of hydrological variables generated by the model and not the structural adequacy of the model vis-à-vis the processes being modelled, i.e. the hydrological soundness of the model" (Klemes, 1986).

- Although my opinion is that input and output concentrations should be mass-proportional or -weighted to be processed in a mass balance model, I deem that the methods used in the paper may be acceptable if the way in which concentrations and masses are managed is fully explicit and the possible consequences of the methods used on the results is appropriately discussed.

Indeed, as precipitation samples for $^2$H are taken at fixed precipitation intervals, the resulting concentrations yield the same result than a mass-weighting. But nothing is said about how the bi-weekly bulk samples (time-proportional) are managed and merged with the mass-proportional automatic samples. I do not mind if mass-proportional concentrations are interpolated to produce a 'continuous' signal because the mass is conserved. Furthermore, nothing is said about the $^3$H sampling; were differences in monthly precipitation taken into account to weight input $^3$H activities as usually done?. Therefore, were $^2$H and $^3$H concentrations managed in the same or in different ways (precipitation weighting for $^2$H and time weighting for $^3$H)?.

Respect to stream water sample concentrations, nothing is said in the manuscript but it should be clearly stated whether these were managed as unweighted discrete irregularly taken samples or were time- or flow-weighted. Furthermore, something about flow-weighting the available concentration samples should be included in the interesting discussion in lines 647-652 where the possible advantages of flow-proportional sampling are commented.

-line 562: "Our conclusions are valid because the model captures accurately the travel times in the Weierbach" is really an inappropriate statement. This seems to claim that the model is

hydrologically sound (in the sense of Klemes, 1986) because it reproduces well something that cannot be validated.

Minor comments:

- Line 295. I understand that model efficiency assessment and subsequently the efficiency thresholds for selection of behavioural models might be different for deuterium and tritium, but nothing is discussed afterwards on the possible role of this difference on some of the results obtained. For instance, more parameter sets are accepted as behavioural for tritium than for deuterium; this may be reasonable because sampling is much intensive for deuterium, so model rejection may be stricter for it, but some comment about this issue would be welcome.

- Line 432: As written, the reader may understand that Stewart et al (2010) found or reported travel time differences up to 5 years at Weierbach.

- line 440: "First, we treated $^2$H and $^3$H equally by calculating TTDs using a coherent mathematical framework for both tracers (i.e. same method and same functional form of TTD)" though sampling and model efficiency were differently managed (as discussed later).

- line 524: "Performance measures $E_2$ and $E_3$ are" not identical but are "both based on minimizing a sum of..."

Klemes, V. (1986) Operational testing of hydrological simulation models. Hydrological Sciences Journal, 31 (1), 13-24

---

## Author Response (AR2)

Dear referees, dear editor,

Thank you for this second opportunity to revise and improve our manuscript and for the further suggestions of improvement. Please find below the referees' comments (in black), followed by our answers (in green) and the location of the changes we made in the manuscript (in blue). Unless stated otherwise, the line numbers we indicate are with respect to the manuscript with tracked changes, not the final revised manuscript. Our answers are followed by the manuscript showing the changes in the text, underscored in blue.

Regards

Nicolas Rodriguez, on behalf of all authors

**Reviewer n°1 (Francesc Gallart, FG):**

FG: The new version of the manuscript entitled 'A comparison of catchment travel times and storage deduced from deuterium and tritium tracers using StorAge Selection functions' by Nicolas B, Rodriguez et al. clearly improves the quality of the previous version in several aspects. From my point of view, this manuscript may be accepted for publication in HESS if minor changes are done for improving its clarity and scientific soundness.

We thank Francesc Gallart for taking again the time to review our work and for providing thoughtful suggestions of improvement.

FG: In the abstract it is resolutely claimed that deuterium and tritium provided similar aging of waters in Weierbach, but the time span of the results is not stated, so the reader may erroneously understand that this result is valid for any catchment with any MTT value. It is necessary to clearly state there that one of the conditions for this result is that "in catchments with limited residence times, radioactive decay may give information that is redundant with the natural variability of the tracer in precipitation" (line 481).

Good remark. However, we disagree with FG's above statement suggesting that this agreement between the tracers will be true only in catchments with "short" MTTs. We never proved this, and this is why we simply stated that there can be redundant information between the tracers when travel times are limited (which does not imply anything regarding catchment with longer travel times). Our opinion remains that if the true (unknown) catchment TTD contains a large fraction of very old water (e.g., > 10 years) and if the methods (tracer sampling, model, numerics…) are adequate (as in our study), then both deuterium and tritium should be able to suggest solutions (TTDs) with long tails which yield a good fit to the tracer output time series. Of course, in that case, tritium will likely result in smaller uncertainties than deuterium (i.e., likely more solutions with long TTD tails) because radioactive decay will more strongly discriminate TTD solutions with a short tail from those with a long tail. This is why tracers have different information contents on travel times, and not necessarily different travel times. Throughout the manuscript, we have been careful to clearly distinguish these two concepts. There is nothing that allows one to say that a priori, deuterium will absolutely not allow TTD solutions with long tails compared to tritium, which is what previous studies tended to suggest whereas it contradicts the physics underlying the transport of water towards an outlet.

 We thus simply added:

"The streamflow mean travel time was estimated at 2.90±0.54 years using $^2$H and 3.12±0.59 years using $^3$H (mean ± one standard deviation). Both tracers consistently suggested that less than 10% of stream water in the Weierbach is older than 5 years."

and we finished the abstract with:

"In the future, it would be useful to similarly test the consistency of travel time estimates and the potential differences in travel time information contents between those tracers in catchments with other characteristics or with a considerable fraction of stream water older than 5 years, since this could emphasize the role of the radioactive decay of tritium for discriminating younger from older water."

FG: line 142: the sentence "The model's ability to simulate stream $^2$H dynamics helped to further confirm that these flow processes are active in the Weierbach" is not acceptable. "Model performances measure the correctness of estimates of hydrological variables generated by the model and not the structural adequacy of the model vis-à-vis the processes being modelled, i.e. the hydrological soundness of the model" (Klemes, 1986).

We changed this sentence to:

"The model based on travel times presented in this study was developed in a step-wise manner based on this hypothesis of streamflow generation, and the consistency between simulated and observed δ²H points toward a robust representation of the key processes."

The last sentences of that paragraph now read:

"Other studies carried out in the Colpach catchment (containing the Weierbach) suggested that first peaks are caused by lateral subsurface flow through a highly conductive soil layer and that second peaks are caused by groundwater flow in the bedrock (Angermann et al., 2017; Loritz et al., 2017). This is contrary to the conclusions from other studies in the Weierbach (Glaser et al., 2016; 2019), **showing that the key processes are still under debate**."

FG: Although my opinion is that input and output concentrations should be mass-proportional or-weighted to be processed in a mass balance model, I deem that the methods used in the paper may be acceptable if the way in which concentrations and masses are managed is fully explicit and the possible consequences of the methods used on the results is appropriately discussed.

Indeed, as precipitation samples for ²H are taken at fixed precipitation intervals, the resulting concentrations yield the same result than a mass-weighting. But nothing is said about how the bi-weekly bulk samples (time-proportional) are managed and merged with the mass proportional automatic samples. I do not mind if mass-proportional concentrations are interpolated to produce a 'continuous' signal because the mass is conserved. Furthermore, nothing is said about the ³H sampling; were differences in monthly precipitation taken into account to weight input ³H activities as usually done? Therefore, were ²H and ³H concentrations managed in the same or in different ways (precipitation weighting for ²H and time weighting for ³H)?

Respect to stream water sample concentrations, nothing is said in the manuscript but it should be clearly stated whether these were managed as unweighted discrete irregularly taken samples or were time- or flow-weighted. Furthermore, something about flow-weighting the available concentration samples should be included in the interesting discussion in lines 647-652 where the possible advantages of flow-proportional sampling are commented.

We added many more details in section 2.2 (see lines 167-170, 188-192, 195-203), and some details in sections 2.3 and 2.4 (see lines 213-214, 217-220, 242-245) to try and remove any remaining ambiguity about concentration weighting. Precipitation samples do not need to be weighted by precipitation amounts. They all are representative of the tracer mass flux in precipitation by design since they all are cumulative samples (and not based on fixed time intervals), and because we calculated the input concentration signals to make sure that the integral of [concentration times precipitation amounts] over a given period is always equal to the total tracer entering the catchment over that period. Simply stated: $C_{model\_input} \times \sum Precip = true \sum (C_{precip} \times Precip)$.

Stream samples are all instantaneous grab samples, and there is no need to weight them by streamflow because the time-varying TTDs already account for the fractions of precipitation water not reaching the stream due to ET or storage and for time-varying discharge rates, unlike the steady-state TTDs which require flow-weighting (see lines 217-220).

We also added these sentences in the discussion (lines 697-700):

"It is important to notice that weighting the available stream samples by streamflow in the calibration (i.e., calibrating on tracer loads instead of concentrations) would not compensate for this relative absence of samples during high flow conditions. In addition, it would bias the

calibrated TTDs towards high flow conditions, while our goal is to have TTDs which accurately represent the functioning of the catchment over all flow conditions (the whole 2015-2017 study period)"

FG: line 562: "Our conclusions are valid because the model captures accurately the travel times in the Weierbach" is really an inappropriate statement. This seems to claim that the model is hydrologically sound (in the sense of Klemes, 1986) because it reproduces well something that cannot be validated.

Good remark, we changed this to:

"The visually satisfactory tracer simulations enhance our confidence that the model accurately simulates travel times in the Weierbach."

FG: Line 295. I understand that model efficiency assessment and subsequently the efficiency thresholds for selection of behavioural models might be different for deuterium and tritium, but nothing is discussed afterwards on the possible role of this difference on some of the results obtained. For instance, more parameter sets are accepted as behavioural for tritium than for deuterium; this may be reasonable because sampling is much intensive for deuterium, so model rejection may be stricter for it, but some comment about this issue would be welcome.

We added this (lines 628-635):

"Our choice of performance measures ($E_2$=NSE and $E_3$=MAE) and selection criteria ($L_2$=0 and $L_3$=0.5) resulted in slightly more TTDs constrained by tritium than TTDs constrained by deuterium (148 curves for $E_2 > 0$ against 181 curves for $E_3 < 0.5$). These numbers are highly sensitive to performance thresholds, and our choices represent the closest match in the number of accepted solutions for each tracer, while considering only meaningful performance criteria variations (i.e., ≥ 0.1) and acceptable model performance. This guarantees a similar treatment of the two tracers (i.e. it avoids biases in travel times for a given tracer), while accepting only satisfying simulations for both tracers. Future work could assess the sensitivity of travel time differences between tracers for other performance measures and thresholds, and for contrasting numbers of accepted solutions."

FG: As written, the reader may understand that Stewart et al (2010) found or reported travel time differences up to 5 years at Weierbach.

We corrected this to:

"The TTDs obtained from each tracer were broadly consistent in shape, and the travel time differences were considerably smaller (i.e., <1 yr) **in the Weierbach** than in a previous comparison study **in four catchments from Germany and New Zealand** (up to 5 yr, Stewart et al., 2010)"

FG: line 440: "First, we treated 2H and 3H equally by calculating TTDs using a coherent mathematical framework for both tracers (i.e. same method and same functional form of TTD)" though sampling and model efficiency were differently managed (as discussed later).

Good suggestion. This now reads:

"First, we treated $^2$H and $^3$H equally by calculating TTDs using a coherent mathematical framework for both tracers (i.e. same method and same functional form of SAS function). However, sampling frequency and model efficiency criteria needed tracer-specific adaptations (see Sect. 4.4.2 and 4.4.3)."

FG: line 524: "Performance measures E2 and E3 are" not identical but are "both based on minimizing a sum of…"

Changed as suggested.

**Reviewer n°2 (R2):**

R2: Thank the authors for taking the time to respond to the previous comments and revise the manuscript. However, there are several important points that I still can't agree with the authors. Also, I still think that the authors miss some critical points and rather rely too heavily on their model results to support their conclusions. Those points obscure what readers can learn from this study. I believe that this manuscript still has potential, but some issues need to be resolved before it can be considered for publication.

We thank R2 for the additional time spent on reviewing our work and for the additional comments. We agree with the reviewer that some of our statements were not nuanced enough and thus underrated some critical learning for the reader. We modified the manuscript accordingly at most instances or reasoned more clearly when we did not.

R2: THE LIMITATIONS OF THE DATA AND THE STUDY SITE

I noticed that the authors mentioned many limitations of this study in one section pretty well, but there are some crucial limitations that can obscure their major arguments significantly. I list some of those limitations here with my opinions.

Figure 3 shows that there are no tritium samples at high flow conditions, and thus, one cannot learn transport dynamics at the high flow conditions using $^3$H no matter which model is used. Thus, I believe that any arguments based on the $^3$H-based model results at high flow conditions (e.g., the TTDs at high flow conditions) are risky because such results are just based on "extrapolations" that the model did. In their result interpretation, the authors mostly used the TTD weighted by discharge, which is in part based on the TTDs estimated at high flow conditions and even gives more weights to those TTDs. Thus, many (and most) of the arguments in the abstract and the conclusion such as "Tritium and stable isotopes both had the ability to reveal short travel times in streamflow", "The travel time differences were small compared to previous studies, and contrary to prior expectations, we found that these differences were more pronounced for young water than for old water", "our results highlight that stable isotopes and tritium have different information contents on travel times but they can still result in similar TTDs.", and "We conclude that stable isotopes do not seem to systematically underestimate travel times or storage compared to tritium" are based on the extrapolation. Drawing scientific conclusions based on extrapolation is risky and not a good practice.

We understand R2's concerns, but we disagree with R2's above claims. Figure 3 shows several samples at high flow conditions (4 samples out of 24 correspond to flows exceeded around 10% of the time only, i.e. to Q > 0.1 mm/h, see Figure 5 and 6), even if they do not represent the highest flows recorded (as pointed to by Francesc Gallart in the last round of revisions, see reply to his previous comments). However, and more importantly, this comes down to a philosophical question whether we can use a model to derive some conclusions or if we need to fully rely on data sets that completely sample all occurring flow stages and scenarios. This of course does not only include sampling along the whole FDC, but all sorts of combinations between flow and antecedent conditions including hysteresis between storage and flow as well as the ET regime. For us it is evident from the literature that "extrapolating" between a limited number of data points using a model is a ubiquitous practice in time-varying travel time studies (see Benettin et al., 2015a; 2017a; Birkel et al., 2015; Harman, 2015; Heidbüchel et al., 2012; Hrachowitz et al., 2013; Klaus et al., 2015, Rodriguez et al., 2018, 2019, 2020). This is especially true for the previous (steady-state) travel times studies using tritium (e.g., Maloszewski & Zuber, 1982, 1993; Gallart et al., 2016), which worked with a much smaller number of tritium samples. In fact, some studies have less than 5 points over several years of observation. As argued by Francesc Gallart, not many

travel time studies tended to report their samples along streamflow values as we did in Figure 3, and we can expect that many also had to considerably "extrapolate" (for low or high flows), using their model in order to derive meaningful conclusions on travel times and related hydrological processes. This is somehow opposite to data-based methods (e.g. Kirchner, 2019), which tend to use only the available data to draw conclusions. As a result, data-based approaches as very data-hungry (typically needing tens of thousands of data points) and cannot be used to derive conclusions between data points or outside of observation periods. Calibrating a model to a tracer time series having less than 100 points as we did is helpful to round this issue of limited data availability. We would like to point out again that our tritium data set is one of the densest ever recorded, as stated in one of the sentences we added to the manuscript in the previous revision (see lines 179-181). However, additional data or different tritium data sets may confirm or challenge our current findings in the future.

We agree that using flow-weighted TTDs will accentuate the role of TTDs at high flows. But time-averaged TTDs will also contain the instantaneous TTDs at high flows. Therefore, there is no obvious solution to avoid using the "extrapolations". The current travel time literature does not contain much guidance on what TTD to use in the analyses for what purpose (i.e. flow-weighted, or time-averaged). We believe that by default, flow-weighted TTDs should be used as they allow a more meaningful comparison between catchments with contrasting flow regimes. For example, the same time-averaged TTD for two contrasting catchments (say, ephemeral stream vs wetland) could hide very different catchment functionings which are better revealed by their different flow-weighted TTDs. If high frequency tritium data sets are available in the future, it would be very helpful to compare the TTDs from high-frequency data with the TTDs obtained by re-sampling a coarser time series (see relevant comments on this lines 556-560), and to see whether time-averaged TTDs yield smaller differences between a full data set and a limited data set than flow-weighted TTDs.

Finally, as explained in section 4.4.3 (lines 703-705): "the larger water mass not sampled for tritium is not leading to a strong bias towards young or old water compared to deuterium. The latter is shown by the good agreement between the TTDs constrained by deuterium and the TTDs constrained by tritium.", and it is worth keeping in mind that the deuterium data set is representative of the higher flow conditions (see Figure 3).

R2: Another problem is that travel time is relatively short in this catchment. It has been argued that the tritium is beneficial as it allows us to examine long time-scale transport dynamics (e.g.,> ~4 years in Stewart et al., 2010).

Again, we understand R2's concern. But first, we would like to stress again that that the statement of Stewart et al. (2010) is true only for very specific conditions (lines 572-576):

"The theoretical span of 0–4 years pointed out in Stewart et al. (2010) should however not be taken as the only range of travel times where $^{18}O$, $^{2}H$, and $^{3}H$ may have redundant information. As clearly written by Stewart et al. (2010), this limit corresponds to a steady-state exponential TTD only, while other TTD shapes (or unsteady TTDs) could yield much higher limits. More importantly, this limit can be lowered by the seasonality of the input function (see Stewart et al., 2010, p. 1647)."

However, in this catchment, the travel time is relatively short in general, and a considerable fraction of TTD (> ~90%) is defined over the travel time less than 5 years (based on Table 3). This short travel time obscures the relative importance of the use of $^{3}H$ to examine longer time scale transport dynamics because longer time scale transport is less important (or negligible) in reproducing the tracer dynamics in this catchment.

We agree that the travel times in our catchment are relatively short and that it may limit the potential of $^3$H for discriminating younger from older water thanks to radioactive decay, as already clearly stated in the previous version of the manuscript (lines 515-520, see below):

"The travel times being below ~5 years in the Weierbach (Table 3) could be another reason for the limited information of $^3$H on older water. $^3$H decays by only about 25 % in 5 years, meaning that all the tritium activities of the water in the Weierbach have varied by at most ~2 T.U. since water entered the catchment. This is much lower than the 10 T.U. amplitude of tritium variations in precipitation. Thus, in catchments with relatively short residence times, radioactive decay may give information that is redundant with the natural variability of the tracer in precipitation".

We are not sure about the intention of this specific comment, however the reviewer seems to target defending the use of tritium. We also see great potential in the use of several tracers to derive catchment TTDs. However, as we argued throughout the manuscript -- and clearly supported by the calculated TTDs -- there should not be any physical difference in age derived from two tracers. Here, we are actually able to reconcile (with uncertainty) TTDs from tritium and deuterium when we are using a coherent methodological framework. Yes, this is done for rather short TTDs, but by no means would that be an argument for expecting different TTDs for the same water parcel when the TTDs become longer. However, data sets to study this in catchments with longer travel times are currently not available, and we are looking forward to see them and similar research for a wide range of TTDs.

It is also worth noting again that numerically, we have travel times up to 100 years in the catchment (Figure 7), but also that the fraction of stream water older than about 5 years (~10%) is not negligible, and that this range of travel times is in fact the one on which the tracers agreed the most (table 3, table B1).

Therefore, I worry if the study site is adequate, and I'm not sure about the worth of most of their arguments such as "The travel time differences were small compared to previous studies, and contrary to prior expectations, we found that these differences were more pronounced for young water than for old water", "we did not find that stable isotopes are blind to old water fractions as suggested by earlier travel time studies ", "Based on the results in our experimental catchment in Luxembourg, we conclude that the perception that stable isotopes systematically truncate the tails of TTDs is not valid", "our results highlight that stable isotopes and tritium have different information contents on travel times but they can still result in similar TTDs.", and "We conclude that stable isotopes do not seem to systematically underestimate travel times or storage compared to tritium". Is a similar conclusion expectable for a catchment that has a longer travel time? Or is it just because the studied catchment has short water travel time in general? If the latter is the case, what can be said about the well-known importance of tritium tracer by studying this catchment?

There is nothing like a non-adequate study site for our posed questions. That said, similar studies in several catchments with a range of travel times would be highly appreciated in the future. No research group has tackled this question until today and thus other data sets are not at hand. However, that a water parcel cannot have two different mean travel times is independent of a catchment, and here we provided first insights that a consistent approach reveals similar TTDs and MTTs for both tracers. In the first review, Francesc Gallart had a similar remark about relatively short travel times, and we answered him:

"we disagree with FG's above statement suggesting that this agreement between the tracers will be true only in catchments with "short" MTTs. We never proved this, and this is why we simply stated that there can be redundant information between the tracers when travel times are limited

(which does not imply anything regarding catchment with longer travel times). Our opinion remains that if the true (unknown) catchment TTD contains a large fraction of very old water (e.g., > 10 years) and if the methods (tracer sampling, model, numerics…) are adequate (as in our study), then both deuterium and tritium should be able to suggest solutions (TTDs) with long tails which yield a good fit to the tracer output time series. Of course, in that case, tritium will result in smaller uncertainties than deuterium (i.e., likely more solutions with long TTD tails) because radioactive decay will more strongly discriminate TTD solutions with a short tail from those with a long tail. This is why tracers have different information contents on travel times, and not necessarily different travel times. Throughout the manuscript, we have been careful to clearly distinguish these two concepts. There is nothing that allows one to say that a priori, deuterium will absolutely not allow TTD solutions with long tails compared to tritium, which is what previous studies tended to suggest whereas it contradicts the physics underlying the transport of water towards an outlet."

This is why we added this to the abstract:

"The streamflow mean travel time was estimated at 2.90±0.54 years using $^2$H and 3.12±0.59 years using $^3$H (mean ± one standard deviation). Both tracers consistently suggested that less than 10% of stream water in the Weierbach is older than 5 years."

and we finished the abstract with:

"In the future, it would be useful to similarly test the consistency of travel time estimates and the potential differences in travel time information contents between those tracers in catchments with other characteristics or with a considerable fraction of stream water older than 5 years, since this could emphasize the role of the radioactive decay of tritium for discriminating younger from older water."

We believe that "the well-known importance of tritium tracer" has in fact been overstated in the previous studies, and mistakenly based on data and methodological limitations that we tried to overcome in this study (see the introduction in our manuscript). Starting from the assumption that tritium is the only tracer revealing old water necessarily leads to circular reasoning. For example, one common mistake from the past has been to focus tritium sampling on baseflow, based on the implicit assumption that it is more useful for old water. This of course naturally biased the results towards old water by sampling design (because baseflow does contain older water than hydrographs by definition). This mislead many to think that tritium reveals older water than deuterium. We could similarly declare by mistake that deuterium is more informative on short travel times than tritium if we sampled deuterium only during large hydrological events (as in isotope hydrograph separation) and not tritium (notice that it seems to be the case, currently). Conversely, assuming nothing a priori and treating the tracers as equally as possible resulted in similar TTDs based on deuterium and tritium despite the different tracer treatments imposed by sampling and calibration differences later on in the analysis.

Finally, we do support the idea that tritium is a very useful tracer, as it is more age-specific than deuterium. This is why we wrote (lines 520-526):

"In a few decades, water recharged in 1980–2000 may have completely left the catchments or may be a negligible part of storage, such that the log($^3$H) of stored water may increase linearly with residence time (see the recent increasing trend in $C_{P,3}$* in Fig. 2). Thus in a few decades, tritium could be even more informative about old water contributions because there may be no travel time ambiguity anymore. Furthermore, the oscillations of tritium in precipitation over long time scales (>10 years) recently detected and related to cycles of solar magnetic activity (Palcsu et al., 2018) may give stream tritium concentrations even more age-specific meaning. Therefore, it is

important to re-iterate the call of Stewart et al. (2012) to start sampling tritium in streams now and for the next decades to use it in travel time analyses."

R2: THE MODEL AND THE RESULT INTERPRETATIONS

Again, their model cannot reproduce some short time scale transport dynamics (based on Figure 5 and the low NSE values).

We wonder if R2 saw the figures we added in the supplement, as we are convinced that they better show the ability of the model for reproducing the short time-scale transport dynamics (flashy peaks) for many events (blue envelopes, fig. S1-S9). We also already explained (also in more detail in Rodriguez and Klaus, 2019) that the NSE is not an absolute, universal, and perfectly objective way to estimate model performance (lines 608-617). This is especially true for tracer simulations, for which customized objective functions or visual inspections are sometimes preferred (Stadnyk et al., 2013; Gallart et al., 2016; Rodriguez and Klaus, 2019). It seems that we are simply reaching a difference of opinion about model performance, and we don't see how we can further improve the manuscript on this aspect.

Gallart, F., Roig-Planasdemunt, M., Stewart, M. K., Llorens, P., Morgenstern, U., Stichler, W., Pfister, L., and Latron, J.: A GLUE-based uncertainty assessment framework for tritium-inferred transit time estimations under baseflow conditions, Hydrological Processes, 30, 4741–4760, https://doi.org/10.1002/hyp.10991, 2016

Stadnyk, T.A., Delavau, C., Kouwen, N. and Edwards, T.W.D. (2013), Towards hydrological model calibration and validation: simulation of stable water isotopes using the isoWATFLOOD model, Hydrol. Process., 27: 3791-3810. doi:10.1002/hyp.9695

Such an inadequate model structure underestimates the information content in $^2$H by resulting in not well constrained posterior parameter distributions for the behavioral models.

In the last round of revisions, we discussed that in our opinion, the parameter distributions are constrained to some extent because the posteriors are not flat. This is more precisely and objectively quantified by the fact that we had non-negligible reductions of entropy and information gains $D_{KL}$ (lines 564-566). Thus, it is clear that there is a constraint from the data, however, we agree that uncertainty remains.

If they had a model that captures short time scale dynamics well, posterior distributions of the associated parameters of such a model could be more constrained than a $^3$H based model.

We believe that this is a strong assumption and that it tends to contradict the philosophy underlying uncertainty analysis (e.g., GLUE, DREAM). In particular, this idea contradicts the "bias-variance trade-off" principle. In simple terms, parameter distributions generally tend to get flatter and flatter with increasing model performance past the optimum point defining the adequate trade-off between model performance (here, NSE) and model complexity (here, uncertainty, roughly proportional to the number of parameters) (James et al., 2013). Thus, a model performing better does not automatically imply that the behavioral parameter ranges will be narrower. For example, we could use a new model version with >100 parameters to reach a perfect model performance (e.g., NSE=1). Then, it is very likely that the parameter distributions will all be flat. This is because we would be going far beyond the optimum for the trade-off.

James, G., Witten, D., Hastie, T., & Tibshirani, R. (2013). An introduction to statistical learning with applications in R, (pp. 33–37). New York: Springer-Verlag. https://doi.org/10.1007/978-1-4614-7138-7

Thus, more information can be learned from $^2$H, compared to a $^3$H-based model, than what is described in this paper. Therefore, I disagree with the authors' argument that a better performing model would not change the conclusion of this study. For example, their argument "Tritium was slightly more informative than stable isotopes for travel time analysis despite a lower number of tracer samples" is susceptible to their model structure.

A priori it would appear that more information can be learned from using $^2$H because of the larger number of stream measurements. However, our results show that the larger number of samples is not enough to tip the scales in favor of $^2$H. This is because $^3$H is inherently more informative on travel times due to radioactive decay (lines 21-22, 509-515, 808-810), and because we estimate that not all $^2$H samples were equally informative on travel times (lines 550-553). We disagree that a model working better for simulating $^2$H would necessarily increase the amount of information learned, because this would not necessarily narrow the posterior parameter distributions (see reply to previous comment). It is true that the information contents on travel times for each tracer are model-dependent in our analysis. Ideally, the travel time information contents might be compared using a shape-free (data-based) TTD estimation method (e.g., Kirchner, 2019). However, it is important to notice that there will always be underlying assumptions affecting the final result, similar to the choice of a model structure in our work. For instance, in the case of Kirchner's method, there is an implicit assumption (among others) of multilinearity between tracer inputs and outputs. This is also a model (a statistical model, precisely), which tends to be overlooked, because the method is deemed "model-free" or "data-based". Moreover, there are currently no data-based methods to estimate time-varying TTDs, while working in unsteady conditions is a requirement nowadays.

Following the reviewer's comments, we modified the manuscript accordingly (lines 560-563):

"Finally, the information contents on travel times that we have derived depend on our model structure (number of control volumes and SAS functional form). More work is needed in developing 'model-free' (e.g., data-based) unsteady TTD estimation methods in order to reduce the dependence of the results on modeling assumptions."

and this to the conclusion (lines 740-742):

"More work is also needed to compare the information contents of the tracers on travel times using data-based approaches in order to avoid a dependence on model structure"

Tritium was useless at wet conditions (because they have no samples at wet conditions), and the $^2$H-based model was not constrained well at wet conditions (which underestimate the information content of $^2$H in their analysis method) because of its structural problem.

These claims remain unsubstantiated and we have to reject them. Tritium was useful at "wet conditions". We sampled streamflow during "wet conditions" (see reply to previous comment). Moreover, "the $^2$H-based model was not constrained well at wet conditions" is also not a correct statement. We did not evaluate parameter distributions for "wet conditions", but for the whole study period (2015--2017). We think that perhaps, R2 confuses the concepts of model performance and parameter identifiability (as shown by the previous comments). What R2 meant is probably that the $^2$H-based model struggled more to simulate "wet conditions", probably meaning the flashy events in R2's mind. To this, we can answer that some limitations of the model to simulate the flashy events for $^2$H correspond mostly to drier conditions (see lines 636-639) during which these flashy events are visible (while during wetter conditions they are "drowned" in large flow volumes associated with a more damped isotopic signature).

R2: Also, the authors reported that the $^3$H-based model learned more information (4.47 bits) compared to the $^2$H-based model (which learned 4.08 bits of information). The authors argued that this is because $^3$H informed the model about ET processes more, compared to $^2$H, based on the posterior distributions of the model parameters (in line 504). However, it didn't come with any scientific reason why $^3$H would inform more about the ET processes. I don't think that there is any literature on it, and I personally can't think of any reason. Without a scientific basis, the result seems just an artifact of their model structure and their method of analysis. Why would $^3$H inform more about the ET processes than $^2$H?

We thank R2 for this remark. However, it is important to not confuse ET travel times and ET processes. We never stated anywhere in the manuscript that tritium informed us more about ET processes. Rather, we wrote (line 542): "This is because tritium considerably informed us about the **travel times** in ET", and (lines 806-807): "Tritium was more informative on **travel times** than deuterium due to its stronger constraint on the parameter values of $\Omega_{ET}$, $\mu_{ET}$ and $\theta_{ET}$".

We hope R2 saw the detailed explanations we added on this in the last version (appendix A2). In brief, the parameters of the ET SAS function ($\mu_{ET}$ and $\theta_{ET}$) have a non-negligible influence on the accuracy of streamflow isotopic simulations. This is even more pronounced for tritium simulations because radioactive decay implies that the age selection patterns for ET (i.e., its SAS function parameters) have a stronger influence on the accuracy of the long-term isotope balance in the catchment than for deuterium. We nevertheless realised that one additional sentence was necessary to strengthen the reasoning.

We modified the sentence lines 543-545 to:

"The particularly large information gains on $\mu_{ET}$ and $\theta_{ET}$ with tritium reveal a stronger influence of $\Omega_{ET}$ on the accuracy of stream tracer simulations than for deuterium, via an indirect influence on isotopic partitioning (App. A2)"

We added the following in appendix A2 (lines 806-812):

"Tritium was more informative on travel times than deuterium due to its stronger constraint on the parameter values of $\Omega_{ET}$, $\mu_{ET}$ and $\theta_{ET}$. Based on the reasoning above, this is simply due to the fact that the relationship between T and $C_{P^*}(T,t)$ is clearer for tritium due to its radioactive decay than for deuterium, for which there is essentially no relationship between travel time and tracer concentrations. In conclusion, information on the parameters of $\Omega_{ET}$ exists in the time series of $C_Q(t)$ and can be extracted by calibrating the model based on SAS functions, particularly from using tritium."

R2: MINOR COMMENTS

Line 424: Typo in $S_T \in [0, +\infty$ "[".

This was already stated in the first reviewer report; however, we do not see a typo. Perhaps due to the PDF reader R2 uses?

$$S_T \in [0, +\infty[$$

$$\quad \Omega_Q \text{ (called he}$$

Figure: picture of line 424 (using Foxit Reader)

R2: Line 471: "The travel time and storage measures estimated from a joint use of $^2$H and $^3$H are the highest (tables 3 and 4)." This result is counter-intuitive. Why the joint use gives the highest travel time and storage, not something in the middle?

This is a good remark. Indeed, our intuition may suggest to use the arithmetic mean of the travel time and storage measures for $^2$H and $^3$H when combining both tracers, effectively resulting in a result "in the middle". However, this would be conceptually wrong, because this would correspond to the travel times and storage measures for the simulations constrained by $^2$H or $^3$H, while we are interested in those corresponding to simulations constrained by $^2$H and $^3$H. This can be seen by considering the various sets (populations) of simulations constrained by a given tracer or by a combination of the tracers. When selecting simulations constrained by $^2$H and $^3$H, we are in fact selecting a subset of all behavioral simulations (see Fig. 4). This subset is not a representative sample of the whole population (see Fig. 4), and it contains a particular selection of simulations which favor longer travel times and storage measures (this is called a sampling bias when making polls). This is because $^2$H and $^3$H have information in common about longer travel times.

We added the following after the cited sentence (lines 507-509):

"These measures are not intermediate values (i.e., the average of the results from the individual tracers) because deuterium and tritium have information in common about longer travel times (i.e., the simulations constrained by both tracers are a specific selection among all accepted simulations, see Fig. 4)."

R2: The 2010-2015 data that was used in the spin-up period need to be presented.

Good remark, we added 2 figures in the supplement.

[Figure]

Figure S16. Spin-up data used in the model for deuterium ($\delta^2$H). The 2010-2015 measured data is looped back many times over the 1915-2015 period (black curve, only one repetition is shown). The 2015-2017 data is not used in the spin-up.

[Figure]

Figure S17. Spin-up data used in the model for streamflow (Q) and precipitation (J). The 2010-2015 measured data is looped back many times over the 1915-2015 period (black curve, only ~1 repetition is shown). The 2015-2017 data is not used in the spin-up.

[revised manuscript text omitted]

The mean and standard deviations are calculated from all retained behavioral solutions for a given criterion. [a] Fraction of "young water" (Kirchner, 2016), younger than 0.2 years

**Table 4.** Storage estimate $S_{95P}$ constrained by deuterium or tritium

| Statistics of $S_{95P}$ | $^2$H ($E_2 > 0$) | $^3$H ($E_3 < 0.5$ T.U.) | $^2$H and $^3$H |
|---|---|---|---|
| Mean $\pm$ st. dev. [mm] | 1275 $\pm$ 245 | 1335 $\pm$ 279 | 1488 $\pm$ 135 |
| Median $\pm$ st. dev. [mm] | 1281 $\pm$ 245 | 1392 $\pm$ 279 | 1505 $\pm$ 135 |
| Min [mm] | 625 | 660 | 1249 |
| Max [mm] | 1744 | 1806 | 1710 |

$S_{95P}$ is calculated as the 95[th] percentile of $\Omega_{tail}$ (eq. 11)

**Table B1.** Results from the Wilcoxon rank sum test comparing the travel time and storage measures between $^2$H and $^3$H behavioral solutions. The null hypothesis is that the measures are extracted from the same underlying distribution for both tracers.

| Travel time or storage measure | Decision about the null hypothesis | p-value |
|---|---|---|
| 10$^{th}$ percentile | Rejected | $3.3 \times 10^{-6}$ |
| 25$^{th}$ percentile | Rejected | $5.9 \times 10^{-8}$ |
| Median | Rejected | $1.5 \times 10^{-8}$ |
| 75$^{th}$ percentile | Rejected | $1.1 \times 10^{-3}$ |
| 90$^{th}$ percentile | Accepted | 0.30 |
| Mean | Rejected | $3.5 \times 10^{-5}$ |
| $F_{yw}$[a] | Accepted | 0.37 |
| F(T < 6 months) | Rejected | $5.3 \times 10^{-6}$ |
| F(T < 1 year) | Rejected | $2.7 \times 10^{-10}$ |
| F(T < 3 years) | Rejected | $2.5 \times 10^{-3}$ |
| $S_{95P}$ | Rejected | $1.4 \times 10^{-2}$ |

All tests were made at the 5% significance level.

[a] Fraction of "young water" (Kirchner, 2016), younger than 0.2 years